# THE ROLE OF MINIMAL COMPLEXITY FUNCTIONS IN UNSUPERVISED LEARNING OF SEMANTIC MAPPINGS

**Tomer Galanti**
The Blavatnik School of Computer Science
Tel Aviv University
Tel Aviv, Israel
tomerga2@post.tau.ac.il

**Lior Wolf**
Facebook AI Research &
The Blavatnik School of Computer Science
Tel Aviv University
Tel Aviv, Israel
wolf@fb.com
wolf@cs.tau.ac.il

**Sagie Benaim**
The Blavatnik School of Computer Science
Tel Aviv University
Tel Aviv, Israel
sagieb@mail.tau.ac.il

## ABSTRACT

We discuss the feasibility of the following learning problem: given unmatched samples from two domains and nothing else, learn a mapping between the two, which preserves semantics. Due to the lack of paired samples and without any definition of the semantic information, the problem might seem ill-posed. Specifically, in typical cases, it seems possible to build infinitely many alternative mappings from every target mapping. This apparent ambiguity stands in sharp contrast to the recent empirical success in solving this problem.

We identify the abstract notion of aligning two domains in a semantic way with concrete terms of minimal relative complexity. A theoretical framework for measuring the complexity of compositions of functions is developed in order to show that it is reasonable to expect the minimal complexity mapping to be unique. The measured complexity used is directly related to the depth of the neural networks being learned and a semantically aligned mapping could then be captured simply by learning using architectures that are not much bigger than the minimal architecture.

Various predictions are made based on the hypothesis that semantic alignment can be captured by the minimal mapping. These are verified extensively. In addition, a new mapping algorithm is proposed and shown to lead to better mapping results.

## 1 INTRODUCTION

Multiple recent reports (Xia et al., 2016; Kim et al., 2017; Zhu et al., 2017; Yi et al., 2017) convincingly demonstrated that one can learn to map between two domains that are each specified merely by a set of unlabeled examples. For example, given a set of unlabeled images of horses, and a set of unlabeled images of zebras, CycleGAN (Zhu et al., 2017) creates the analog zebra image for a new image of a horse and vice versa.

These recent methods employ two types of constraints. First, when mapping from one domain to another, the output has to be indistinguishable from the samples of the new domain. This is enforced using GANs (Goodfellow et al., 2014) and is applied at the distribution level: the mapping of horse images to the zebra domain should create images that are indistinguishable from the training images of zebras and vice versa. The second type of constraint enforces that for every single sample, transforming it to the other domain and back (by a composition of the mappings in the two directions) results in the original sample. This is enforced for each training sample from either domain: every training image of a horse (zebra), which is mapped to a zebra (horse) image and then back to the source domain, should be as similar as possible to the original input image.

In another example, taken from DiscoGAN (Kim et al., 2017), a function is learned to map a handbag to a shoe of a similar style. One may wonder why striped bags are not mapped, for example, to shoes with a checkerboard pattern. If every striped pattern in either domain is mapped to a checkerboard pattern in the other and vice-versa, then both the distribution constraints and the circularity constraints might hold. The former could hold since both striped and checkerboard patterned objects would be generated. Circularity could hold since, for example, a striped object would be mapped to a checkerboard object in the other domain and then back to the original striped object.

One may claim that the distribution of striped bags is similar to those of striped shoes and that the distribution of checkerboard patterns is also the same in both domains. In this case, the alignment follows from fitting the shapes of the distributions. This explanation is unlikely, since no effort is being made to create handbags and shoes that have the same distributions of these properties, as well as many other properties.

Our work is dedicated to the alternative hypothesis that the target mapping is implicitly defined by being approximated by the lowest-complexity mapping that has a low discrepancy between the mapped samples and the target distribution, i.e., the property that even a good discriminator cannot distinguish between the generated samples and the target ones. In Sec. 2 we explore the inherent ambiguity of cross domain mapping. In Sec. 3, we present the hypothesis and two verifiable predictions, as well as a new unsupervised mapping algorithm. In Sec. 4, we show that the number of minimal complexity mappings is expected to be small. Sec. 5 verifies the various predictions. Some context to our work, including classical ideas such as Occam's Razor, MDL, and Kolmogorov complexity are discussed in Sec. 6.

## 2 THE UNSUPERVISED ALIGNMENT PROBLEM

The learning algorithm is provided with only two unlabeled datasets: one includes i.i.d samples from the first distribution and the second includes i.i.d samples from the other distribution (all notations are listed in Appendix B, Tab. 5).

$$x_i \in \mathcal{X}_A \text{ for } i = 1 \ldots m \text{ where } x_i \overset{\text{i.i.d}}{\sim} D_A \text{ and } \mathcal{X}_A \text{ denotes the space of domain } A = (\mathcal{X}_A, D_A)$$

$$x_j \in \mathcal{X}_B \text{ for } j = 1 \ldots n \text{ where } x_j \overset{\text{i.i.d}}{\sim} D_B \text{ and } \mathcal{X}_B \text{ denotes the space of domain } B = (\mathcal{X}_B, D_B) \tag{1}$$

To semantically tie the two distributions together, a generative view can be taken. This view is well aligned with the success of GAN-based image generation, e.g., (Radford et al., 2015), in mapping random input vectors into realistic-looking images. Let $z \in \mathcal{X}$ be a random vector that is distributed according to the distribution $D_Z$ and which we employ to denote the semantic essence of samples in $\mathcal{X}_A$ and $\mathcal{X}_B$. We denote $D_A = y_A \circ D_Z$ and $D_B = y_B \circ D_Z$, where the functions $y_A : \mathcal{X} \to \mathcal{X}_A$ and $y_B : \mathcal{X} \to \mathcal{X}_B$ (see Fig. 1), and $f \circ D$ denotes the distribution of $f(x)$, where $x \sim D$. Following the circularity-based methods (Xia et al., 2016; Kim et al., 2017; Zhu et al., 2017; Yi et al., 2017), we assume that both $y_A$ and $y_B$ are invertible.

The assumption of invertibility is further justified by the recent success of supervised pre-image computation methods (Dosovitskiy & Brox, 2016). In unsupervised learning, given training samples, one may be expected to be able to recover the underlying properties of the generated samples, even with very weak supervision (Chen et al., 2016). However, if the target function between domains $A$ and $B$ is not invertible, because for each member of $A$ there are a few possible members of $B$ (or vice versa), we can add a stochastic component to $A$ that is responsible for choosing which member in $B$ to take, given a member of $A$. For example, if $A$ is a space of handbag images and $B$ is a space of shoes, such that for every handbag, there are a few analogous shoes, then a stochastic variable can be added such that given a handbag, one shoe is selected among the different analog shoes.

We denote by $y_{AB} = y_B \circ y_A^{-1}$, the function that maps the first domain to the second domain. It is semantic in the sense that it goes through the shared semantic space $\mathcal{X}$. The goal of the learner is to fit a function $h \in \mathcal{H}$, for some hypothesis class $\mathcal{H}$ that is closest to $y_{AB}$,

$$\inf_{h \in \mathcal{H}} R_{D_A}[h, y_{AB}], \tag{2}$$

where $R_D[f_1, f_2] = \mathbb{E}_{x \sim D} \ell(f_1(x), f_2(x))$, for a loss function $\ell : \mathbb{R} \times \mathbb{R} \to \mathbb{R}$ and a distribution $D$.

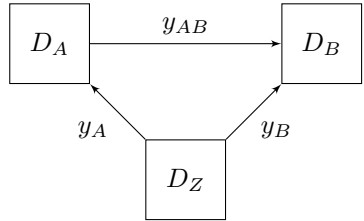

Figure 1: The mappings between the domains $A$, $B$, and $Z$.

It is not clear that such fitting is possible without further information. Assume, for example, that there is a natural order on the samples in $\mathcal{X}_B$. A mapping that transforms an input sample $x \in \mathcal{X}_A$ to the sample that is next in order to $y_{AB}(x)$, could be just as feasible. More generally, one can permute the samples in $\mathcal{X}_A$ by some function $\Pi$ that replaces each sample with another sample that has a similar likelihood (see Def. 1 below) and learn $h$ that satisfies $h = y_{AB} \circ \Pi$. We call this difficulty "the alignment problem" and our work is dedicated to understanding the plausibility of learning despite this problem.

In multiple recent contributions (Xia et al., 2016; Kim et al., 2017; Zhu et al., 2017; Yi et al., 2017) circularity is employed. Circularity requires the recovery of both $y_{AB}$ and $y_{BA} = y_A \circ y_B^{-1}$ simultaneously. Namely, functions $h$ and $h'$ are learned jointly by minimizing the risk:

$$\inf_{h,h' \in \mathcal{H}} \text{disc}_{\mathcal{C}}(h \circ D_A, D_B) + \text{disc}_{\mathcal{C}}(h' \circ D_B, D_A)$$
$$+ R_{D_A}[h' \circ h, \text{Id}_A] + R_{D_B}[h \circ h', \text{Id}_B] \tag{3}$$

where $\text{disc}_{\mathcal{C}}(D_1, D_2) = \sup_{c_1, c_2 \in \mathcal{C}} |R_{D_1}[c_1, c_2] - R_{D_2}[c_1, c_2]|$ denotes the discrepancy between distributions $D_1$ and $D_2$ that is implemented with a GAN (Ganin et al., 2016).

The first term in Eq. 3 ensures that the samples generated by mapping domain $A$ to domain $B$ follow the distribution of samples in domain $B$. The second term is the analog term for the mapping in the other direction. The last two terms ensure that mapping a sample from one domain to the second and back, results in the original sample.

While the circularity constraints, expressed as the last two terms in Eq. 3, are elegant and do not require additional supervision, for every invertible permutation $\Pi$ of the samples in domain $B$ (not to be confused with a permutation of the vector elements of the representation of samples in $B$) we have

$$(h' \circ \Pi^{-1}) \circ (\Pi \circ h) = h \circ h' \approx \text{Id}_A, \text{ and}$$
$$(\Pi \circ h) \circ (h' \circ \Pi^{-1}) = \Pi \circ (h \circ h') \circ \Pi^{-1} \approx \Pi \circ \text{Id}_B \circ \Pi^{-1} = \text{Id}_B. \tag{4}$$

Therefore, every circularity preserving pair $h$ and $h'$ gives rise to many possible solutions of the form $\tilde{h} = h \circ \Pi$ and $\tilde{h}' = \Pi^{-1} \circ h'$. If $\Pi$ happens to satisfy $D_B(x) \approx D_B(\Pi(x))$, then the discrepancy terms in Eq. 3 also remain largely unchanged. Circularity by itself cannot, therefore, explain the recent success of unsupervised mapping.

## 3 THE SIMPLICITY HYPOTHESIS

Despite the availability of a large number of alternative hypotheses $h'$ that satisfy the constraints of Eq. 3, the methods of Xia et al. (2016); Kim et al. (2017); Zhu et al. (2017); Yi et al. (2017) enjoy empirical success, Why?

Our hypothesis is that the small-discrepancy mapping of the lowest complexity approximates the alignment of the target function. We further hypothesize that when performing research in unsupervised mapping, goldilock architectures are selected. These architectures are complex enough to allow small discrepancies but not complex enough to support mappings that are not minimal in complexity. By doing so, one of the minimal-complexity low-discrepancy mappings is learned.

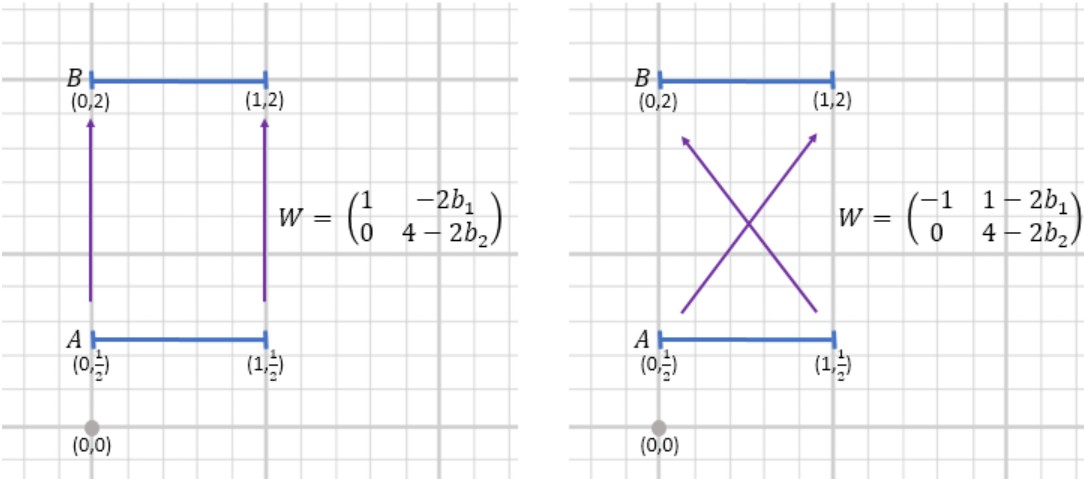

Figure 2: An illustrative example where the two domains are line segments in $\mathbb{R}^2$. There are infinitely many mappings that preserve the uniform distribution on the two segments. However, only two stand out as "semantic". These are exactly the two mappings that can be captured by a neural network with only two hidden neurons and Leaky ReLU activations, i.e., by a function $h(x) = \sigma_a(Wx + b)$, for a weight matrix $W$ and the bias vector $b$.

### 3.1 AN ILLUSTRATIVE EXAMPLE

In order to illustrate our hypothesis, we present a very simple toy example, depicted in Fig. 2. Consider the domain $A$ of uniformly distributed points $(x_1, x_2)^\top \in \mathbb{R}^2$, where $0 \leq x_1 < 1$ and $x_2 = 0.5$. Let $B$ be a similar domain, except $x_2 = 2$. We are interested in learning the mapping $y_{AB}^{2D}((x_1, 0.5)^\top) = (x_1, 2)^\top$. We note that there are infinitely many mappings from domain $A$ to $B$ that satisfy the constraints of Eq. 3.

However, when we learn the mapping using a neural network with one hidden layer of size 2, and Leaky ReLU activations[1] (Maas et al., 2013), $y_{AB}^{2D}$ is one of only two options. In this case $h(x) = \sigma_a(Wx + b)$, for $W \in \mathbb{R}^{2\times2}, b \in \mathbb{R}^2$ and where $\sigma_a$ is applied per coordinate. The only admissible solutions are of the form $W_b = \begin{pmatrix} 1 & -2b_1 \\ 0 & 4 - 2b_2 \end{pmatrix}$ or $W_b' = \begin{pmatrix} -1 & 1 - 2b_1 \\ 0 & 4 - 2b_2 \end{pmatrix}$ and $b = (b_1, b_2)^\top$, which are identical, for every $b$, to $y_{AB}^{2D}$ or to an alternative $y_{AB}^{2D'}((x_1, 0.5)^\top) = (1 - x_1, 2)^\top$. Exactly the same situation holds for any pair of line segments in $\mathbb{R}_+^d$.

Therefore, by restricting the hypothesis space of $h$, we eliminate all alternative solutions, except two. These two are exactly the two mappings that would commonly be considered "more semantic" than any other mapping, and can be expressed as the simplest possible mapping through a shared one dimensional space. While this is an extreme example, we believe that the principle is general since limiting the complexity of the admissible solutions eliminates the solutions that are derived from $y_{AB}$ by permuting the samples in the space $\mathcal{X}_A$, because such mixing requires added complexity.

### 3.2 A COMPLEXITY MEASURE FOR FUNCTIONS

In this work, we focus on functions of the form

$$f := F[W_{n+1}, ..., W_1] = W_{n+1} \circ \sigma \circ ... \circ \sigma \circ W_2 \circ \sigma \circ W_1 \qquad (5)$$

here, $W_1, ..., W_{n+1}$ are invertible linear transformations from $\mathbb{R}^M$ to itself. In addition, $\sigma$ is a non-linear element-wise activation function. We will mainly focus on $\sigma$ that is Leaky ReLU with parameter $0 < a \neq 1$. In addition, for any function $f$, we define the complexity of $f$, denoted by

---

[1]$\sigma_a(x) = \text{Ind}[x < 0]ax + \text{Ind}[x \geq 0]x$, for the indicator function $\text{Ind}[q]$ which maps a true value to one, zero otherwise.

$C(f)$ as the minimal number $n$ such that there are invertible linear transformations $W_1, ..., W_{n+1}$ that satisfy $f = F[W_{n+1}, ..., W_1]$.

Our function complexity framework, therefore, measures the complexity of a function as the depth of a neural network which implements it, or the shallowest network, if there are multiple such networks. In other words, we use the number of layers of a network as a proxy for the Kolmogorov complexity of functions, using layers in lieu of the primitives of the universal Turing machines, which is natural for studying functions that can be computed by feedforward neural networks.

Note that capacity is typically controlled by means of norm regularization, which is optimized during training. Here, the architecture is bounded to a certain number of layers. This measure of complexity is intuitive and provides a clear and stable stratification of functions.

Norm capacity (for norms larger than zero) are not effective in comparing functions of different architectures. In Sec. 5, we demonstrate that the L1 and L2 norms of the desired mapping are within the range of norms that are obtained when employing bigger or smaller architectures. Other ways to define the complexity of functions, such as the VC-dimension (Vapnik & Chervonenkis, 1971b) and Rademacher complexity (Bartlett & Mendelson, 2003), are not suitable for measuring the complexity of individual functions, since their natural application is in measuring the capacity of classes of functions.

### 3.3    CONSEQUENCES OF THE SIMPLICITY HYPOTHESIS

The simplicity hypothesis leads to concrete predictions, which are verified in Sec. 5. The first one states that in contrast to the current common wisdom, one can learn a semantically aligned mapping between two spaces without any matching samples and even without circularity.

**Prediction 1.** *When learning with a small enough network in an unsupervised way a mapping between domains that share common characteristics, the GAN constraint in the target domain is sufficient to obtain a semantically aligned mapping.*

The strongest clue that helps identify the alignment of the semantic mapping from the other mappings is the suitable complexity of the network that is learned. A network with a complexity that is too low cannot replicate the target distribution, when taking inputs in the source domain (high discrepancy). A network that has a complexity that is too high, would not learn the minimal complexity mapping, since it could be distracted by other alignment solutions.

We believe that the success of the recent methods results from selecting the architecture used in an appropriate way. For example, DiscoGAN (Kim et al., 2017) employs either eight or ten layers, depending on the dataset. We make the following prediction:

**Prediction 2.** *When learning in an unsupervised way a mapping between domains, the complexity of the network needs to be carefully adjusted.*

This prediction is also surprising, since in supervised learning, extra depth is not as detrimental, if at all. As far as we know, this is the first time that this clear distinction between supervised and unsupervised learning is made [2].

### 3.4    ALIGNMENT WITH NON-MINIMAL ARCHITECTURES

If the simplicity hypothesis is correct, then in order to capture the target alignment, one would need to learn with the minimal complexity architecture that supports a small discrepancy. However, deeper architectures can lead to even smaller discrepancies and to better outcomes.

In order to enjoy both the alignment provided by our hypothesis and the improved output quality, we propose to find a function $h$ of a non-minimal complexity $k_2$ that minimizes the following objective function

$$\min_{h \text{ s.t } C(h)=k_2} \left\{ \text{disc}(h \circ D_A, D_B) + \lambda \inf_{g \text{ s.t } C(g)=k_1} R_{D_A}[h, g] \right\} \tag{6}$$

---

[2]The MDL literature was developed when people believed that small hypothesis classes are desired for both supervised and unsupervised learning.

where $k_1$ is the minimal complexity for mapping with low discrepancy between domain $A$ and domain $B$. In other words, we suggest to find a function $h$ that is both a high complexity mapping from domain $A$ to $B$ and is close to a function of low complexity that has low discrepancy.

There are alternative ways to implement an algorithm that minimizes the objective function presented in Eq. 6. Assuming, based on this equation, that for $h$ that minimizes the objective function, the corresponding $g^* = \underset{g \text{ s.t } C(g) = k_1}{\arg\inf} R_{D_A}[h, g]$ has a (relatively) small discrepancy, leads to a two-step algorithm. The algorithm first finds a function $g$ that has small complexity and small discrepancy and then finds $h$ of a larger complexity that is close to $g$. This is implemented in Alg. 1. Note that in the first step, $k_1$ is being estimated, for example, by gradually increasing its value, until $g$ with a discrepancy lower than a threshold $\epsilon_0$ is found. We suggest to use a liberal threshold, since the goal of the network $g$ is to provide alignment and not the lowest possible discrepancy.

---

**Algorithm 1** Complexity Based Regularization Alignment

---

**Require:** Unlabeled training sets $S_A \overset{\text{i.i.d}}{\sim} D_A^m$ and $S_B \overset{\text{i.i.d}}{\sim} D_B^n$, a desired complexity $k_2$, and a trade-off parameter $\lambda$

1: Identify a complexity $k_1$, which leads to a small discrepancy $\underset{g \text{ s.t: } C(g) = k_1}{\min} \text{disc}(g \circ D_A, D_B)$.

2: Train $g$ of complexity $k_1$ to minimize $\text{disc}(g \circ D_A, D_B)$.

3: Train $h$ of complexity $k_2$ to minimize $\text{disc}(h \circ D_A, D_B) + \lambda R_{D_A}[h, g]$.

---

## 4 COUNTING MINIMAL COMPLEXITY MAPPINGS

Recall, from Sec. 2, that disc is the discrepancy distance, which is based on the optimal discriminator. Also discussed were the functions $\Pi$, that switches between members in the domain $B$ that have similar probabilities. These are defined using the discrepancy distance as follows (simplified version; the definitions and results of this section are stated more broadly in Appendix A):

**Definition 1** (Density preserving mapping). *Let $X = (\mathcal{X}, D_X)$ be a domain. A $\epsilon_0$-density preserving mapping over $X$ (or an $\epsilon_0$-DPM for short) is a function $\Pi$ such that*

$$\text{disc}(\Pi \circ D_X, D_X) \leq \epsilon_0 \tag{7}$$

*We denote the set of all $\epsilon_0$-DPMs of complexity $k$ by $\text{DPM}_{\epsilon_0}(X; k) := \big\{ \Pi \big| \text{disc}(\Pi \circ D_X, D_X) \leq \epsilon_0 \text{ and } C(\Pi) = k \big\}$.*

Below, we define a similarity relation between functions that reflects whether the two are similar. In this way, we are able to bound the number of different (non-similar) minimal complexity mappings by the number of different DPMs.

**Definition 2.** *Let $D$ be a distribution. We denote $f \overset{D}{\underset{\epsilon_0}{\sim}} g$, if $C(f) = C(g)$ and there are minimal decompositions: $f = F[W_{n+1}, ..., W_1]$ and $g = F[V_{n+1}, ..., V_1]$ such that: $\forall i \in [n+1] : \text{disc}(F[W_i, ..., W_1] \circ D, F[V_i, ..., V_1] \circ D) \leq \epsilon_0$.*

Put differently, two functions of the same complexity have this relation, if for every step of their processing, the activations of the matching functions are similar.

The defined relation is reflexive and symmetric, but not transitive. Therefore, there are many different ways to partition the space of functions into disjoint subsets such that in each subset, any two functions have the closeness property. We count the number of functions as the minimal number of subsets required in order to cover the entire space. This quantity is denoted by $N(\mathcal{U}, \sim_{\mathcal{U}})$ where $\mathcal{U}$ is the set and $\sim_{\mathcal{U}}$ is the closeness relation. The formal presentation is in Def. 9, which slightly generalizes the notion of covering numbers (Anthony & Bartlett, 2009).

Informally, the following theorem states that the number of minimal low-discrepancy mappings is upper bounded by both the number of DPMs of a certain size over $D_A$ and over $D_B$. This result is useful, since DPMs are expected to be rare in real-world domains. When imagining mapping a space to itself, in a way that preserves the distribution, one first considers symmetries. Near-perfect symmetries are rare in natural domains, and when these occur, e.g., (Kim et al., 2017), they form well-understood ambiguities. Another option that can be considered is that of replacing specific samples

in domain $B$ with other samples of the same probability. However, these very local discontinuous mappings are of very high complexity, since this complexity is required for reducing the modeling error for discontinuous functions. One can also consider replacing larger sub-domains with other sub-domains such that the distribution is preserved. This could be possible, for example, if the distribution within the sub-domains is almost uniform (unlikely), or if it is estimated inaccurately due to the limitations of the training set.

We, therefore, make the following prediction.

**Prediction 3.** *The number of DPMs of low complexity is small.*

Given two domains $A$ and $B$, there is a certain complexity $C_{A,B}^{\epsilon_0}$, which is the minimal complexity of the networks needed in order to achieve discrepancy smaller than $\epsilon_0$ for mapping the distribution $D_A$ to the distribution $D_B$. The set of minimal complexity mappings, i.e., mappings of complexity $C_{A,B}^{\epsilon_0}$ that achieve $\epsilon_0$ discrepancy is denoted by $H_{\epsilon_0}(A, B) :=$ $\left\{ h \mid C(h) \leq C_{A,B}^{\epsilon_0} \text{ and } \mathrm{disc}(h \circ D_A, D_B) \leq \epsilon_0 \right\}$. The following theorem shows that the covering number of this set is similar to the covering number of the DPMs. Therefore, if prediction 3 above holds, the number of minimal low-discrepancy mappings is small.

**Theorem 1** (Informal). *Let $\sigma$ be a Leaky ReLU with parameter $0 < a \neq 1$ and assume identifiability. Let $\epsilon_0$, $\epsilon_1$ and $\epsilon_2 < \epsilon_1$ be three positive constants and $A = (\mathcal{X}_A, D_A)$ and $B = (\mathcal{X}_B, D_B)$ are two domains. Then,*

$$
\mathrm{N}\left( H_{\epsilon_0}(A, B), \underset{\epsilon_1}{\overset{D_A}{\sim}} \right) \leq \min \begin{cases} \mathrm{N}\left( \mathrm{DPM}_{\epsilon_0}\left( A; 2C_{A,B}^{\epsilon_0} \right), \underset{\epsilon_2}{\overset{D_A}{\sim}} \right) \\ \mathrm{N}\left( \mathrm{DPM}_{\epsilon_0}\left( B; 2C_{A,B}^{\epsilon_0} \right), \underset{\epsilon_2}{\overset{D_B}{\sim}} \right) \end{cases} \tag{8}
$$

*Proof.* See Appendix D.

The theorem assumes identifiability. In the context of neural networks, the general question of uniqueness up to invariants, also known as identifiability, is an open question. Several authors have made progress in this area for different neural network architectures. The most notable work has been done by Fefferman & Markel (1993) that proves identifiability for $\sigma = \tanh$. Furthermore, the representation is unique up to some invariants. Other works (Williamson & Helmke, 1995; F. Albertini & Maillot, 1993; Kurková & Kainen, 2014; Sussmann, 1992) prove such uniqueness for neural networks with only one hidden layer and various activation functions. Similarly, in Lem. 3 in the Appendix, we show that identifiability holds for Leaky ReLU networks with one hidden layer.

## 5 EXPERIMENTS

The first group of experiments is dedicated to test the validity of the three predictions made, in order to give further support to the simplicity hypothesis. Next, we evaluate the success of the proposed algorithm in comparison to the DiscoGAN method of Kim et al. (2017).

We chose to experiment with the DiscoGAN architecture since it focuses on semantic tasks that contain a lesser component of texture or style transfer. The CycleGAN architecture of Zhu et al. (2017) inherits much from the style transfer architecture of Pix2Pix Isola et al. (2017), and the discrepancy term is based on a patch-based analysis, which introduces local constraints that could mask the added freedom introduced by adding layers. In addition, the U-net architecture of Ronneberger et al. (2015) used by Isola et al. (2017) deviates from the connectivity pattern of our model.

Experiments in this architecture and with the architecture of DualGAN (Yi et al., 2017), which focuses on tasks similar to CycleGAN, and shares many of the architectural choices, including U-nets and the use of patches, are left for future work.

### 5.1 EMPIRICAL VALIDATION OF THE PREDICTIONS

Prediction 1 states that since the unsupervised mapping methods are aimed at learning minimal complexity low discrepancy functions, GANs are sufficient. In the literature (Zhu et al., 2017; Kim et al., 2017), learning a mapping $h : \mathcal{X}_A \to \mathcal{X}_B$, based only on the GAN constraint on $B$, is presented

as a failing baseline. In (Yi et al., 2017), among many non-semantic mappings obtained by the GAN baseline, one can find images of GANs that are successful. However, this goes unnoticed.

In order to validate the prediction that a purely GAN based solution is viable, we conducted a series of experiments using the DiscoGAN architecture and GAN loss only. We consider image domains $A$ and $B$, where $\mathcal{X}_A = \mathcal{X}_B = \mathbb{R}^{3 \times 64 \times 64}$.

In DiscoGAN, the generator is built of: (i) an encoder consisting of convolutional layers with $4 \times 4$ filters followed by Leaky ReLU activation units and (ii) a decoder consisting of deconvolutional layers with $4 \times 4$ filters followed by a ReLU activation units. Sigmoid is used for the output layer. Between four to five convolutional/deconvolutional layers are used, depending on the domains used in $A$ and $B$ (we match the published code architecture per dataset). The discriminator is similar to the encoder, but has an additional convolutional layer as the first layer and a sigmoid output unit.

The first set of experiments considers the CelebA face dataset. Transformations are learned between the subset of images labeled as male and those labeled as female, as well as from blond to black hair and eyeglasses to no eyeglasses. The results are shown in Fig. 3, 4, and 5, (resp.). It is evident that the output image is highly related to the input images.

In the case of mapping handbags to shoes, as seen in Fig. 6, the GAN does not provide a meaningful solution. However, in the case of edges to shoes and vice versa (Fig. 7), the GAN solution is successful.

In Prediction 2, we predict that the selection of the right number of layers is crucial in unsupervised learning. Using fewer layers than needed, will not support the modeling of the target alignment between the domains. In contrast, adding superfluous layers would mean that more and more alternative mappings obscure the target transformation.

In (Kim et al., 2017), 8 or 10 layers are employed (counting both convolution and deconvolution) depending on the experiment. In our experiment, we vary the number of layers and inspect the influence on the results. The experiments are also repeated for the Wasserstein GAN loss (using the same architecture) in Appendix E.

These experiments were done on the CelebA gender conversion task, where 8 layers are employed in the experiments of (Kim et al., 2017). Using the public implementation and adding and removing layers, we obtain the results in Fig. 8– 13. Note that since the encoder and the decoder parts of the learned network are symmetrical, the number of layers is always even. As can be seen, changing the number of layers has a dramatic effect on the results. The best results are obtained at 6 or 8 layers with 6 having the best alignment and 8 having better discrepancy. The results degrade quickly, as one deviates from the optimal value. Using fewer layers, the GAN fails to produce images of the desired class. Adding layers, the semantic alignment is lost, just as expected.

Note that Kim et al. (2017) have preferred low discrepancy over alignment in their choice. In other words, the selected architecture of size $k = 8$ presents acceptable images at the price of lower alignment compared to an architecture of size $k - 2$. This is probably a result of ambiguity that is already present at the size $k$ architecture. On the other hand, the smaller architecture of size $k - 2$ does not produce images of extremely low discrepancy, and there is no architecture that benefits both, an extremely low discrepancy and high alignment. This is observed for example in Fig. 8 where females are translated to males. For 4 layers the discrepancy is too low and the mapping fails to produce images of males. For 6 layers, the discrepancy is relatively low and the alignment is at its highest. For 8 layers, the discrepancy is at its lowest value, nevertheless, the alignment is worse.

While our discrete notion of complexity seems to be highly related to the quality of the results, the norm of the weights do not seem to point to a clear architecture, as shown in Tab. 2(a). Since the table compares the norms of architectures of different sizes, we also approximated the functions using networks of a fixed depth $k = 18$ and then measured the norm. These results are presented in Tab. 2(b). In both cases, the optimal depth, which is 6 or 8, does not appear to have a be an optimum in any of the measurements.

Prediction 3 states that there are only a handful of DPMs, except for the identity function. In order to verify it, we trained a DiscoGAN from a distribution $A$ to itself with an added loss of the form $-\sum_{x \in A} |x - h(x)|$. In our experiment, testing network complexities from 2 to 12, we could not find a DPM, see Fig. 16 and Tab. 3. For lower complexities, the identity was learned despite the added

loss. For higher complexities, the network learned the identity while changing the background color. For even higher complexities, other mapping emerged. However, these mappings did not satisfy the circularity constraint, and are unlikely to be DPMs.

## 5.2 RESULTS FOR ALG. 1

The goal of Alg. 1 is to find a well-aligned solution with higher complexity than the minimal solution and potentially smaller discrepancy. It has two stages. In the first one, $k_1$, which is the minimal complexity that leads to a low discrepancy, is identified. This follows a set of experiments that are similar to the one that is captured, for example, by Fig. 2. To demonstrate robustness, we select a single value of $k_1$ across all experiments. Specifically, we use $k_1 = 6$, which, as discussed above, typically leads to a low (but not very low) discrepancy, while the alignment is still unambiguous.

Once $g$ is trained, we proceed to the next step of optimizing a second network of complexity $k_2$. Note that while the first function ($g$) uses the complete DiscoGAN architecture, the second network ($h$) only employs a one-directional mapping, since alignment is obtained by $g$. Figs. 21– 29 depict the obtained results, for a varying number of layers. First, the result obtained by the DiscoGAN method with $k_1$ is displayed. The results of applying Alg. 1 are then displayed for a varying $k_2$.

As can be seen, our algorithm leads to more sophisticated mappings. Kim et al. (2017) have noted that their solutions are, at many times, related to texture or style transfer and, for example, geometric transformations are not well captured. The new method is able to better capture such complex transformations. Consider the case of mapping male to female in Fig. 20, first row. A man with a beard is mapped to a female image. While for $g$ the beard is still somewhat present, it is not so for $h$ with $k_2 > k_1$. On the female to male mappings in Fig. 21 it is evident in most mappings that $g$ produces a more blurred image, while $h$ is more coherent for $k_2 > k_1$. Another example is in the blond to black hair mapping in Fig. 22. In the 5th row, the style transfer nature of $g$ is evident, since it maps a red object behind the head together with the whole blond hair, producing an unrealistic black hair. $h$ of complexity $k_2 = 8$ is able to separate that object from the hair, and in $k_2 > 8$ it produces realistic looking black hair. This kind of transformation requires more than a simple style transfer. On the edges to shoes and edges to handbags mappings of Fig. 26 and Fig. 28, while the general structure is clearly present, it is significantly sharpened by mapping $h$ with $k_2 > k_1$.

For the face datasets, we also employ face descriptors in order to learn whether the mapping is semantic. Namely, we can check if the identity is preserved post mapping by comparing the VGG face descriptors of Parkhi et al. (2015). One can assume that two images that match will have many similar features and so the VGG representation will be similar. The cosine similarities are used, as is commonly done.

In addition, we train a linear classifier in the space of the VGG face descriptors in order to distinguish between Male/Female, Eyeglasses/No-eyeglasses, and Blond/Black. This way, we can check, beyond discrepancy, that the mapping indeed transforms between the domains. The training samples in domains $A$ and $B$ are used to train this classifier, which is then applied to a set of test images before and after mapping, measuring the accuracy. The higher the accuracy, the better the separation.

Tab. 4 presents the results for both the $k_1$ layers network $g$, alternative networks $g$ of higher complexity (shown as baseline only), and the network $h$ trained using Alg. 1. We expect the alignment of $g$ to be best at complexity $k_1$, and worse due to the loss of discrepancy for alternative network $g$ with complexity $k > k_1$. We expect this loss of alignment to be resolved for networks $h$ trained with Alg. 1.

In the experiments of black to blond hair and blond to black hair mappings, we note that $h$ with $k_2 = 8$ has the best descriptor similarity, and very good separation accuracy and discrepancy. Higher values of $k_2$ are best in terms of separation accuracy and discrepancy, but lose somewhat in descriptor similarity. A similar situation occurs for male to female and female to male mappings and in eyeglasses to non-eyeglasses, where $k_2 = 8$ results in the best similarity score and higher values of $k_2$ result in better separation accuracy and discrepancy.

It is interesting to note, that the distance between $g$ and $h$ is also minimal for $k_2 = 8$. Perhaps, with more effective optimization, higher complexities could also maintain similarity, while delivering lower discrepancies.

## 6 DISCUSSION

Our stratified complexity model is related to structural risk minimization (SRM) by Vapnik & Chervonenkis (1971a), which employs a hierarchy of nested subsets of hypothesis classes in order of increasing complexity. In our stratification, which is based on the number of layers, the complexity classes are not necessarily nested. A major emphasis in SRM is the dependence on the number of samples: the algorithm selects the hypothesis from one of the nested hypothesis classes depending on the amount of training data. In our case, one can expect higher values of $k_2$ to be beneficial as the number of training samples grows. However, the exact characterization of this relation is left for future work.

Alg. 1 can be seen as a form of distillation. The first step of the algorithm finds the minimal complexity for mapping between the two domains and obtains the first generator. Then, a second generator, with a large complexity, is trained while being encouraged to output images which are close to the output of the first generator. This resembles the distillation methods proposed by Hinton et al. (2015) and later analyzed by Hand & Voroninski (2017).

Since the method depicted in Alg. 1 optimizes, among other things, the architecture of the network, our method is somewhat related to work that learn the network's structure during training, e.g., (Saxena & Verbeek, 2016; Wen et al., 2016; Liu et al., 2015; Feng & Darrell, 2015; Lebedev & Lempitsky, 2016). This body of work, which deals exclusively with supervised learning, optimizes the networks loss by modifying both the parameters and the hyperparameters. For GAN based loss, this would not work, since with more capacity, one can reduce the discrepancy but quickly lose the alignment.

Indeed, we point to a key difference between supervised learning and unsupervised learning. While in the former, deeper networks, which can learn even random labels, work well (Zhang et al., 2017), unsupervised learning requires a careful control of the network capacity. This realization, which echoes the application of MDL for model selection in unsupervised learning (Zemel, 1994), was overshadowed by the overgeneralized belief that deeper networks lead to higher accuracy.

The limitations of unsupervised based learning that are due to symmetry, are also a part of our model. For example, the mapping of cars in one pose to cars in the mirrored pose that sometimes happens in (Kim et al., 2017), is similar in nature to the mapping of $x$ to $1 - x$ in the simple example given in Sec. 3.1. Such symmetries occur when we can divide $y_{AB}$ into two functions $y_{AB} = y_2 \circ y_1$ such that a function $W$ is a linear mapping and also a DPM of $y_1 \circ D_A$ and, therefore, $D_B \approx y_2 \circ W \circ y_1$.

While we focus on unsupervised learning, the emergence of semantics when learning with a restricted capacity is widely applicable, such as with autoencoders, transfer learning, semi-supervised learning and elsewhere. As an extreme example, Sutskever et al. (2015) present empirical evidence that a meaningful mapper can be learned, even from very few examples, if the network trained is kept small.

## 7 CONCLUSION

The recent success in mapping between two domains in an unsupervised way and without any existing knowledge, other than network hyperparameters, is nothing less than extraordinary and has far reaching consequences. As far as we know, nothing in the existing machine learning or cognitive science literature suggests that this would be possible.

We provide an intuitive definition of function complexity and employ it in order to identify minimal complexity mappings, which we conjecture play a pivotal role in this success. If our hypothesis is correct, simply by training networks that are not too complex, the target mapping stands out from all other alternative mappings.

Our analysis leads directly to a new unsupervised cross domain mapping algorithm that is able to avoid the ambiguity of such mapping, yet enjoy the expressiveness of deep neural networks. The experiments demonstrate that the analogies become richer in details and more complex, while maintaining the alignment.

We show that the number of low-discrepancy mappings that are of low-complexity is expected to be small. Our main proof is based on the assumption of identifiability, which constitutes an open question. We hope that there would be a renewed interest in this problem, which has been open for

decades for networks with more than a single hidden layer and is unexplored for modern activation functions.

ACKNOWLEDGMENTS

This project has received funding from the European Research Council (ERC) under the European Union's Horizon 2020 research and innovation programme (grant ERC CoG 725974). The authors would like to thank Etai Littwin, Moustapha Cisse, Léon Bottou, Arthur Szlam, and Ofir Yakovian for insightful discussions. The contribution of Tomer Galanti is part of Ph.D. thesis research conducted at Tel Aviv University.

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

Table 1: Comparing the VGG descriptor similarity, separation accuracy and discrepancy for varying complexity $k$

|  |  | $k = 4$ | $k = 6$ | $k = 8$ | $k = 10$ | $k = 12$ | $k = 14$ |
|---|---|---|---|---|---|---|---|
| Male to Female | Discrepancy | 0.527 | 0.203 | 0.091 | 0.094 | 0.083 | 0.086 |
|  | Similarity | 0.301 | 0.269 | 0.103 | 0.106 | 0.089 | 0.100 |
|  | Separation | 0.938 | 0.932 | 0.940 | 0.940 | 0.940 | 0.938 |
| Female to Male | Discrepancy | 0.882 | 0.122 | 0.150 | 0.075 | 0.076 | 0.091 |
|  | Similarity | 0.303 | 0.260 | 0.110 | 0.105 | 0.093 | 0.100 |
|  | Separation | 0.798 | 0.865 | 0.860 | 0.87 | 0.857 | 0.866 |
| Blond to Black Hair | Discrepancy | 0.467 | 0.214 | 0.092 | 0.097 | 0.094 | 0.081 |
|  | Similarity | 0.365 | 0.287 | 0.240 | 0.106 | 0.091 | 0.0870 |
|  | Separation | 0.903 | 0.925 | 0.922 | 0.917 | 0.922 | 0.923 |
| Black to Blond Hair | Discrepancy | 0.663 | 0.264 | 0.073 | 0.094 | 0.084 | 0.076 |
|  | Similarity | 0.337 | 0.270 | 0.240 | 0.106 | 0.087 | 0.085 |
|  | Separation | 0.941 | 0.941 | 0.911 | 0.916 | 0.915 | 0.917 |
| Eyeglasses to Non-Eyeglasses | Discrepancy | 0.323 | 0.159 | 0.071 | 0.082 | 0.083 | 0.081 |
|  | Similarity | 0.470 | 0.391 | 0.347 | 0.114 | 0.125 | 0.146 |
|  | Separation | 0.786 | 0.785 | 0.828 | 0.843 | 0.849 | 0.828 |
| Non Eyeglasses to Eyeglasses | Discrepancy | 0.577 | 0.518 | 0.236 | 0.263 | 0.093 | 0.085 |
|  | Similarity | 0.452 | 0.373 | 0.364 | 0.105 | 0.108 | 0.127 |
|  | Septation | 0.748 | 0.749 | 0.766 | 0.848 | 0.832 | 0.840 |

Table 2: (a) Norms of the various mappings $h$ for mapping Males to Females using the DiscoGAN architecture. (b) Norms of 18-layer networks that approximates the mappings obtained with a varying number of layers.

|  | Norm | Number of layers | | | | |
|---|---|---|---|---|---|---|
|  |  | 4 | 6 | 8 | 10 | 12 |
| A to B | L1 norm | 6382 | 23530 | 36920 | 44670 | 71930 |
|  | Average L1 norm per layer | 1064 | 2353 | 2637 | 2482 | 3270 |
|  | L2 norm | 18.25 | 29.24 | 28.44 | 31.72 | 36.57 |
|  | Average L2 norm per layer | 7.084 | 8.353 | 7.154 | 6.708 | 7.009 |
| B to A | L1 norm | 6311 | 21240 | 31090 | 37380 | 64500 |
|  | Average L1 norm per layer | 1052 | 2124 | 2221 | 2077 | 2932 |
|  | L2 norm | 18.36 | 26.79 | 25.85 | 28.36 | 34.99 |
|  | Average L2 norm per layer | 7.161 | 7.757 | 6.552 | 6.058 | 6.771 |

(a)

|  | Norm | Number of layers | | | | |
|---|---|---|---|---|---|---|
|  |  | 4 | 6 | 8 | 10 | 12 |
| A to B | L1 norm | 317200 | 228700 | 356500 | 247200 | 164200 |
|  | Average L1 norm per layer | 9329 | 6726 | 10485 | 7271 | 4829 |
|  | L2 norm | 528.1 | 401.7 | 559.6 | 410.1 | 346.8 |
|  | Average L2 norm per layer | 3.031 | 2.284 | 3.242 | 2.257 | 1.890 |
| B to A | L1 norm | 316900 | 194500 | 353900 | 171500 | 228900 |
|  | Average L1 norm per layer | 9323 | 5719 | 10410 | 5045 | 6733 |
|  | L2 norm | 523.2 | 375.9 | 555.7 | 346.5 | 373.3 |
|  | Average L2 norm per layer | 3.003 | 2.029 | 3.210 | 1.921 | 2.289 |

(b)

Table 3: Seeking DPMs: the distance from the identity and the discrepancy (GAN loss) for various numbers of layers, where training a DiscoGAN from a dataset to itself.

| Dataset | loss | Number of layers: | | | | | |
|---|---|---|---|---|---|---|---|
| | | 4 | 6 | 8 | 10 | 12 | 14 |
| Males | $\sum_{x \in A} \|x - h(x)\|$ | **0.09** | 0.42 | 0.45 | 0.45 | 0.45 | 0.45 |
| | Discrepancy | 0.37 | 0.60 | 0.27 | 0.20 | 0.17 | **0.10** |
| Females | $\sum_{x \in A} \|x - h(x)\|$ | **0.06** | 0.36 | 0.43 | 0.42 | 0.44 | 0.45 |
| | Discrepancy | 0.32 | 0.40 | 0.15 | **0.11** | **0.11** | **0.11** |
| Handbags | $\sum_{x \in A} \|x - h(x)\|$ | **0.10** | 0.28 | 0.37 | 0.37 | 0.38 | 0.37 |
| | Discrepancy | **0.13** | 0.28 | 0.24 | 0.14 | 0.15 | 0.20 |
| Shoes | $\sum_{x \in A} \|x - h(x)\|$ | **0.06** | 0.15 | 0.29 | 0.30 | 0.30 | 0.30 |
| | Discrepancy | 0.15 | 0.28 | 0.20 | 0.15 | **0.10** | **0.10** |
| Edges of handbags | $\sum_{x \in A} \|x - h(x)\|$ | **0.28** | 0.55 | 0.51 | 0.52 | 0.50 | 0.49 |
| | Discrepancy | **0.18** | 0.28 | 0.58 | 0.47 | 0.40 | 0.35 |
| Edges of shoes | $\sum_{x \in A} \|x - h(x)\|$ | **0.23** | 0.50 | 0.59 | 0.55 | 0.49 | 0.43 |
| | Discrepancy | **0.17** | 0.21 | 0.65 | 0.46 | 0.45 | 0.45 |

(a)
Input

(b)
Output

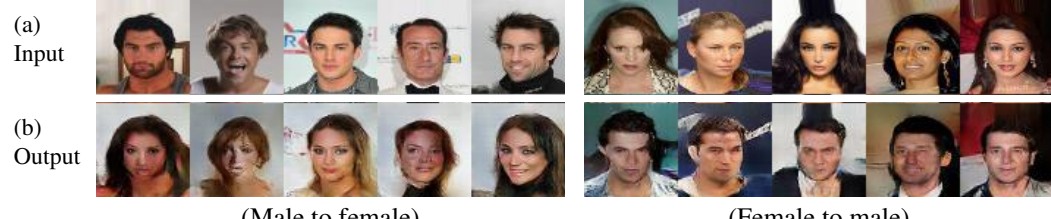

(Male to female)                    (Female to male)

Figure 3: Results for celebA Male to Female transfer (a) Input (b) The mapping obtained by the GAN loss without additional losses.

(a)
Input

(b)
Output

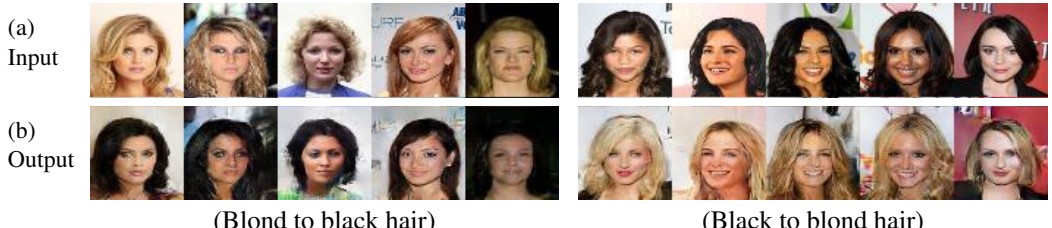

(Blond to black hair)                    (Black to blond hair)

Figure 4: Same as Fig. 3 for black to blond hair conversion.

(a)
Input

(b)
Output

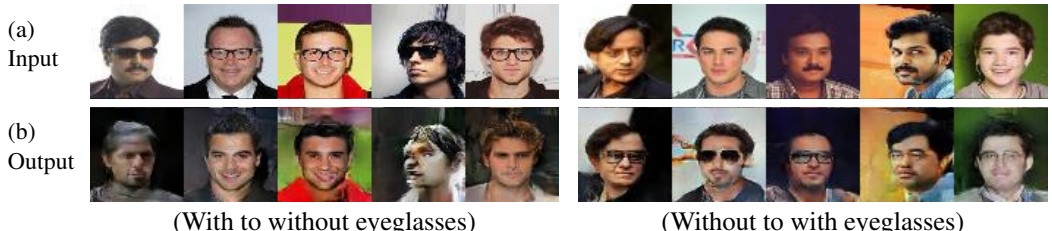

(With to without eyeglasses)                    (Without to with eyeglasses)

Figure 5: Same as Fig. 3 for eyeglasses to no eyeglasses conversion.

(a)
Input

(b)
Output

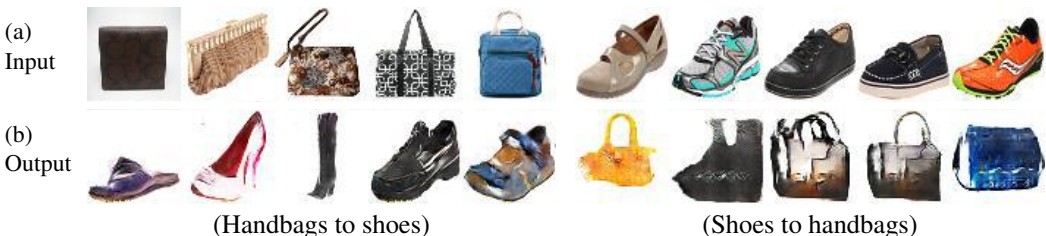

(Handbags to shoes)                    (Shoes to handbags)

Figure 6: Same as Fig. 3 for handbag to shoes and shoes to handbag mapping.

(a)
Input

(b)
Output

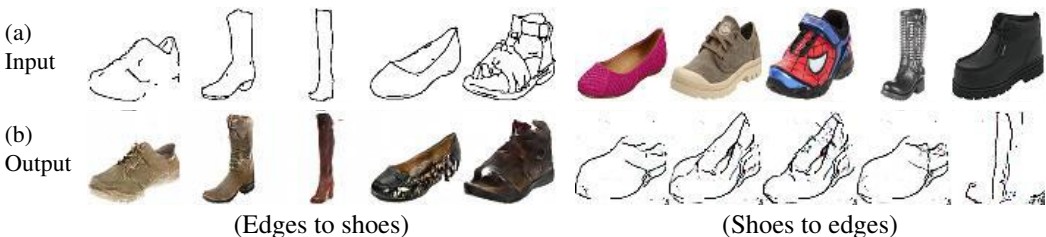

(Edges to shoes)                    (Shoes to edges)

Figure 7: Same as Fig. 3 for edges to shoes and shoes to edges conversion.

Table 4: Results for Alg. 1 for different datasets. VGG Similarity is given in the first column. The second column gives the separation value using the linear classifier. In the third column, we measure the discrepancy of the mapping. The last column provides the distance of $h$ to $g$, where applicable.

| Dataset | $f$ | Complexity | Descriptor Similarity | Separation Accuracy | Discrepancy $\mathrm{disc}(f \circ D_A, D_B)$ | Distance $R_{D_A}[h, g]$ |
|---|---|---|---|---|---|---|
| Male to Female | $g$ | $k_1 = 6$ | 0.267 | 0.928 | 0.230 | - |
| | $g$ | $k = 8$ | 0.280 | 0.938 | **0.077** | - |
| | $g$ | $k = 10$ | 0.106 | 0.940 | 0.094 | - |
| | $g$ | $k = 12$ | 0.089 | 0.940 | 0.083 | - |
| | $h$ | $k_2 = 8$ | **0.316** | 0.933 | 0.087 | **0.054** |
| | $h$ | $k_2 = 10$ | 0.204 | 0.937 | 0.109 | 0.075 |
| | $h$ | $k_2 = 12$ | 0.197 | **0.941** | 0.127 | 0.077 |
| Female to Male | $g$ | $k_1 = 6$ | 0.268 | 0.848 | 0.310 | - |
| | $g$ | $k = 8$ | 0.260 | 0.848 | 0.107 | - |
| | $g$ | $k = 10$ | 0.105 | 0.870 | 0.075 | - |
| | $g$ | $k = 12$ | 0.093 | 0.857 | 0.076 | - |
| | $h$ | $k_2 = 8$ | **0.304** | 0.878 | 0.107 | **0.056** |
| | $h$ | $k_2 = 10$ | 0.215 | **0.884** | 0.082 | 0.083 |
| | $h$ | $k_2 = 12$ | 0.214 | 0.883 | **0.073** | 0.082 |
| Blond to Black Hair | $g$ | $k_1 = 6$ | 0.287 | 0.925 | 0.214 | - |
| | $g$ | $k = 8$ | 0.24 | 0.922 | **0.092** | - |
| | $g$ | $k = 10$ | 0.106 | 0.917 | 0.097 | - |
| | $g$ | $k = 12$ | 0.091 | 0.922 | 0.094 | - |
| | $h$ | $k_2 = 8$ | **0.293** | 0.926 | 0.136 | **0.152** |
| | $h$ | $k_2 = 10$ | 0.197 | 0.926 | 0.225 | 0.161 |
| | $h$ | $k_2 = 12$ | 0.199 | **0.928** | 0.092 | 0.161 |
| Black to Blond Hair | $g$ | $k_1 = 6$ | 0.270 | 0.941 | 0.264 | - |
| | $g$ | $k = 8$ | 0.24 | 0.911 | **0.073** | - |
| | $g$ | $k = 10$ | 0.106 | 0.916 | 0.094 | - |
| | $g$ | $k = 12$ | 0.087 | 0.915 | 0.084 | - |
| | $h$ | $k_2 = 8$ | **0.287** | 0.938 | 0.077 | **0.146** |
| | $h$ | $k_2 = 10$ | 0.179 | 0.946 | 0.165 | 0.149 |
| | $h$ | $k_2 = 12$ | 0.180 | **0.952** | 0.168 | 0.152 |
| Eyeglasses to Non-Eyeglasses | $g$ | $k_1 = 6$ | **0.391** | 0.785 | 0.159 | - |
| | $g$ | $k = 8$ | 0.347 | 0.828 | **0.071** | - |
| | $g$ | $k = 10$ | 0.114 | 0.843 | 0.082 | - |
| | $g$ | $k = 12$ | 0.125 | 0.849 | 0.083 | - |
| | $h$ | $k_2 = 8$ | **0.391** | 0.786 | 0.097 | **0.058** |
| | $h$ | $k_2 = 10$ | 0.283 | 0.847 | 0.180 | 0.083 |
| | $h$ | $k_2 = 12$ | 0.274 | **0.860** | 0.148 | 0.081 |
| Non-Eyeglasses to Eyeglasses | $g$ | $k_1 = 6$ | 0.373 | 0.749 | 0.518 | - |
| | $g$ | $k = 8$ | 0.364 | 0.766 | 0.236 | - |
| | $g$ | $k = 10$ | 0.105 | **0.848** | 0.263 | - |
| | $g$ | $k = 12$ | 0.108 | 0.832 | **0.093** | - |
| | $h$ | $k_2 = 8$ | **0.389** | 0.780 | 0.300 | **0.063** |
| | $h$ | $k_2 = 10$ | 0.272 | 0.807 | 0.370 | 0.083 |
| | $h$ | $k_2 = 12$ | 0.282 | 0.803 | 0.409 | 0.081 |

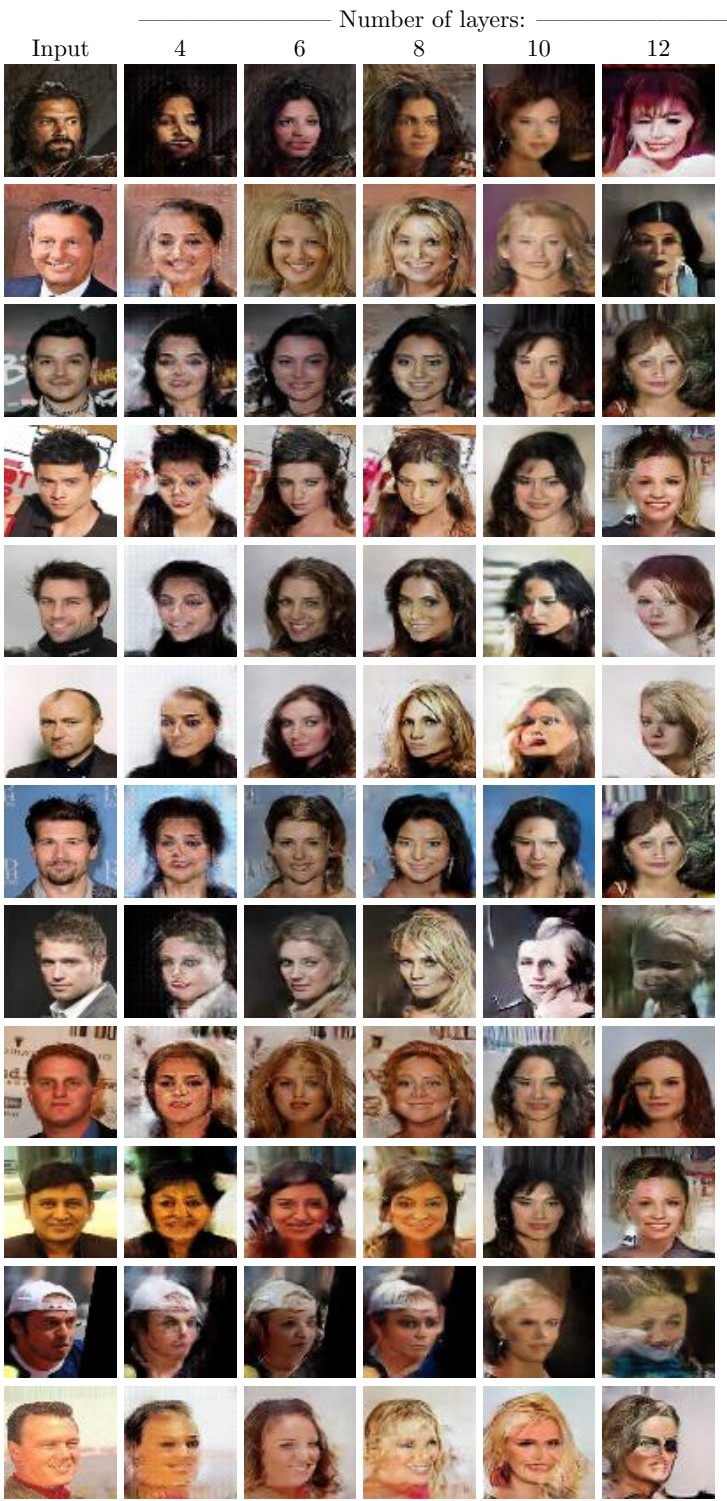

Figure 8: Results for celebA Male to Female transfer for networks with different number of layers.

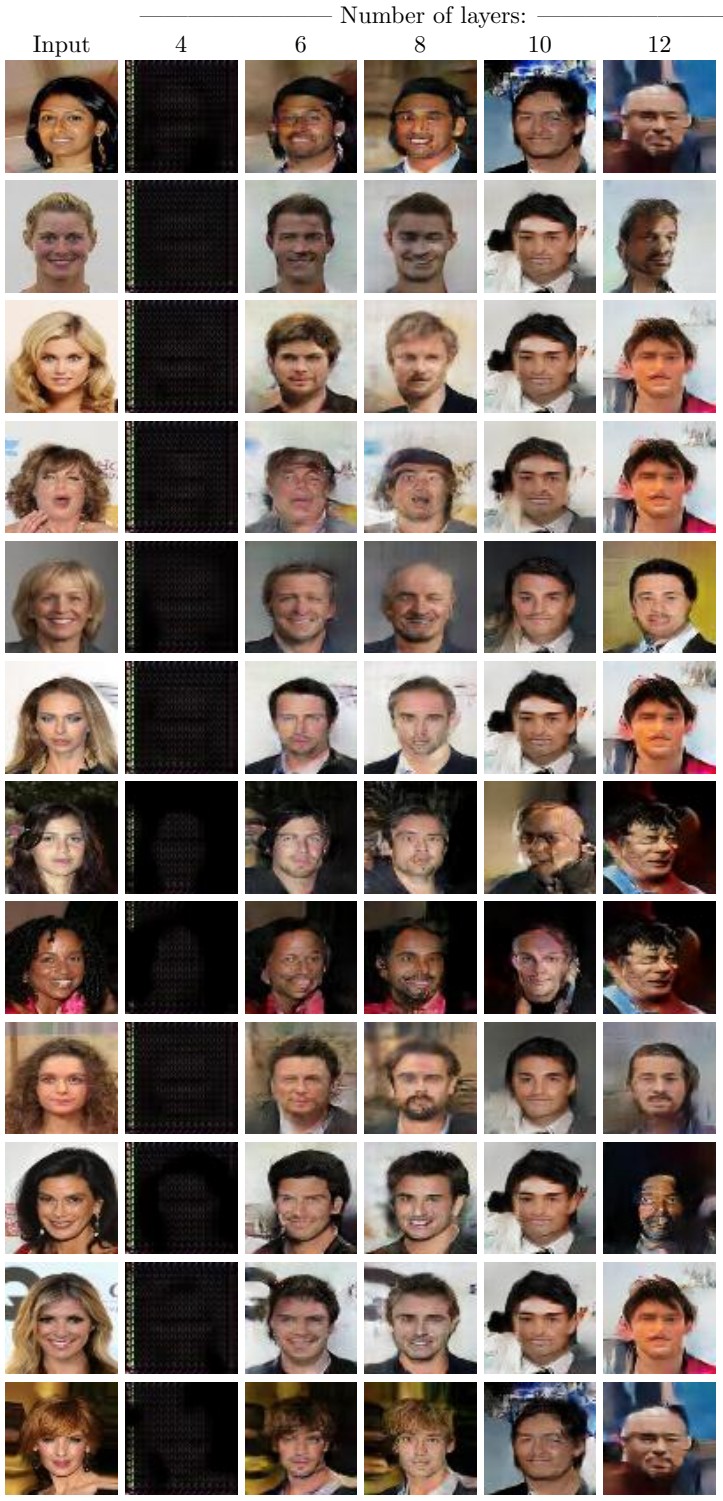

Figure 9: Results for celebA Female to Male transfer for networks with different number of layers. The case of 4 layers failed to produce acceptable results.

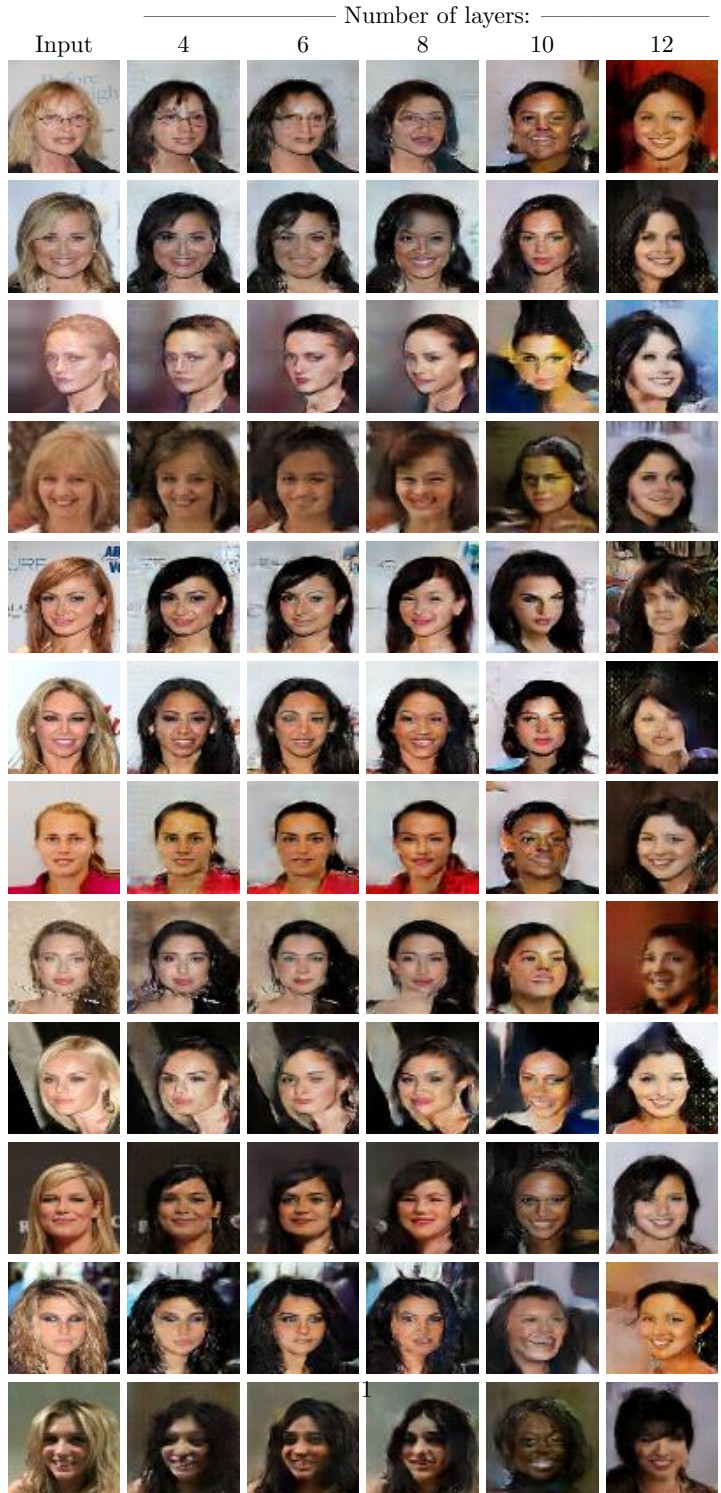

Figure 10: Results for celebA Blond to Black Hair transfer for networks with different number of layers.

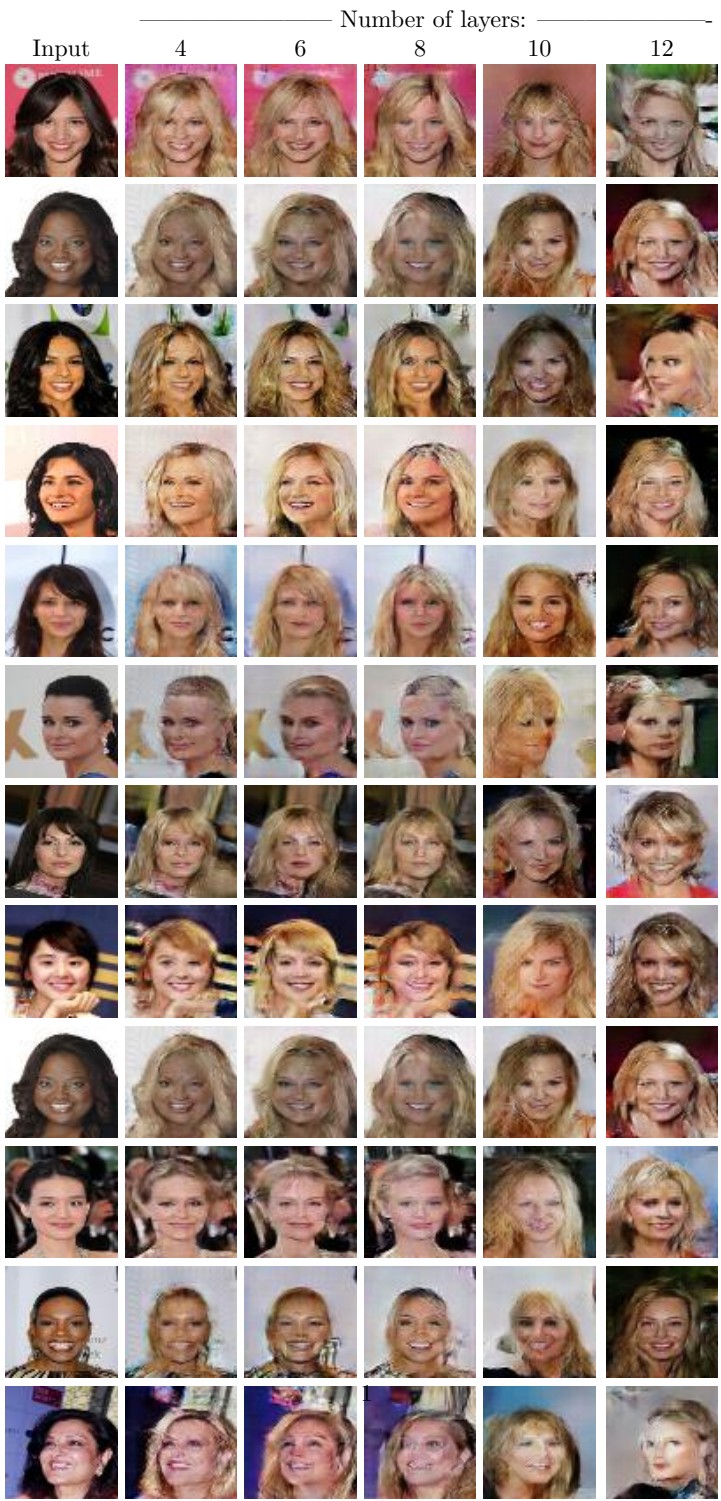

Figure 11: Results for celebA Black Hair to Blond transfer for networks with different number of layers.

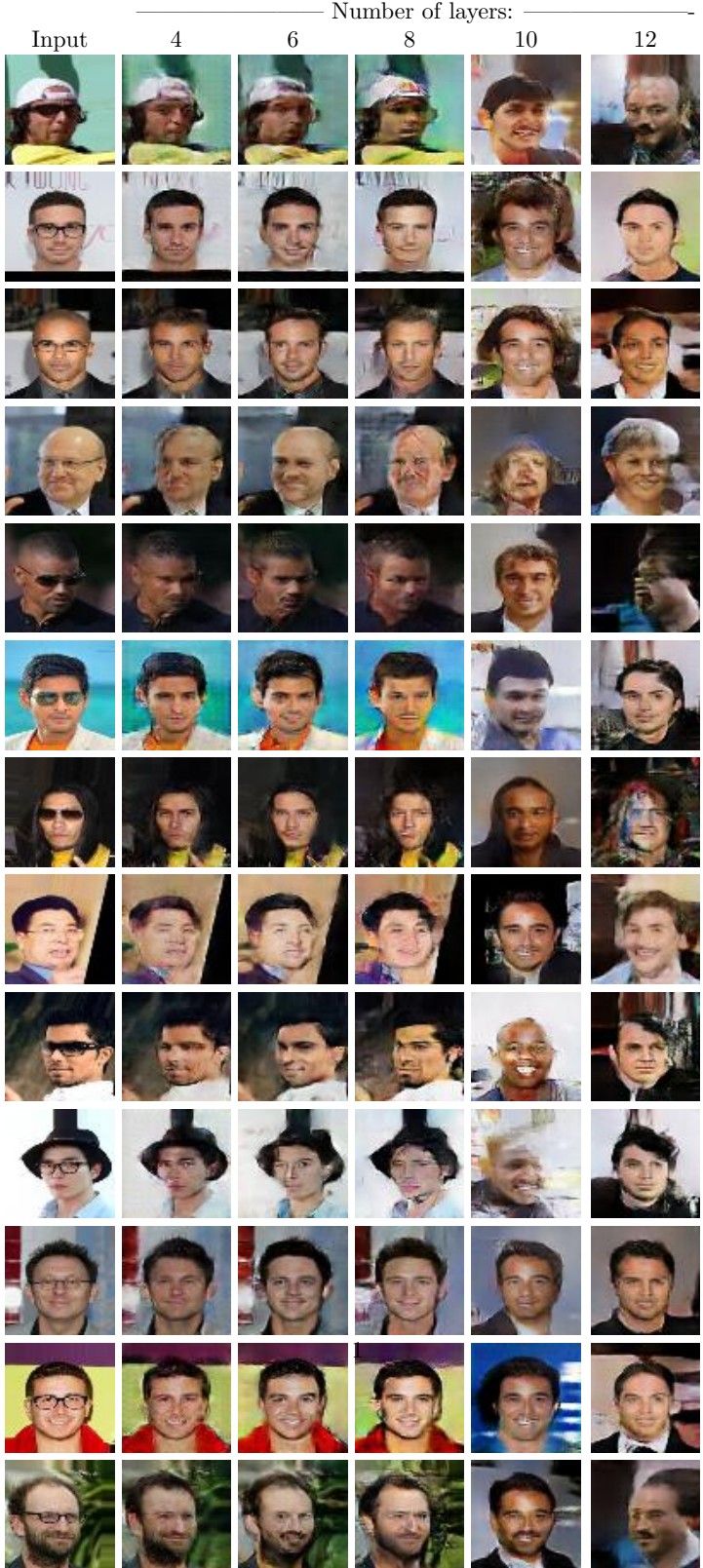

Figure 12: Results for celebA Eyeglasses to Non-Eyeglasses transfer for networks with different number of layers.

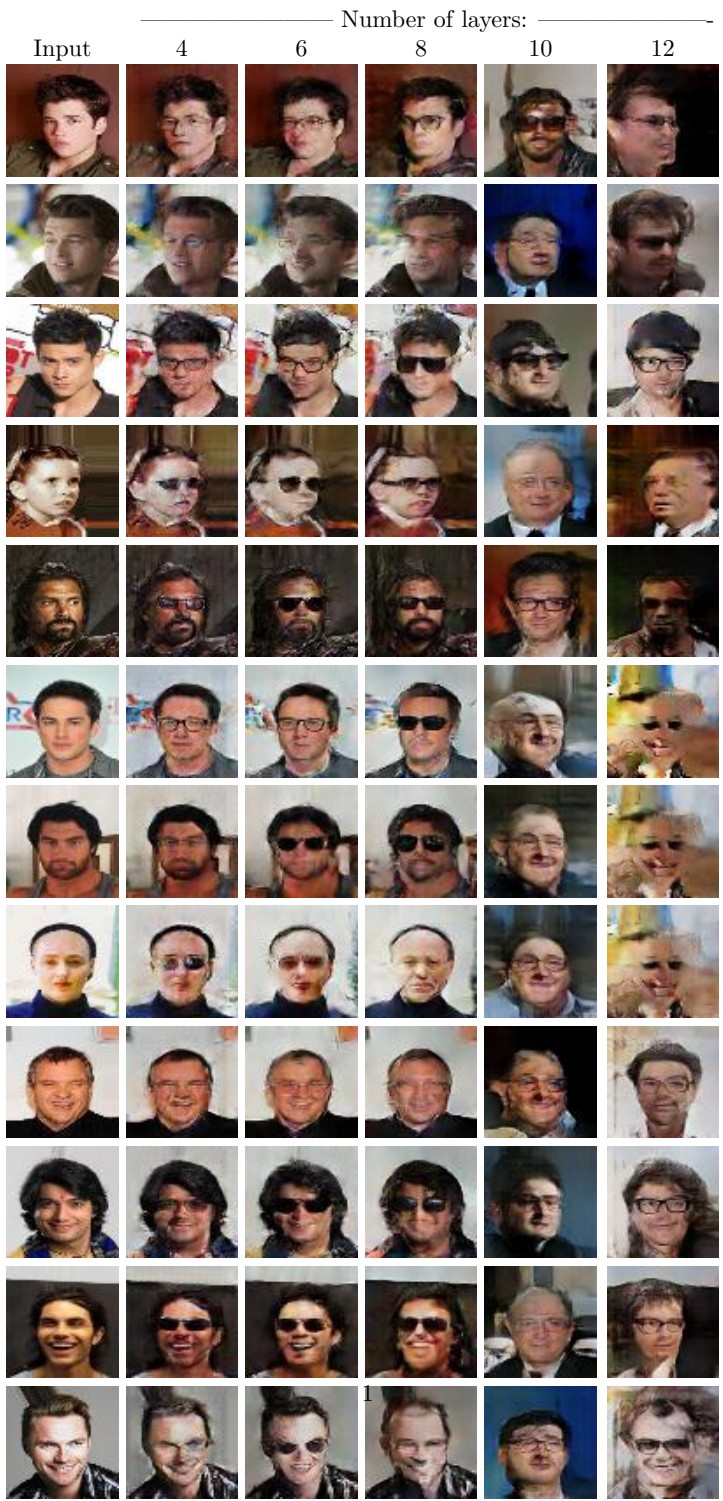

Figure 13: Results for celebA Non-Eyeglasses to Eyeglasses transfer for networks with different number of layers.

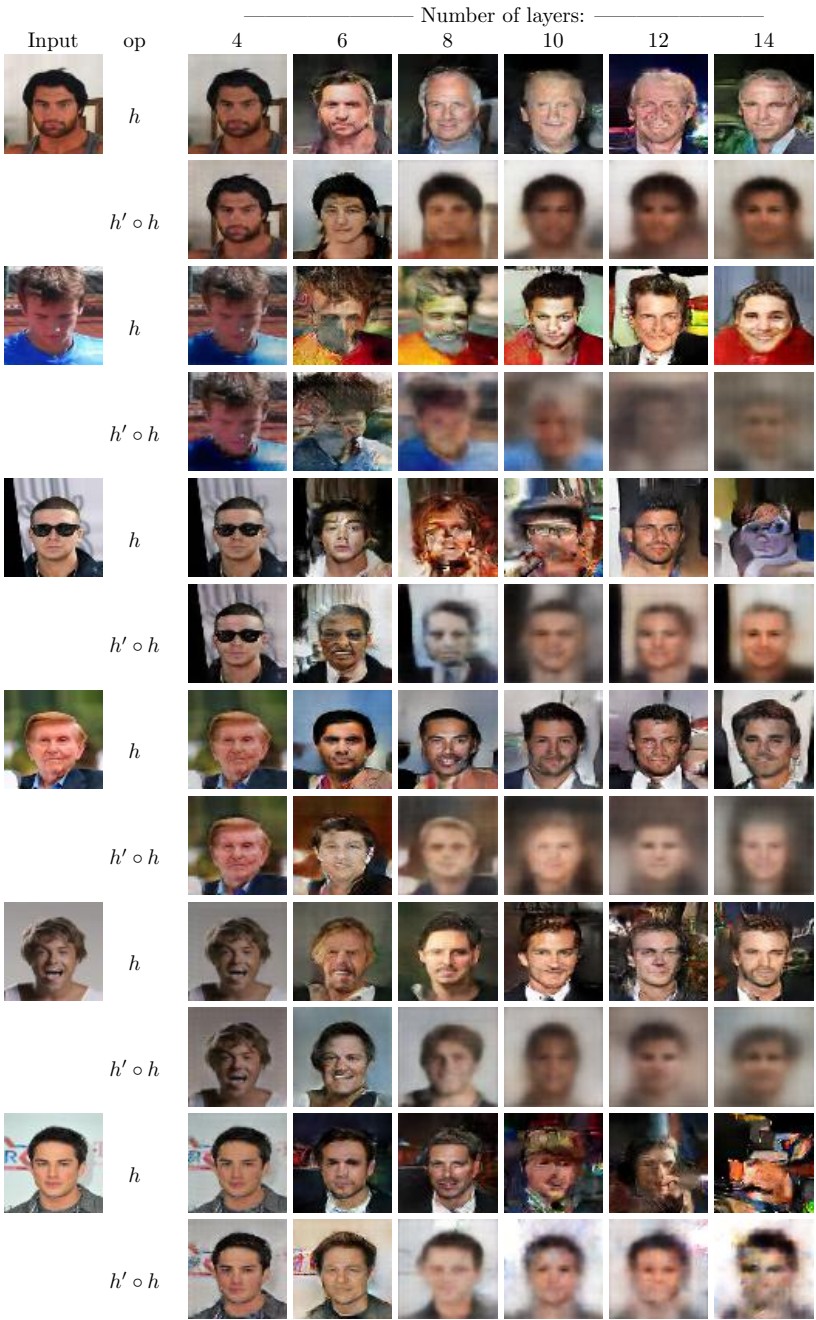

Figure 14: Results for mapping Males to itself (B=A) using a DiscoGAN architecture and enforcing that the mapping is not the identity mapping. The odd rows present the learned mapping $h$, and the even rows present the full cycle $h' \circ h$.

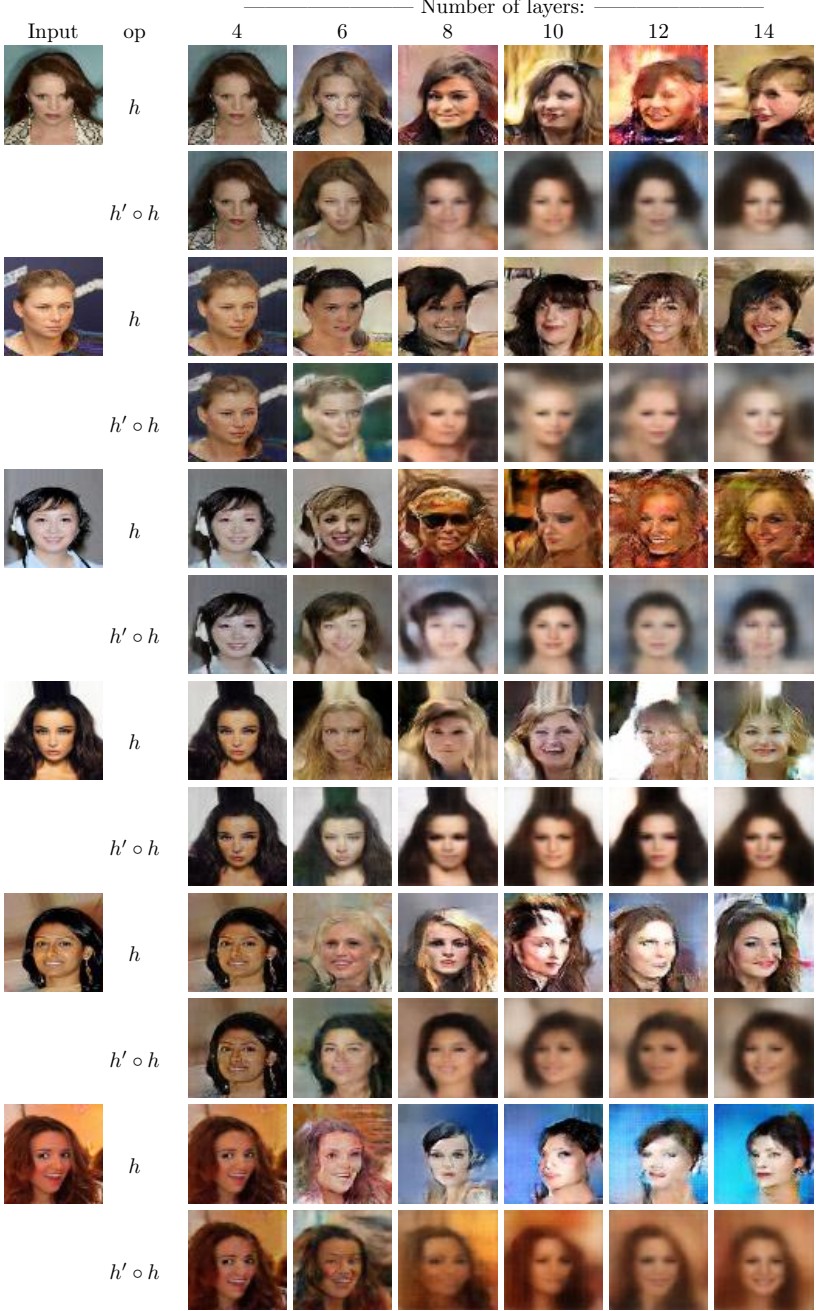

Figure 15: Results for mapping the Females to itself (B=A) using a DiscoGAN architecture and enforcing that the mapping is not the identity mapping. The odd rows present the learned mapping $h$, and the even rows present the full cycle $h' \circ h$.

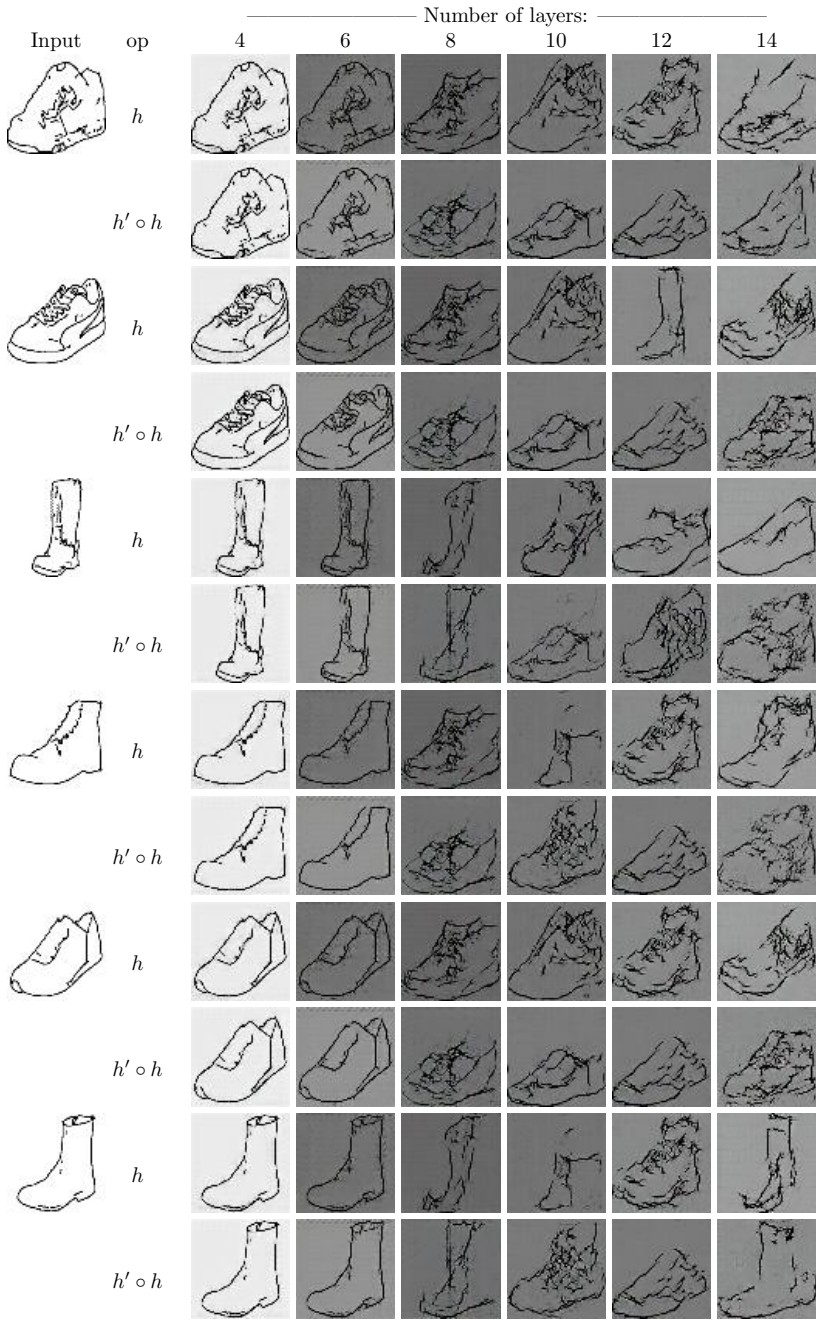

Figure 16: Results for mapping shoe edges to itself (B=A) using a DiscoGAN architecture and enforcing that the mapping is not the identity mapping. The odd rows present the learned mapping $h$, and the even rows present the full cycle $h' \circ h$.

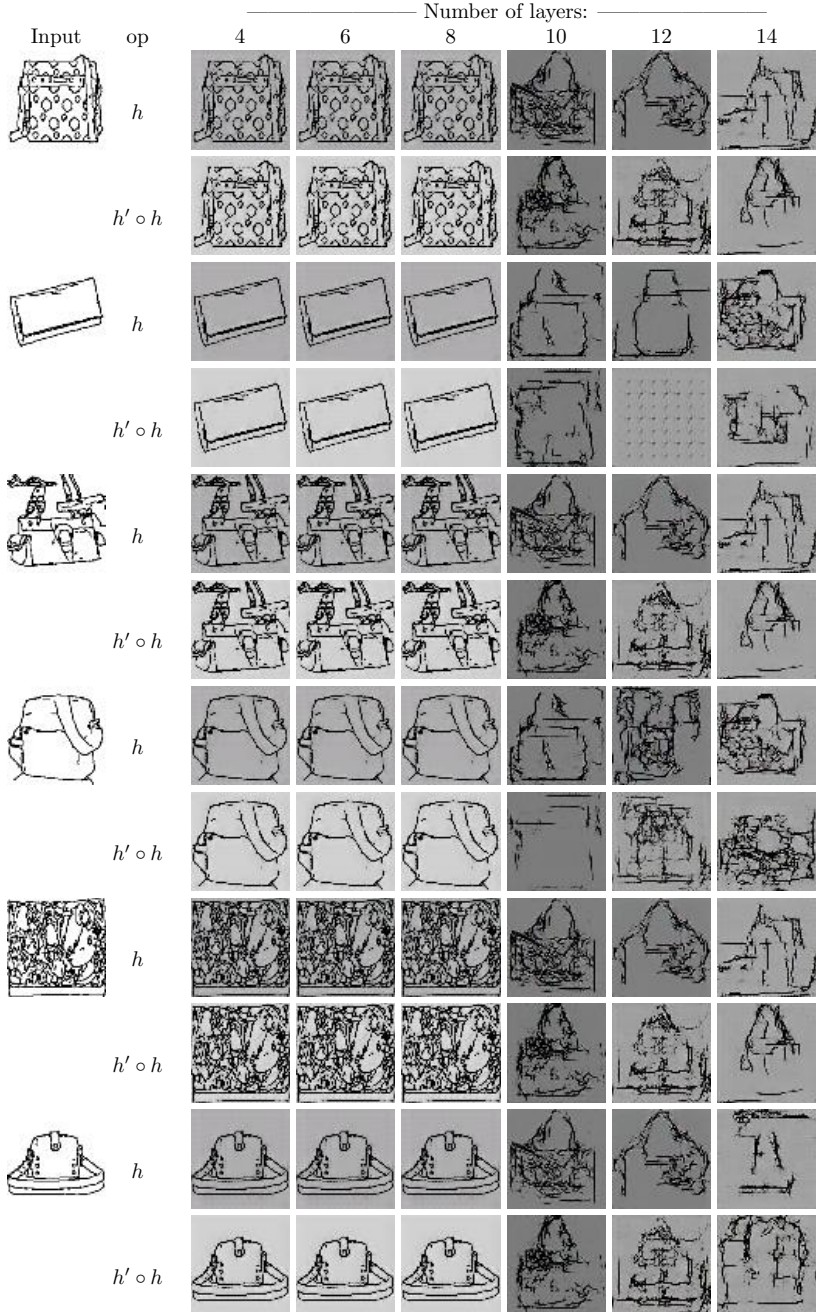

Figure 17: Results for mapping handbag edges to itself (B=A), using a DiscoGAN architecture and enforcing that the mapping is not the identity mapping. The odd rows present the learned mapping $h$, and the even rows present the full cycle $h' \circ h$.

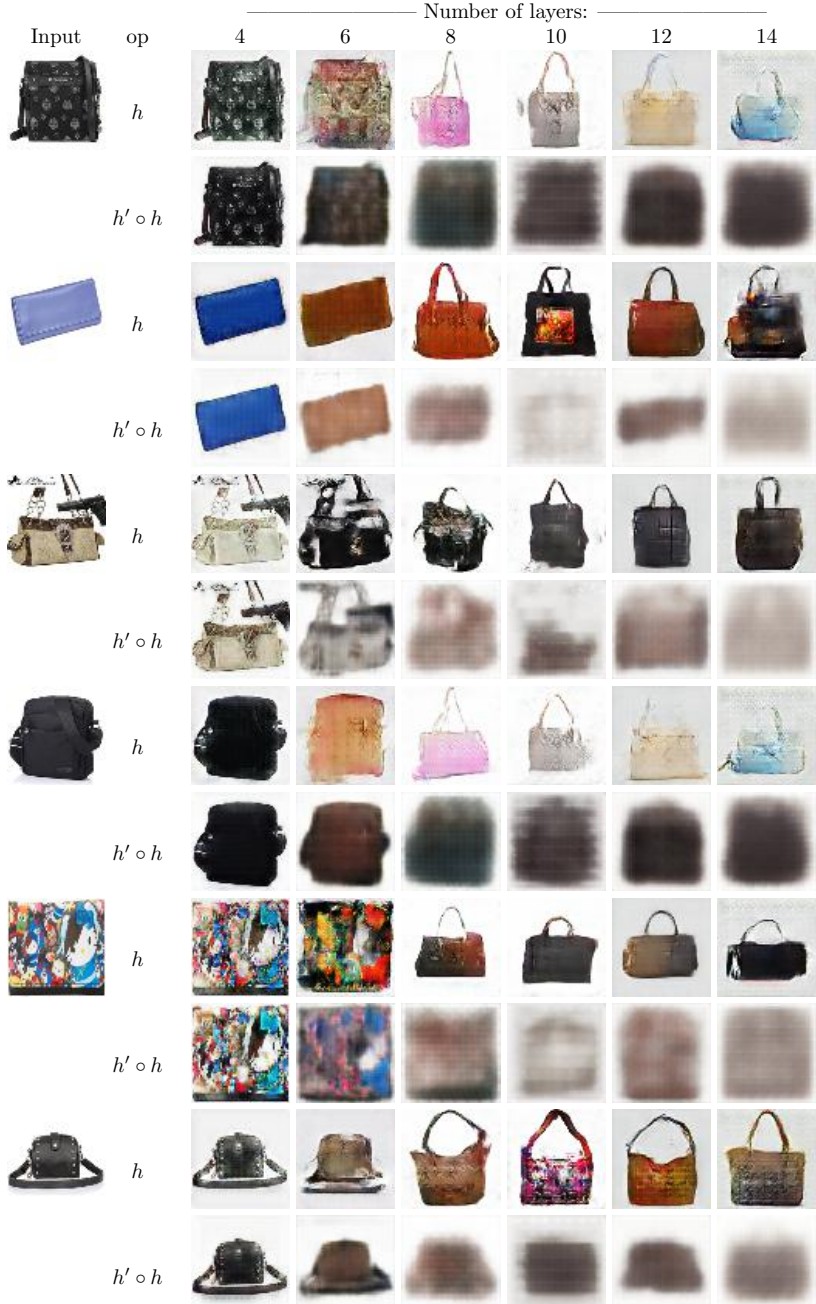

Figure 18: Results for mapping handbags to itself (B=A), using a DiscoGAN architecture and enforcing that the mapping is not the identity mapping. The odd rows present the learned mapping $h$, and the even rows present the full cycle $h' \circ h$.

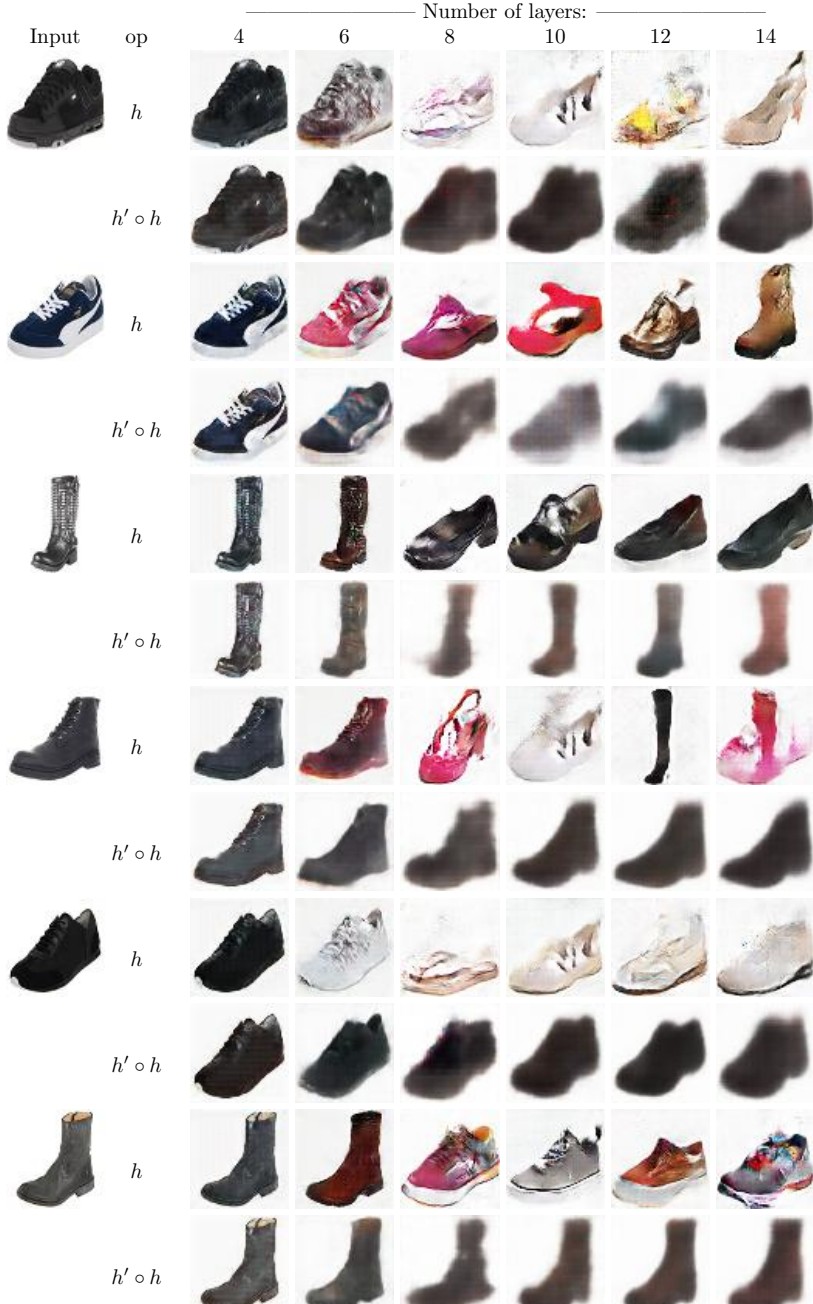

Figure 19: Results for mapping shoes to itself (B=A) using a DiscoGAN architecture and enforcing that the mapping is not the identity mapping. The odd rows present the learned mapping $h$, and the even rows present the full cycle $h' \circ h$.

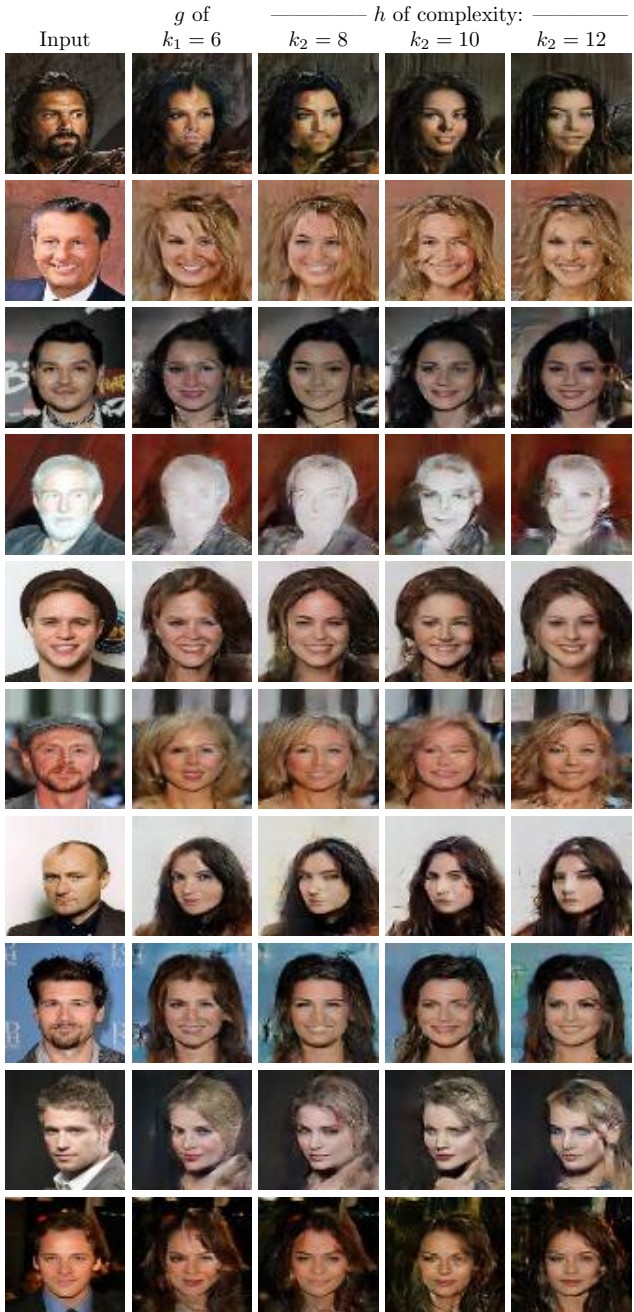

Figure 20: Results for Alg. 1 on Male2Female dataset for mapping male to female. Shown is a minimal complexity mapping $g$ that has low discrepancy, and various mappings $h$ obtained by the method.

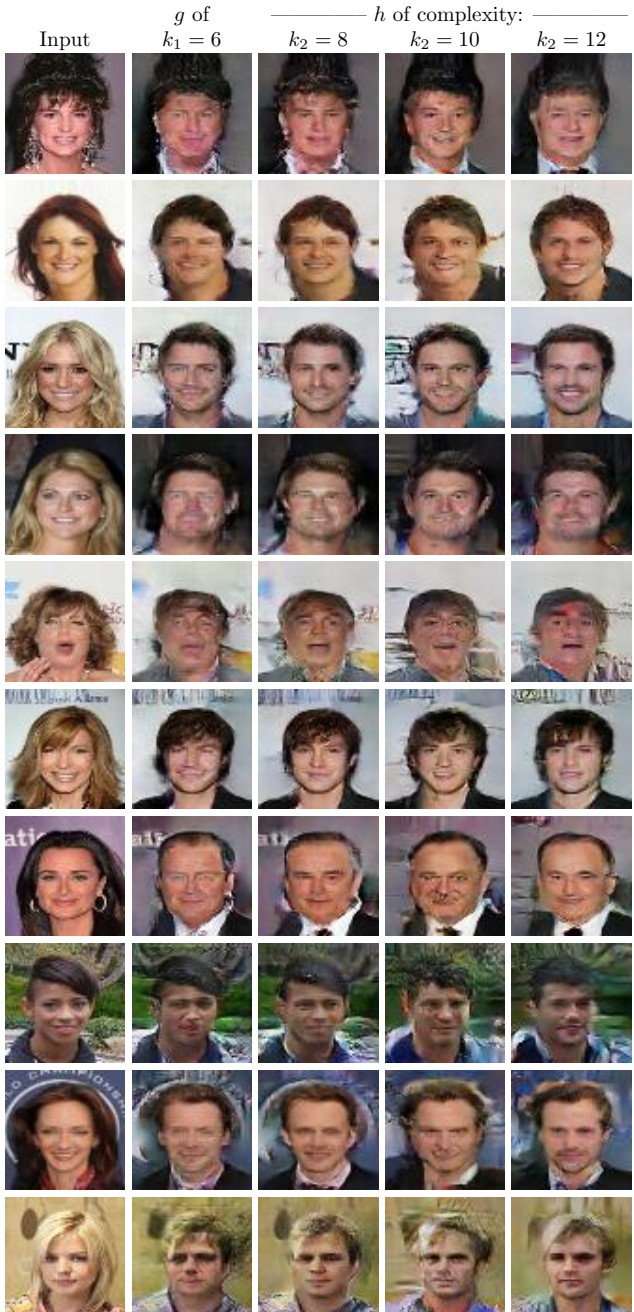

Figure 21: Results for Alg. 1 on Male2Female dataset for mapping female to male. Shown is a minimal complexity mapping $g$ that has low discrepancy, and various mappings $h$ obtained by the method.

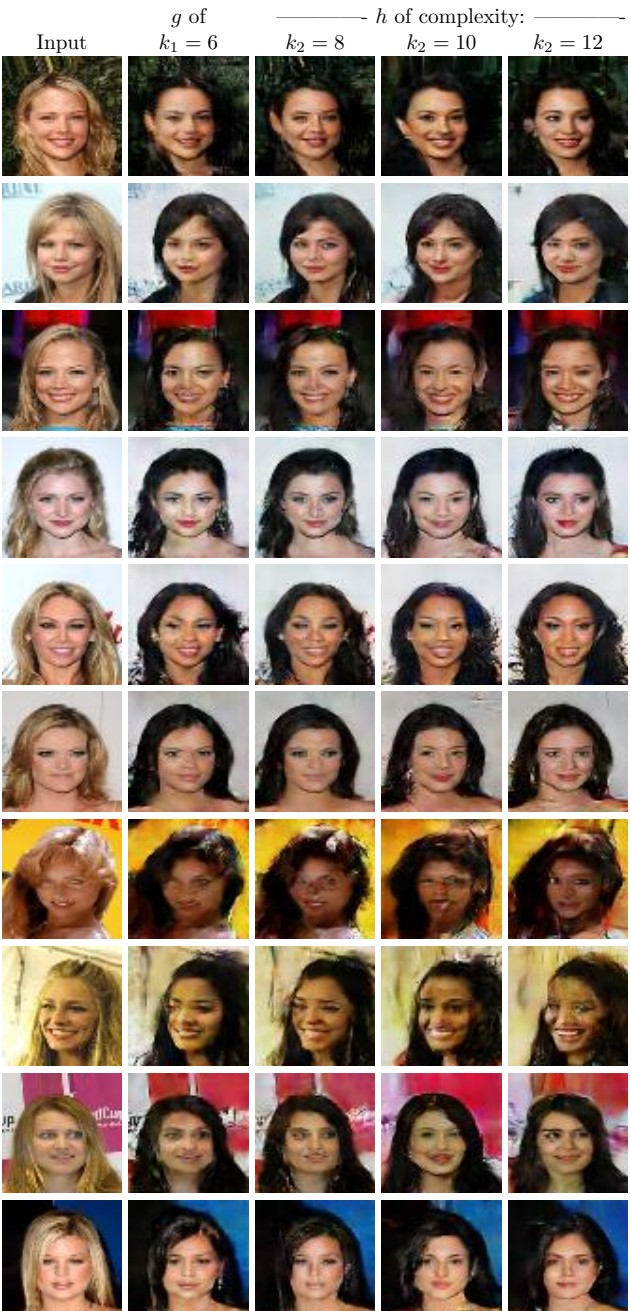

Figure 22: Results for Alg. 1 on celebA dataset for mapping blond to black. Shown is a minimal complexity mapping $g$ that has low discrepancy, and various mappings $h$ obtained by the method.

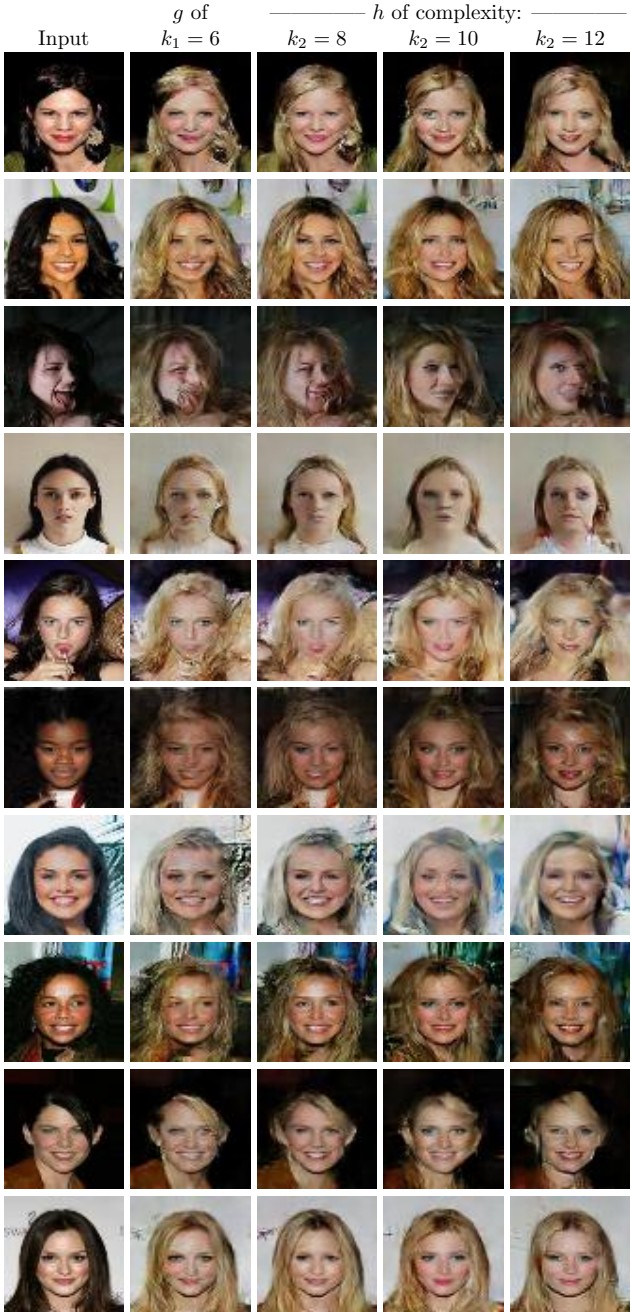

Figure 23: Results for Alg. 1 on celebA dataset for mapping black to blond. Shown is a minimal complexity mapping $g$ that has low discrepancy, and various mappings $h$ obtained by the method.

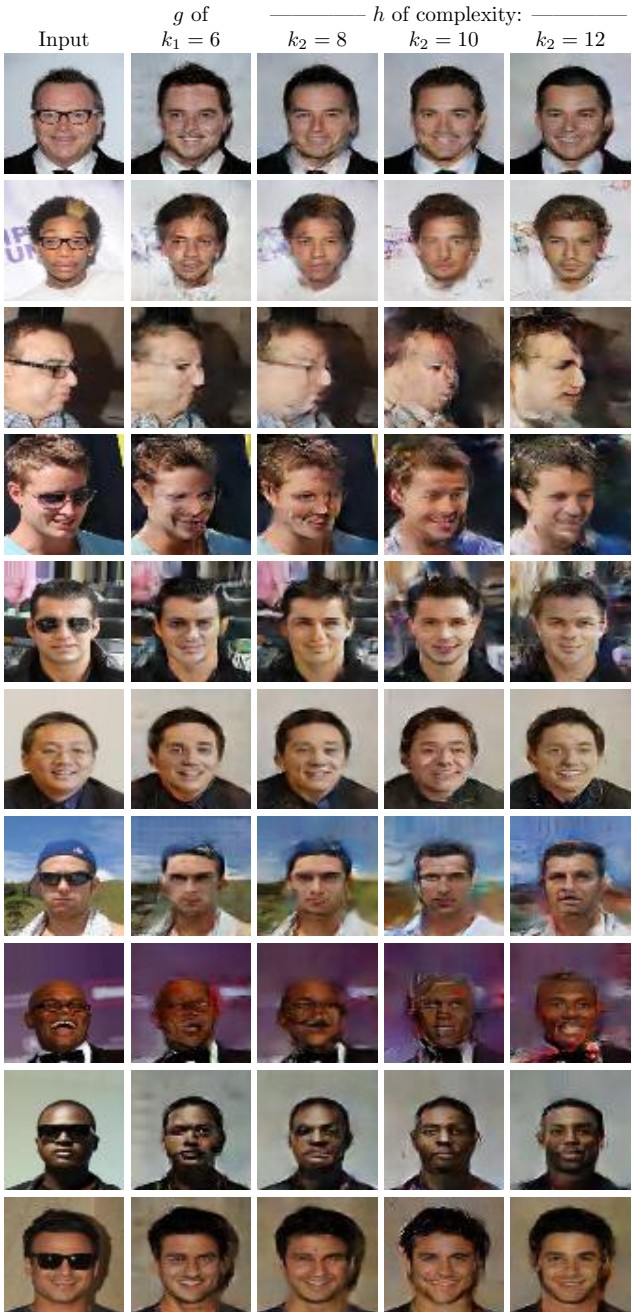

Figure 24: Results for Alg. 1 on Eyeglasses dataset for mapping eyeglasses to no eyeglasses. Shown is a minimal complexity mapping $g$ that has low discrepancy, and various mappings $h$ obtained by the method.

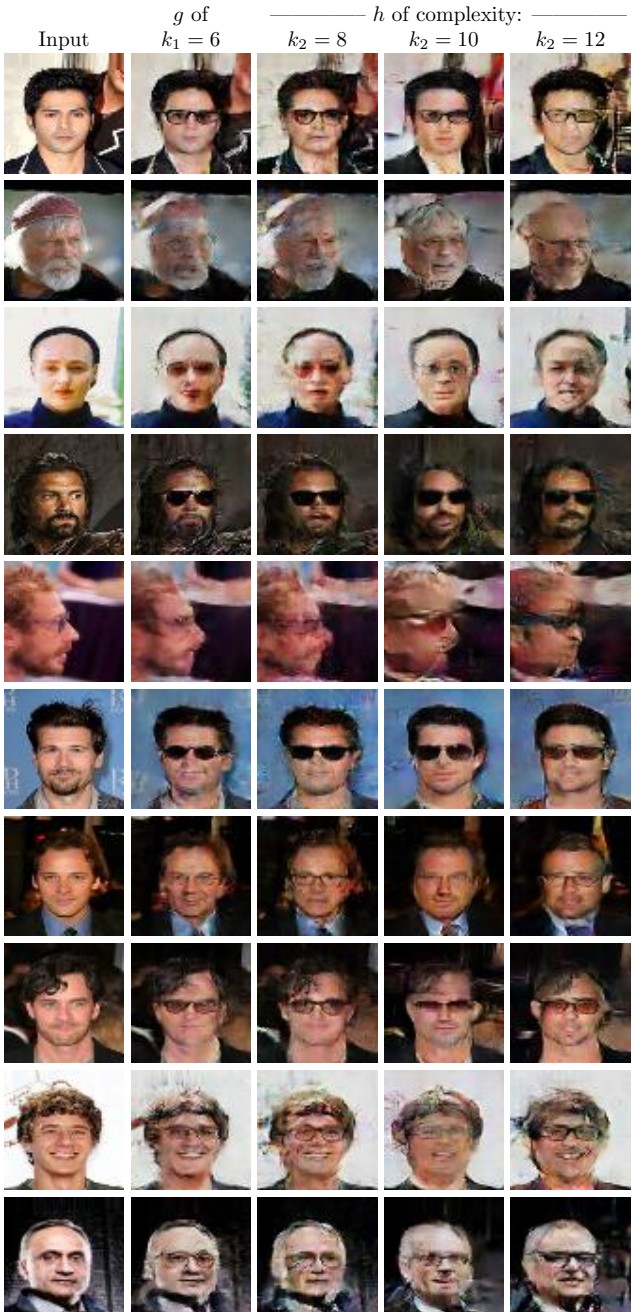

Figure 25: Results for Alg. 1 on Eyeglasses dataset for mapping no eyeglasses to eyeglasses. Shown is a minimal complexity mapping $g$ that has low discrepancy, and various mappings $h$ obtained by the method.

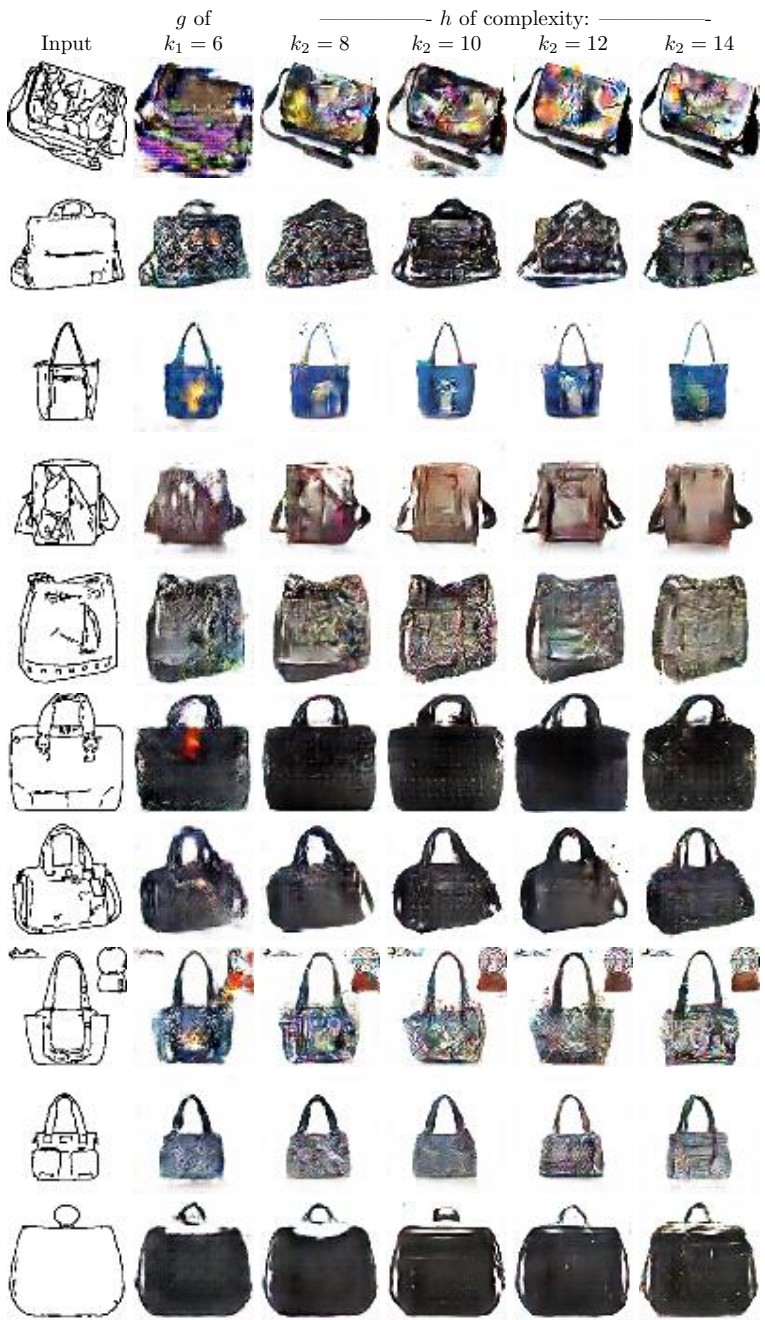

Figure 26: Results for Alg. 1 on Edges2Handbags dataset for mapping edges to handbags. Shown is a minimal complexity mapping $g$ that has low discrepancy, and various mappings $h$ obtained by the method.

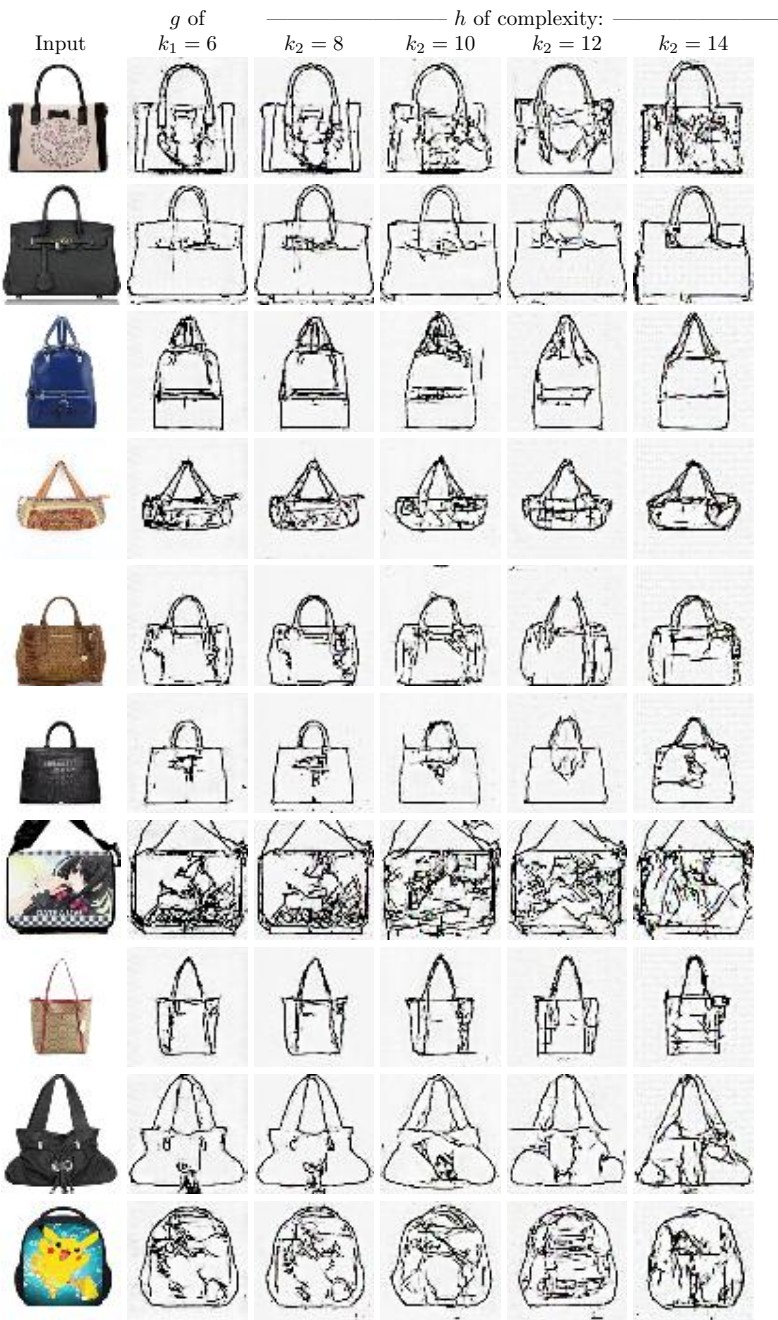

Figure 27: Results for Alg. 1 on Edges2Handbags dataset for mapping handbags to edges. Shown are a minimal complexity mapping $g$ that has low discrepancy, and various mappings $h$ obtained by the method.

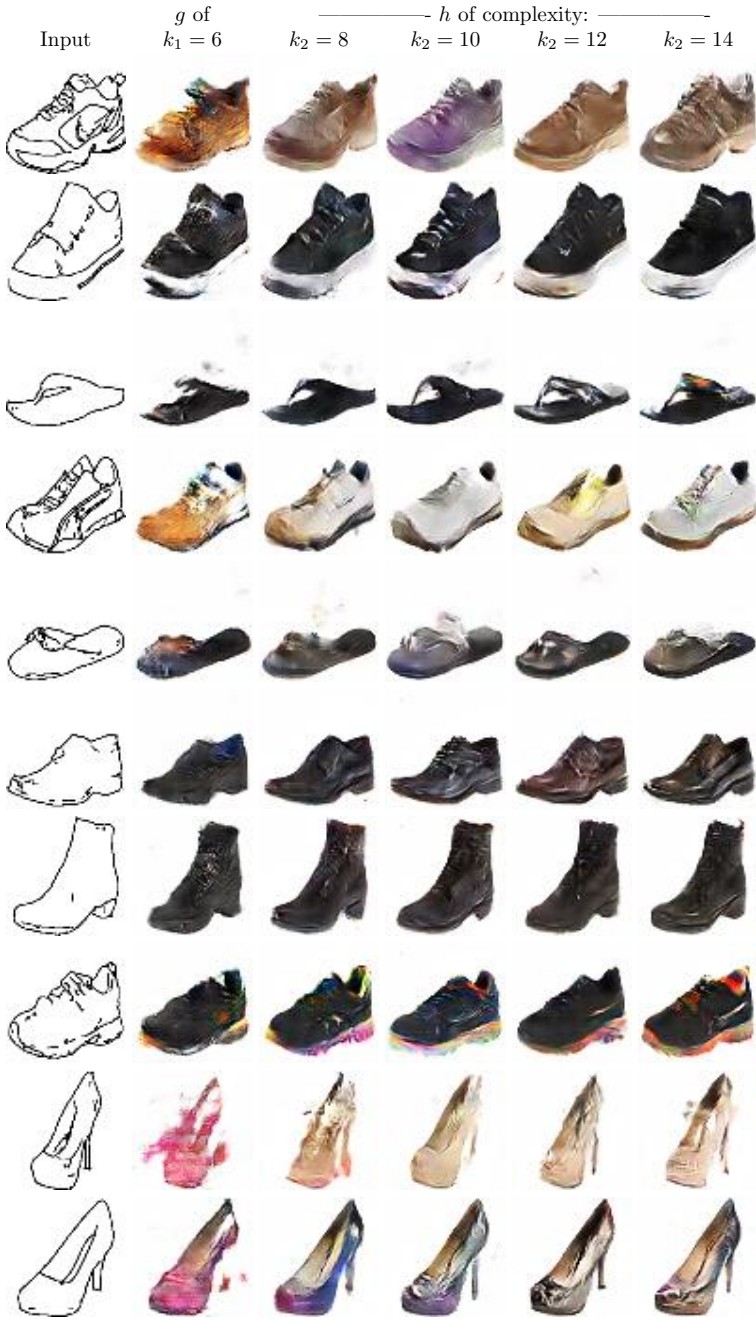

Figure 28: Results for Alg. 1 on Edges2Shoes dataset for mapping edges to shoes. Shown are a minimal complexity mapping $g$ that has low discrepancy, and various mappings $h$ obtained by the method.

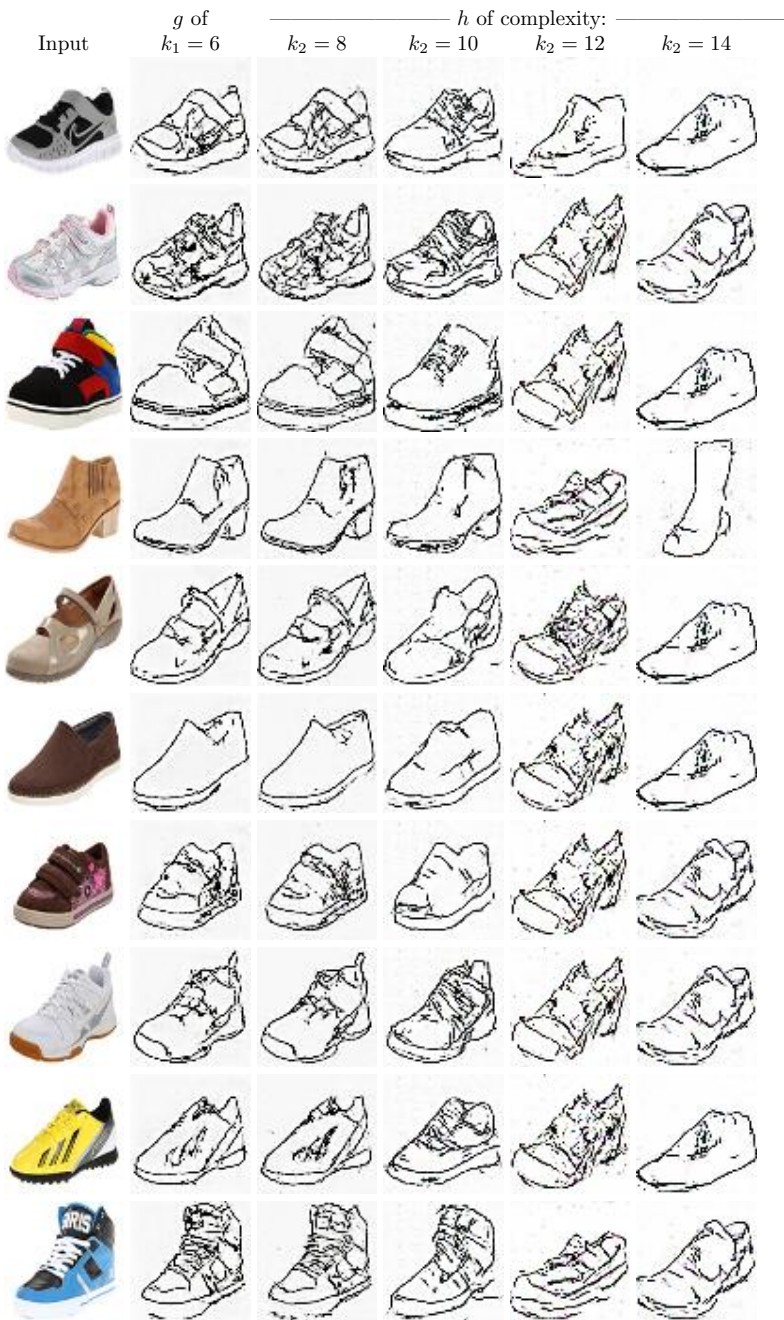

Figure 29: Results for Alg. 1 on Edges2Shoes dataset for mapping shoes to edges. Shown are a minimal complexity mapping $g$ that has low discrepancy, and various mappings $h$ obtained by the method.

# A   A GENERALIZED AND FORMAL STATEMENT OF THE RESULTS

For brevity, we have not presented our results in the most general way. For example, in Def. 1, we did not bound the complexity of the discriminators. For the same reason, some of our terms were described and not yet formally defined.

## A.1   A COMPLEXITY MEASURE FOR FUNCTIONS

In order to model the composition of neural networks, we define a complexity measurement that assigns a value based on the number of simple functions that make up a complex function.

**Definition 3** (Stratified complexity model (SCM)). *A stratified complexity model $\mathcal{N} := \mathrm{SCM}[\mathcal{C}]$ is a hypothesis class of functions $p : \mathbb{R}^M \to \mathbb{R}^M$ specified by a set of functions $\mathcal{C}$. Every function $p$ in $\mathcal{N}$ has an appropriate decomposition:*

- *$\mathcal{N} = \bigcup_{n=0}^{\infty} \mathcal{C}^n$ (where, $\mathcal{C}^n = \{p_n \circ ... \circ p_1 | p_1, ..., p_n \in \mathcal{C}\}$ and $\mathcal{C}^0 = \{\mathrm{Id}\}$).*

- *Every function in $\mathcal{C}$ is invertible.*

A SCM partitions a set of invertible functions into disjoint *complexity classes*,

$$\mathcal{C}_0 := \left\{ p \in \mathcal{N} \,\middle|\, \forall\, n \in \mathbb{N},\ q \in \mathcal{C}^n : p \circ q, q \circ p, p^{-1} \circ q, q \circ p^{-1} \in \mathcal{C}^n \right\}$$
$$\mathcal{C}_n := \mathcal{C}^n \setminus \left[ \bigcup_{i=0}^{n-1} \mathcal{C}^i \cup \mathcal{C}_0 \right]$$
(9)

When considering simple functions $p_i$ that are layers in a neural network, each complexity class contains the functions that are implemented by networks of $n$ hidden layers. In addition, we denote the *complexity of a function $p$*:

$$C(p) := \arg_{n \in \mathbb{N} \cup \{0\}} \{p \in \mathcal{C}_n\}$$
(10)

If the complexity of a function $p$ equals $n$, then any appropriate decomposition $p = p_n \circ ... \circ p_1$ will be called a *minimal decomposition* of $p$. According to this measurement, the complexity of a function $p$ is determined by the minimal number of primitive functions required in order to represent it.

In this work, we focus our attention on SCMs that represent the architectures of fully connected neural networks with layers of a fixed size, i.e.,

**Definition 4** (NN-SCM). *A NN-SCM is a SCM $\mathcal{N} = \mathrm{SCM}[\mathcal{C}]$ that satisfies the following conditions:*

- *$\mathcal{C} = \left\{ W_2 \circ \sigma \circ W_1 \,\middle|\, W_1, W_2 \in \mathbb{R}^{M \times M} \text{ and } W_1, W_2 \text{ are invertible} \right\}$. Here, $W_1, W_2$ denote both linear transformations and the associated matrix forms.*

- *$\sigma$ is a non-linear element-wise activation function.*

*For brevity, we denote $\mathcal{N} := \mathrm{SCM}[\sigma]$ to refer to a NN-SCM with the activation function $\sigma$.*

The NN-SCM with the Leaky ReLU activation function is of a particular interest, since (Kim et al., 2017; Zhu et al., 2017) employ it as the main activation function (plain ReLUs and $\tanh$ are also used). In the NN-SCM framework, to specify the function obtained by a decomposition $W_n \circ \sigma \circ W_{n-1} \circ \sigma \circ ... \circ \sigma \circ W_1$ we simply write:

$$F[W_n, ..., W_1] := W_n \circ \sigma \circ W_{n-1} \circ \sigma \circ ... \circ \sigma \circ W_1$$
(11)

It is useful to characterize the effect of inversion on the complexity of functions, since, for example, we consider both $h' = \Pi \circ h$ and $h = \Pi^{-1} \circ h'$. The following lemma states that, in the case of NN-SCM with $\sigma$ that is the Leaky ReLU, the complexity of the inverse function is the same as that of the original function.

**Lemma 1.** *Let $\mathcal{N} = \mathrm{SCM}[\sigma]$ be a NN-SCM with $\sigma$ that is the Leaky ReLU with parameter $0 < a \neq 1$. Then, for any $u \in \mathcal{N}$, $C(u^{-1}) = C(u)$.*

*Proof.* First, we denote $C'(p)$ the minimal number $n$ such that there are invertible linear mappings $W_1, ..., W_{n+1}$ such that $p = F[W_{n+1}, ..., W_1]$ (if $p = $ Id then $C'(p) = 0$). This complexity measure is similar to the complexity measure $C$. For a function $p$ such that $C(p) \neq 0$, we have, $C(p) = C'(p)$. Nevertheless, for $p$ such that $C(p) = 0$, it is not necessarily true that $C'(p) = 0$. For example, if $p \neq$ Id is an invertible linear mapping, we have, $C(p) = 0$ and $C'(p) = 2$. Let $p = F[W_2, W_1] = W_2 \circ \sigma \circ W_1$ be any function such that $C(p) = 1$. We consider that:

$$\sigma^{-1} = -\text{Id} \circ \sigma \circ -\text{Id}/a = F[-\text{Id}, -\text{Id}/a] \tag{12}$$

Therefore,

$$F[W_2, W_1]^{-1} = -W_1^{-1} \circ \sigma \circ -W_2^{-1}/a = F[-W_1^{-1}, -W_2^{-1}/a] \tag{13}$$

In particular, $C'(p^{-1}) \leq 1$. If $C'(p^{-1}) = 0$, then, Id $= -W_1^{-1} \circ \sigma \circ -W_2^{-1}/a$ and, therefore, $\sigma$ is a linear mapping - in contradiction. Thus, $C'(p^{-1}) = 1$.

Next, we would like to show that for any $u \in \mathcal{N}$, $C'(u^{-1}) = C'(u)$. Let $u$ such that $C'(u) = 0$. Then, $u = u^{-1} = $ Id and therefore, $C'(u^{-1}) = 0$. Let $u = F[W_{n+1}, ..., W_1]$ be a function such that $C'(u) = n > 0$. Then,

$$u = F[W_{n+1}, W_n] \circ F[\text{Id}, W_{n-1}] \circ ... \circ F[\text{Id}, W_1] \tag{14}$$

In particular,

$$u^{-1} = F[\text{Id}, W_1]^{-1} \circ ... \circ F[\text{Id}, W_{n-1}]^{-1} \circ F[W_{n+1}, W_n]^{-1} \tag{15}$$

or,

$$u^{-1} = F[-W_1^{-1}, W_2^{-1}/a, ..., W_n^{-1}/a, -W_{n+1}^{-1}/a] \tag{16}$$

Therefore, by Lem. 7,

$$C'(u^{-1}) \leq C'(F[\text{Id}, W_1]^{-1}) + ... + C'(F[\text{Id}, W_{n-1}]^{-1}) + C'(F[W_{n+1}, W_n]^{-1}) = n \tag{17}$$

On the other hand, if $v = u^{-1}$, $n = C'(u) = C'(v^{-1}) \leq C'(v) = C'(u^{-1}) \leq n$ and $C'(u^{-1}) = C'(u)$. Finally, we would like to show that for every $u \in \mathcal{N}$, we have: $C(u^{-1}) = C(u)$. If $C(u) = 0$, then, by Lem. 11, $C(u^{-1}) = 0$. On the other hand, if $C(u) \neq 0$, then, by Lem. 11, $C(u^{-1}) \neq 0$ and by the above: $C(u) = C'(u) = C'(u^{-1}) = C(u)$. $\qquad \square$

## A.2 MINIMAL COMPLEXITY MAPPINGS

Based on our simplicity hypothesis, we present a definition of a minimal complexity mapping that is both intuitive and well-defined in concrete complexity terms. Given two distributions $D_A$ and $D_B$, a minimal complexity mapping $f : \mathcal{X}_A \to \mathcal{X}_B$ between domains $A$ and $B$ is a mapping that has minimal complexity among the functions $h : \mathcal{X}_A \to \mathcal{X}_B$ that satisfy $h \circ D_A \approx D_B$.

Consider, again, the example of a line segment in $\mathbb{R}^M$ (Sec. 3.1) and the semantic space of the interval, $[0, 1] \subset \mathbb{R}$. The two linear mappings, which map either segment ends to $0$ and the other to $1$ are minimal, when using $f$ that are ReLU based neural networks. Other mappings to this segment are possible, simply by permuting points on the segment in $\mathbb{R}^M$. However, these alternative mappings have higher complexity, since the two mappings above are the only ones with the minimal possible complexity.

In order to measure the distance between $h \circ D_A$ and $D_B$, we use the discrepancy distance, $\text{disc}_{\mathcal{D}}$. In this work, we focus on classes of discriminators $\mathcal{D}$ of the form $\mathcal{D}_m := \{u | C(u) \leq m\}$ for some $m \in \mathbb{N}$. In addition, for simplicity, we will write $\text{disc}_m := \text{disc}_{\mathcal{D}_m}$.

**Definition 5** (Minimal complexity mappings)**.** *Let $\mathcal{N} = \text{SCM}[\mathcal{C}]$. Let $A = (\mathcal{X}_A, D_A)$ and $B = (\mathcal{X}_B, D_B)$ be two domains. We define the $(m, \epsilon_0)$-minimal complexity between $A$ and $B$ as:*

$$C_{A,B}^{m,\epsilon_0} := \min_{i \in \mathbb{N} \cup \{0\}} \{\exists h \text{ s.t } C(h) = i \text{ and } \text{disc}_m(h \circ D_A, D_B) \leq \epsilon_0\} \tag{18}$$

*The set of $(m, \epsilon_0)$-minimal complexity mappings between $A$ and $B$ is:*

$$H_{\epsilon_0}(A, B; m) := \left\{ h \Big| C(h) \leq C_{A,B}^{m,\epsilon_0} \text{ and } \text{disc}_m(h \circ D_A, D_B) \leq \epsilon_0 \right\} \tag{19}$$

We note that for any fixed $\epsilon_0 > 0$, the sequence $\{C_{A,B}^{m,\epsilon_0}\}_{m=0}^{\infty}$ is monotonically increasing as $m$ tends to infinity. In addition, we assume that for every two distributions of interest, $D_I$ and $D_J$, and an error rate $\epsilon_0 > 0$, there is a function $h$ of finite complexity such that $\text{disc}_{\infty}(h \circ D_I, D_J) \leq \epsilon_0$. Therefore, the sequence $\{C_{A,B}^{m,\epsilon_0}\}_{m=0}^{\infty}$ is upper bounded by $C(h)$ for all $m \in \mathbb{N} \cup \{0\}$. In particular, there is a minimal value $m_0 > 0$ such that $C_{A,B}^{m,\epsilon_0} = C_{A,B}^{m_0,\epsilon_0}$ for all $m \geq m_0$. We denote: $E_{A,B}^{\epsilon_0} := m_0$ and $C_{A,B}^{\epsilon_0} := C_{A,B}^{m_0,\epsilon_0}$. For simplicity, sometimes we will assume that $m = \infty$. In this case, we will write $H_{\epsilon_0}(A, B) := H_{\epsilon_0}(A, B; \infty)$.

## A.3  IDENTIFIABILITY

Every neural network implementation gives rise to many alternative implementations by performing simple operations, such as permuting the units of any hidden layer, and then permuting back as part of the linear mapping in the next layer. Therefore, it is first required to identify and address the set of transformations that could be inconsequential to the function which the network computes.

**Definition 6** (Invariant set). *Let $\mathcal{N} = \text{SCM}[\sigma]$ be a NN-SCM. The invariant set $\text{Invariant}(\mathcal{N})$ is the set of all $\tau : \mathbb{R}^M \to \mathbb{R}^M$ that satisfy the following conditions:*

- *$\tau : \mathbb{R}^M \to \mathbb{R}^M$ is an invertible linear transformation.*

- *$\sigma \circ \tau = \tau \circ \sigma$.*

*Functions in $\text{Invariant}(\mathcal{N})$ are called invariants or invariant functions.*

For example, for neural networks with the $\tanh$ activation function, the set of invariant functions contains the linear transformations that take vectors, permute them and multiply each coordinate by $\pm 1$. Formally, each $\tau = [\epsilon_1 \cdot \text{e}_{\pi(1)}, ..., \epsilon_M \cdot \text{e}_{\pi(M)}]$ where $\text{e}_i$ is the $i$'th standard basis vector, $\pi$ is a permutation over $[M]$ and $\epsilon_i \in \{\pm 1\}$ (Fefferman & Markel, 1993).

In the following lemma, we characterize the set of all invariant functions for $\sigma$ that is Leaky ReLU with parameter $0 < a \neq 1$.

**Lemma 2.** *Let $\mathcal{N} = \text{SCM}[\sigma]$ with $\sigma$ be Leaky ReLU with parameter $0 < a \neq 1$. Then,*

$$\text{Invariant}(\mathcal{N}) = \left\{ \tau \in \mathbb{R}^{M \times M} \ \Big| \ \tau = [c_1 \cdot \text{e}_{\pi(1)}, ..., c_M \cdot \text{e}_{\pi(M)}], \text{ where } \forall i \in [M] : c_i > 0 \text{ and } \pi \in \text{Sym}_M \right\}$$
(20)

*Here, $\text{e}_i$ denotes the $i$'th standard basis vector in $\mathbb{R}^M$ and $\text{Sym}_M$ is the set of permutations of $[M]$.*

*Proof.* Let $\tau$ be an invertible linear mapping satisfying $\sigma \circ \tau = \tau \circ \sigma$. We consider that for all $i \in [M]$ and vector $x$; $\sigma(\langle \tau_i, x \rangle) = \langle \tau_i, \sigma(x) \rangle$, where $\tau_i$ is the $i$'th row of $\tau$ and $\tau_{i,j}$ is the $(i, j)$ entry of $\tau$. For $x = \text{e}_j$, we have:

$$\tau_{i,j} = \sigma(\tau_{i,j}) \tag{21}$$

For $x = -\text{e}_j$, we have:

$$-a\tau_{i,j} = \sigma(-\tau_{i,j}) \tag{22}$$

If $\tau_{i,j} < 0$, then the first equation leads to contradiction. Otherwise, the equations are both satisfied.

Finally, for $x = \text{e}_j - \text{e}_k$, we have:

$$\tau_{i,j} - a\tau_{i,k} = \sigma(\tau_{i,j} - \tau_{i,k}) \tag{23}$$

If $\tau_{i,j} - \tau_{i,k} = 0$, then, $\tau_{i,j} - a\tau_{i,k} = 0$ and since $a \neq 1, 0$, we have, $\tau_{i,j} = \tau_{i,k} = 0$. If $\tau_{i,j} - \tau_{i,k} \geq 0$, then, $\tau_{i,j} - \tau_{i,k} = \tau_{i,j} - a\tau_{i,k}$ that gives $\tau_{i,k} = 0$. If $\tau_{i,j} - \tau_{i,k} \leq 0$, then, $a(\tau_{i,j} - \tau_{i,k}) = \tau_{i,j} - a\tau_{i,k}$ that yields $\tau_{i,j} = 0$. Therefore, for each $i \in [M]$ there is at most one entry $\tau_{i,j}$ that is not 0. If for all $j \in [M]$, $\tau_{i,j} = 0$, then the mapping $\tau$ is not invertible, in contradiction. Therefore, for each $i \in [M]$ there is exactly one entry $\tau_{i,j} > 0$ (it is non-negative as shown above). Finally, if there are $i_1 \neq i_2$ such that $\tau_{i_1,j}, \tau_{i_2,j} \neq 0$ then the matrix is invertible. Therefore, $\tau$ is a member of the set defined in Eq. 20. In addition, it is easy to see that every member of the noted set satisfies the conditions of the invariant set. Thus, we obtain the desired equation. $\square$

Our analysis is made much simpler, if every function has one invariant representation up to a sequence of manipulations using invariant functions that do not change the essence of the processing at each layer.

**Assumption 1** (Identifiability). *Let $\mathcal{N} = \text{SCM}[\sigma]$ with $\sigma$ that is Leaky ReLU with parameter $0 < a \neq 1$. Then, every function $p \in \mathcal{N}$ is identifiable (with respect to $\text{Invariant}(\mathcal{N})$), i.e., for any two minimal decompositions, $p = F[W_{n+1}, ..., W_1] = F[V_{n+1}, ..., V_1]$, there are invariants $\tau_1, ..., \tau_n \in \text{Invariant}(\mathcal{N})$ such that:*

$$V_1 = \tau_1 \circ W_1, \ \forall i = 2, ..., n : V_i = \tau_i \circ W_i \circ \tau_{i-1}^{-1} \text{ and } V_{n+1} = W_{n+1} \circ \tau_n^{-1} \tag{24}$$

Uniqueness up to invariants, also known as identifiability, forms an open question. Fefferman & Markel (1993) proved identifiability for the $\tanh$ activation function. Other works (Williamson & Helmke, 1995; F. Albertini & Maillot, 1993; Kurková & Kainen, 2014; Sussmann, 1992) prove such uniqueness for neural networks with only one hidden layer and various classical activation functions. In the following lemma, we show that identifiability holds for Leaky ReLU networks with only one hidden layer.

**Lemma 3.** *Let $\mathcal{N} = \text{SCM}[\sigma]$ with $\sigma$ that is Leaky ReLU with parameter $0 < a \neq 1$. Any function $p$ such that $C(p) = 1$ is identifiable, i.e, if $p = F[W_2, W_1] = F[V_2, V_1]$, then, $W_1 = \tau \circ V_1$ and $W_2 = V_2 \circ \tau^{-1}$ for some $\tau \in \text{Invariant}(\mathcal{N})$.*

*Proof.* An alternative representation of the equation is:

$$(\sigma \circ W_1 \circ V_1^{-1}) = (W_2^{-1} \circ V_2 \circ \sigma) \tag{25}$$

We would like to prove that if $\sigma \circ U = V \circ \sigma$ then $V = U$. We have:

$$\sigma \circ U(x) = V \circ \sigma(x) \tag{26}$$

In particular, if $v_i$ is the $i$'th row of $V$ (similarly $u_i$) and $x = e_j$:

$$\sigma(u_{i,j}) = \sigma(\langle u_i, e_j^\top \rangle) = \langle v_i, \sigma(e_j^\top) \rangle = v_{i,j} \tag{27}$$

where $v_{i,j}$ is the $(i, j)$ entry of $V$ (similarly $u_{i,j}$). Similarly, for $x = -e_j$:

$$\sigma(-u_{i,j}) = \sigma(\langle u_i, -e_j^\top \rangle) = \langle v_i, \sigma(-e_j^\top) \rangle = -av_{i,j} \tag{28}$$

If $u_{i,j}$ is negative, we obtain: $au_{i,j} = v_{i,j}$ (the first equation) and $-u_{i,j} = -av_{i,j}$ (the second equation) that yields $a = 1$ in contradiction. Therefore, $u_{i,j} \geq 0$ and $u_{i,j} = v_{i,j}$ (the second equation).

We conclude that $W_1 \circ V_1^{-1} = W_2^{-1} \circ V_2 := \tau$. Finally, since $(\sigma \circ W_1 \circ V_1^{-1}) = (W_2^{-1} \circ V_2 \circ \sigma)$ we have $\sigma \circ \tau = \tau \circ \sigma$ and $\tau$ is invertible linear mapping. Differently said, $W_1 = \tau \circ V_1$ and $W_2 = V_2 \circ \tau^{-1}$ such that $\tau \in \text{Invariant}(\mathcal{N})$. $\square$

As far as we know, there are no other results continuing the identifiability line of work for activation functions such as Leaky ReLU. Uniqueness, which is stronger than identifiability, since it means that even multiple representations with different number of layers do not exist, does not hold for these activation functions. To see this, note that for every $M \times M$ invertible linear mapping $W$, the following holds:

$$U \circ \sigma \circ W = U \circ \sigma \circ W \circ \sigma \circ -\text{Id} \circ \sigma \circ -\text{Id}/a \tag{29}$$

where $\sigma$ is the Leaky ReLU activation function with parameter $a$. We conjecture that for networks with Leaky ReLU activations identifiability holds, or at least for networks with a fixed number of neurons per layer. In addition to identifiability, we make the following assumption, which states that almost all mappings are non-degenerate.

**Assumption 2.** *Let $\mathcal{N} = \text{SCM}[\sigma]$ with $\sigma$ that is Leaky ReLU with parameter $0 < a \neq 1$. Assume that the set of $(W_1, ..., W_{n+1}) \in \mathbb{R}^{M \times M \times m}$ such that $C(F[W_{n+1}, ..., W_1]) = n$ is dense in $\mathbb{R}^{M \times M \times m}$.*

### A.4 COUNTING MINIMAL COMPLEXITY MAPPINGS

In the unsupervised alignment problem, the algorithms are provided with only two unmatched datasets of samples from the domains $A$ and $B$ and the task is to learn a well-aligned function between them. Since we hypothesize that the alignment of the target mapping is typically captured by the lowest complexity low-discrepancy mapping, we develop the machinery needed in order to show that such mappings are rare.

Recall that $\mathrm{disc}_m$ is the discrepancy distance for discriminators of complexity up to $m$. In Sec. 2, we have discussed the functions $\Pi$ which replaces between members in the domain $B$ that have similar probabilities. Formally, these are defined using the discrepancy distance.

**Definition 7** (Density preserving mapping). *Let $\mathcal{N} = \mathrm{SCM}[\mathcal{C}]$ and $X = (\mathcal{X}, D_X)$ a domain. A $(m, \epsilon_0)$-density preserving mapping over $X$ (or an $(m, \epsilon_0)$-DPM for short) is a function $\Pi$ such that*

$$\mathrm{disc}_m(\Pi \circ D_X, D_X) \leq \epsilon_0 \tag{30}$$

*We denote the set of all $(m, \epsilon_0)$-DPMs of complexity $k$ by $\mathrm{DPM}_{\epsilon_0}(X; m, k) := \left\{ \Pi \big| \mathrm{disc}_m(\Pi \circ D_X, D_X) \leq \epsilon_0 \text{ and } C(\Pi) = k \right\}$.*

We would like to bound the number of mappings that are both low-discrepancy and low-complexity by the number of DPMs. We consider that there are infinitely many DPMs. For example, if we slightly perturb the weights of a minimal representation of a DPM, $\Pi$, we obtain a new DPM. Therefore, we define a similarity relation between functions that reflects whether the two are similar. In this way, we are able to bound the number of different (non-similar) minimal-complexity mappings by the number of different DPMs.

**Definition 8** (Closeness between pairs of distributions or functions). *Let $\mathcal{N} = \mathrm{SCM}[\sigma]$.*

- *We denote $D_1 \underset{m,\epsilon_0}{\sim} D_2 \iff \mathrm{disc}_m(D_1, D_2) \leq \epsilon_0$.*

- *We denote $f \underset{m,\epsilon_0}{\overset{D}{\sim}} g$, if $C(f) = C(g) =: n$ and there are minimal decompositions: $f = F[W_{n+1}, ..., W_1]$ and $g = F[V_{n+1}, ..., V_1]$ such that $\forall i \in [n+1]: F[W_i, ..., W_1] \circ D \underset{m,\epsilon_0}{\sim} F[V_i, ..., V_1] \circ D$.*

The defined relation is reflexive and symmetric, but not transitive. Therefore, there are many different ways to partition the space of functions into disjoint subsets such that in each subset, any two functions are similar. We count the number of functions up to the similarity as the minimal number of subsets required in order to cover the entire space. This idea is presented in Def. 9, which slightly generalizes the notion of covering numbers (Anthony & Bartlett, 2009).

**Definition 9** (Covering number). *Let $(\mathcal{U}, \sim_{\mathcal{U}})$ be a set and a reflexive and symmetric relation. A covering of $(\mathcal{U}, \sim_{\mathcal{U}})$, is a tuple $(\mathcal{U}, \equiv_{\mathcal{U}})$ such that: $\equiv_{\mathcal{U}}$ is an equivalence relation and $u_1 \equiv_{\mathcal{U}} u_2 \implies u_1 \sim_{\mathcal{U}} u_2$. The covering number of $(\mathcal{U}, \sim_{\mathcal{U}})$, denoted by $\mathrm{N}(\mathcal{U}, \sim_{\mathcal{U}})$, is:*

$$\min \big| \mathcal{U}/ \equiv_{\mathcal{U}} \big| \text{ s.t: the minimum is taken over } (\mathcal{U}, \equiv_{\mathcal{U}}) \text{ that is a covering of } (\mathcal{U}, \sim_{\mathcal{U}}) \tag{31}$$

*Here, $\mathcal{U}/ \equiv_{\mathcal{U}}$ is the quotient set of $\mathcal{U}$ by $\equiv_{\mathcal{U}}$.*

Thm. 1 below states that the number of low discrepancy mappings of complexity $C_{A,B}^{\epsilon_0}$ is upper bounded by the number of DPMs of size $2C_{A,B}^{\epsilon_0}$. By prediction 3, the number of such DPMs is small. The theorem employs the following weak assumption. In Lem. 19, we prove that this assumption holds for the case of a continuous risk if the discriminators have bounded weights.

**Assumption 3.** *Let $\mathcal{N} = \mathrm{SCM}[\sigma]$ with $\sigma$ that is Leaky ReLU with parameter $0 < a \neq 1$. For every $m > 0$ (possibly $\infty$) and $n > 0$, the function $\mathrm{disc}_m(F[W_n, ..., W_1] \circ D_1, D_2)$ is continuous as a function of the weights of $W_1, ..., W_n \in \mathbb{R}^{M \times M}$.*

**Theorem 1.** *Let $\mathcal{N} = \mathrm{SCM}[\sigma]$ with $\sigma$ that is Leaky ReLU with parameter $0 < a \neq 1$. Assume Assumptions 1, 2 and 3. Let $\epsilon_0$, $\epsilon_1$ and $\epsilon_2$ be three constants such that $\epsilon_0 < \epsilon_1/4$ and $\epsilon_2 < \epsilon_1 - 4\epsilon_0$ be three positive constants and $A = (\mathcal{X}_A, D_A)$ and $B = (\mathcal{X}_B, D_B)$ are two domains. Then,*

$$\mathrm{N}\left(H_{\epsilon_0}(A,B), \overset{D_A}{\underset{\epsilon_1}{\sim}}\right) \leq \lim_{\epsilon \to 0} \min \begin{cases} \mathrm{N}\left(\mathrm{DPM}_{2\epsilon_0+\epsilon}\left(A; 2C_{A,B}^{\epsilon_0}\right), \overset{D_A}{\underset{\epsilon_2}{\sim}}\right) \\ \mathrm{N}\left(\mathrm{DPM}_{2\epsilon_0+\epsilon}\left(B; 2C_{A,B}^{\epsilon_0}\right), \overset{D_B}{\underset{\epsilon_2}{\sim}}\right) \end{cases} \tag{32}$$

*Proof.* See Sec. D.

# B   SUMMARY OF NOTATION

Tab. 5 lists the symbols used in our work.

Table 5: Summary of Notation

| Symbol | Explanation |
|---|---|
| $\mathcal{X}$ | A feature space |
| $\mathcal{X}_A, \mathcal{X}_B$ | The sample spaces of $A$ and $B$ (resp.) |
| $D_A, D_B$ | Distributions over $\mathcal{X}_A$ and $\mathcal{X}_B$ (resp.) |
| $A, B$ | Two domains; Specified by $(\mathcal{X}_A, D_A)$ and $(\mathcal{X}_B, D_B)$ (resp.) |
| $y_A, y_B$ | Functions from the feature space to the domains, $y_A : \mathcal{X} \to \mathcal{X}_A$ and $y_B : \mathcal{X} \to \mathcal{X}_B$ |
| $D_Z$ | A distribution over a feature space $\mathcal{X}$ |
| $y_{AB}, y_{BA}$ | $y_{AB} = y_B \circ y_A^{-1}$ and $y_{BA} = y_A \circ y_B^{-1}$ |
| $\ell$ | Loss function $\ell : \mathbb{R} \times \mathbb{R} \to \mathbb{R}$ |
| $R_D[f_1, f_2]$ | The risk function $R_D[f_1, f_2] = \mathbb{E}_{x \sim D} \ell(f_1(x), f_2(x))$ where $\ell$ is a loss function and $D$ is a distribution |
| $\text{disc}_{\mathcal{D}}(D_1, D_2)$ | The discrepancy between two distributions $D_1$ and $D_2$, i.e, $\text{disc}_{\mathcal{D}}(D_1, D_2) = \sup_{c_1, c_2 \in \mathcal{D}} \|R_{D_1}[c_1, c_2] - R_{D_2}[c_1, c_2]\|$ |
| $\sigma$ | A non-linear element-wise activation function |
| $\mathcal{C}$ | A class of functions; in most cases $\mathcal{C} = \{W_2 \circ \sigma \circ W_1 \mid W_1, W_2 \in \mathbb{R}^{M \times M}$ are invertible linear transformation$\}$ |
| $F[W_n, ..., W_1]$ | $F[W_n, ..., W_1] = W_n \circ \sigma \circ W_n \circ \sigma \circ ... \circ \sigma \circ W_2 \circ \sigma \circ W_1$ |
| $\mathcal{N} = \text{SCM}[\mathcal{C}]$ | A SCM specified by a class of functions $\mathcal{C}$ (see Def. 3) |
| $\mathcal{N} = \text{SCM}[\sigma]$ | A NN-SCM specified by the activation function $\sigma$ (see Def. 4) |
| $C(p)$ | The complexity of a function $p$ (see Eqs. 9, 10) |
| $\text{Invariant}(\mathcal{N})$ | The invariant set of $\mathcal{N}$ (see Def. 6) |
| $\tau$ | An invariant function (see Def. 6) |
| $\mathcal{D}_m$ | $\mathcal{D}_m = \{u | C(u) \le m\}$ |
| $\text{disc}_m, \text{disc}$ | $\text{disc}_m := \text{disc}_{\mathcal{D}_m}$ and $\text{disc} := \text{disc}_{\mathcal{D}_\infty}$ |
| $C_{A,B}^{m,\epsilon_0}$ | The $(m, \epsilon_0)$-minimal complexity between $A$ and $B$ (see Def. 5) |
| $C_{A,B}^{\epsilon_0}, E_{A,B}^{\epsilon_0}$ | $C_{A,B}^{\epsilon_0} = \max_{m \ge 1} C_{A,B}^{m,\epsilon_0}$ and $E_{A,B}^{\epsilon_0} = \arg\min_m [C_{A,B}^{m,\epsilon_0} = C_{A,B}^{\epsilon_0}]$ |
| $H_{\epsilon_0}(A, B; m)$ | The set of $(\epsilon_0, m)$-minimal complexity mappings between $A$ and $B$ (see Def. 5) |
| $H_{\epsilon_0}(A, B)$ | $H_{\epsilon_0}(A, B) = H_{\epsilon_0}(A, B; \infty)$ |
| $S_1 \circ S_2$ | A composition of sets, $S_1 \circ S_2 = \{s_1 \circ s_2 | s_1 \in S_1$ and $s_2 \in S_2\}$ |
| $D_1 \underset{m,\epsilon}{\sim} D_2, D_1 \underset{\epsilon}{\sim} D_2$ | $\text{disc}_m(D_1, D_2) \le \epsilon$ and $\text{disc}(D_1, D_2) \le \epsilon$ (see Def. 8) |
| $f \overset{D}{\underset{m,\epsilon}{\sim}} g, f \overset{D}{\underset{\epsilon}{\sim}} g$ | $\text{disc}_m(f \circ D, g \circ D) \le \epsilon$ and $\text{disc}(f \circ D, g \circ D) \le \epsilon$ (see Def. 8) |
| $\text{N}(\mathcal{U}, \sim_{\mathcal{U}})$ | The covering number of $\mathcal{U}$ with respect to relation $\sim_{\mathcal{U}}$ on $\mathcal{U}$ (see Def.9) |
| $X :\leftarrow x$ | $x$ is assigned to $X$ |

## C    LEMMAS

In this section, we prove various lemmas that are used in the proof of Thm. 1. In Sec. C.1 we present the assumptions taken in various lemmas in the appendix. In Sec. C.2 we prove useful inequalities involving the discrepancy distance. Sec. C.3 provides lemmas concerning the defined complexity measure and invariant functions. The lemmas in Sec. C.4 concern the properties of inverse functions.

### C.1    ASSUMPTIONS

We list the assumptions employed in our proofs. Assumptions 1 and 2 were already presented and are heavily used. Assumptions 3 and its relaxation 4 are mild assumptions that were taken for convenience.

**Assumption 1** (Identifiability). *Let $\mathcal{N} = \text{SCM}[\sigma]$ with $\sigma$ that is Leaky ReLU with parameter $0 < a \neq 1$. Then, every function $p \in \mathcal{N}$ is identifiable (with respect to $\text{Invariant}(\mathcal{N})$), i.e., for any two minimal decompositions, $p = F[W_{n+1}, ..., W_1] = F[V_{n+1}, ..., V_1]$, there are invariants $\tau_1, ..., \tau_n \in \text{Invariant}(\mathcal{N})$ such that:*

$$V_1 = \tau_1 \circ W_1, \ \forall i = 2, ..., n : V_i = \tau_i \circ W_i \circ \tau_{i-1}^{-1} \text{ and } V_{n+1} = W_{n+1} \circ \tau_n^{-1} \tag{24}$$

**Assumption 2.** *Let $\mathcal{N} = \text{SCM}[\sigma]$ with $\sigma$ that is Leaky ReLU with parameter $0 < a \neq 1$. Assume that the set of $(W_1, ..., W_{n+1}) \in \mathbb{R}^{M \times M \times m}$ such that $C(F[W_{n+1}, ..., W_1]) = n$ is dense in $\mathbb{R}^{M \times M \times m}$.*

**Assumption 3.** *Let $\mathcal{N} = \text{SCM}[\sigma]$ with $\sigma$ that is Leaky ReLU with parameter $0 < a \neq 1$. For every $m > 0$ (possibly $\infty$) and $n > 0$, the function $\text{disc}_m(F[W_n, ..., W_1] \circ D_1, D_2)$ is continuous as a function of the weights of $W_1, ..., W_n \in \mathbb{R}^{M \times M}$.*

In the case that the norm of the discriminator is bounded, Lem 19, it follows from the following assumption, which is well-justified, (cf. Shalev-Shwartz & Ben-David (2014), page 162, Eq.14.13).

**Assumption 4.** *Let $\mathcal{N} = \text{SCM}[\sigma]$ with $\sigma$ that is Leaky ReLU with parameter $0 < a \neq 1$. For all $m > 0$, the function $R_D[F[V_m, ..., V_1], F[W_m, ..., W_1]]$ is continuous as a function of $V_m, ..., V_1, W_m, ..., W_1$.*

### C.2    PROPERTIES OF DISCREPANCIES

**Lemma 4.** *Let $\mathcal{D}_1$ and $\mathcal{D}_2$ be two classes of functions and $D_1$, $D_2$ two distributions. Assume that $\mathcal{D}_1 \circ \{p\} \subset \mathcal{D}_2$, then,*
$$\text{disc}_{\mathcal{D}_1}(p \circ D_1, p \circ D_2) \leq \text{disc}_{\mathcal{D}_2}(D_1, D_2) \tag{33}$$
*In particular, if $m \geq k + C(p)$, then,*
$$\text{disc}_k(p \circ D_1, p \circ D_2) \leq \text{disc}_m(D_1, D_2) \tag{34}$$

*Proof.* By the definition of discrepancy:

$$\begin{aligned}\text{disc}_{\mathcal{D}_1}(p \circ D_1, p \circ D_2) &= \sup_{c_1, c_2 \in \mathcal{D}_1} \left| R_{p \circ D_1}[c_1, c_2] - R_{p \circ D_2}[c_1, c_2] \right| \\ &= \sup_{c_1, c_2 \in \mathcal{D}_1} \left| R_{D_1}[c_1 \circ p, c_2 \circ p] - R_{D_2}[c_1 \circ p, c_2 \circ p] \right| \end{aligned} \tag{35}$$

Since $\mathcal{D}_1 \circ \{p\} \subset \mathcal{D}_2$ we have:

$$\begin{aligned}\text{disc}_{\mathcal{D}_1}(p \circ D_1, p \circ D_2) &= \sup_{c_1, c_2 \in \mathcal{D}_1} \left| R_{D_1}[c_1 \circ p, c_2 \circ p] - R_{D_2}[c_1 \circ p, c_2 \circ p] \right| \\ &\leq \sup_{u_1, u_2 \in \mathcal{D}_2} \left| R_{D_1}[u_1, u_2] - R_{D_2}[u_1, u_2] \right| = \text{disc}_{\mathcal{D}_2}(D_1, D_2) \end{aligned} \tag{36}$$

The second inequality is a special case for $\mathcal{D}_1 = \mathcal{D}_k$ and $\mathcal{D}_2 = \mathcal{D}_m$.    $\square$

**Lemma 5.** *Let $A = (\mathcal{X}_1, D_1)$ and $B = (\mathcal{X}_2, D_2)$ be two domains and $D_Z$ a distribution.*

1. *Assume that $m \geq k + C(p)$. Then,*
$$\text{disc}_k(p \circ D_1, D_3) \leq \text{disc}_m(D_1, D_2) + \text{disc}_k(p \circ D_2, D_3) \tag{37}$$

2. *Let $y_1$, $y_2$ and $y = y_2 \circ y_1^{-1}$ be three functions and $m \geq k + C(y_2)$. Then,*
$$\text{disc}_k(y \circ D_1, D_2) \leq \text{disc}_m(D_Z, y_1^{-1} \circ D_1) + \text{disc}_k(y_2 \circ D_Z, D_2) \tag{38}$$

3. *Let $h$ be any function and $m \geq k + C(h^{-1})$. Then,*
$$\text{disc}_k(D_1, h^{-1} \circ D_2) \leq \text{disc}_m(h \circ D_1, D_2) \tag{39}$$

*Proof.* 1. Follows from Lem. 4, since $m \geq k + C(p)$, we have:
$$\text{disc}_k(p \circ D_1, p \circ D_2) \leq \text{disc}_m(D_1, D_2) \tag{40}$$
Therefore, by the triangle inequality,
$$
\begin{aligned}
\text{disc}_k(p \circ D_1, D_3) &\leq \text{disc}_k(p \circ D_1, p \circ D_2) + \text{disc}_k(p \circ D_2, D_3) \\
&\leq \text{disc}_m(D_1, D_2) + \text{disc}_k(p \circ D_2, D_3)
\end{aligned}
\tag{41}
$$
2. We use Lem. 4 with $p :\leftarrow y_2$, $\mathcal{D}_1 :\leftarrow \mathcal{D}_k$, and $\mathcal{D}_2 :\leftarrow \mathcal{D}_m$ and $\mathcal{D}_k \circ \{y_2\} \subset \mathcal{D}_2$:

$$\text{disc}_k(y_2 \circ D_Z, y \circ D_1) = \text{disc}_k(y_2 \circ D_Z, y_2 \circ y_1^{-1} \circ D_1) \leq \text{disc}_m(D_Z, y_1^{-1} \circ D_1) \tag{42}$$
Therefore, by the triangle inequality,
$$
\begin{aligned}
\text{disc}_k(y \circ D_1, D_2) &\leq \text{disc}_k(y_2 \circ D_Z, D_2) + \text{disc}_k(y_2 \circ D_Z, y \circ D_1) \\
&\leq \text{disc}_k(y_2 \circ D_Z, D_2) + \text{disc}_m(D_Z, y_1^{-1} \circ D_1)
\end{aligned}
\tag{43}
$$
3. Follows immediately from Lem. 4 for $p :\leftarrow h^{-1}$ and $\mathcal{D}_k \circ \{h^{-1}\} \subset \mathcal{D}_m$. $\qquad\square$

### C.3 PROPERTIES OF THE COMPLEXITY MEASURE AND INVARIANTS

**Lemma 6.** *Let $\mathcal{N} = \text{SCM}[\mathcal{C}]$. In addition, let $u, v$ be any two functions. Then,*
$$\max\{C(u) - C(v^{-1}), C(v) - C(u^{-1})\} \leq C(u \circ v) \leq C(u) + C(v) \tag{44}$$

*Proof.* We begin with the case $C(v) = 0$. In this case, $C(u \circ v) = C(u) = C(u) + C(v)$. By definition, $C(v) = 0$ implies that $C(v^{-1}) = 0$ and $C(u \circ v) = C(u) - C(v^{-1})$. Finally, $C(u) - C(v^{-1}) = C(u) = C(u \circ v)$. The case $C(u) = 0$ is analogous. Next, we assume that $C(u) = n > 0$ and $C(v) = m > 0$. Let $u = u_n \circ ... \circ u_1$ and $v = v_m \circ ... \circ v_1$ be minimal decompositions of $u$ and $v$ (resp.). Therefore, we can represent, $u \circ v = u_n \circ ... \circ u_1 \circ v_m \circ ... \circ v_1$. In particular, $C(u \circ v) \leq n + m = C(u) + C(v)$.

The lower bound follows immediately from the upper bound:
$$C(u) = C(u \circ v \circ v^{-1}) \leq C(u \circ v) + C(v^{-1}) \implies C(u) - C(v^{-1}) \leq C(u \circ v) \tag{45}$$
By similar considerations, we also have: $C(v) - C(u^{-1}) \leq C(u \circ v)$. $\qquad\square$

For a given function $u \in \mathcal{N} = \text{SCM}[\mathcal{C}]$, we define,
$$C'(u) = \arg_n\{u \in \mathcal{C}^n\} \tag{46}$$
**Lemma 7.** *Let $\mathcal{N} = \text{SCM}[\mathcal{C}]$. In addition, let $u, v$ be any two functions. Then,*
$$\max\{C'(u) - C'(v^{-1}), C'(v) - C'(u^{-1})\} \leq C'(u \circ v) \leq C'(u) + C'(v) \tag{47}$$

*Proof.* We begin by proving the upper bound. We assume $C'(u) = n$ and $C'(v) = m$. Let $u = u_n \circ ... \circ u_1$ and $v = v_m \circ ... \circ v_1$ be minimal decompositions of $u$ and $v$ (resp.). Therefore, we can represent, $u \circ v = u_n \circ ... \circ u_1 \circ v_m \circ ... \circ v_1$. In particular, $C'(u \circ v) \leq n + m = C'(u) + C'(v)$. The lower bound follows immediately from the upper bound:

$$C'(u) = C'(u \circ v \circ v^{-1}) \leq C'(u \circ v) + C'(v^{-1}) \implies C'(u) - C'(v^{-1}) \leq C'(u \circ v) \tag{48}$$
By similar considerations, $C'(v) - C'(u^{-1}) \leq C'(u \circ v)$. $\qquad\square$

**Lemma 8.** Invariant($\mathcal{N}$) *is closed under inverse and composition, i.e,*

$$\tau \in \text{Invariant}(\mathcal{N}) \iff \tau^{-1} \in \text{Invariant}(\mathcal{N}) \tag{49}$$

*And,*

$$\tau_1, \tau_2 \in \text{Invariant}(\mathcal{N}) \implies \tau_1 \cdot \tau_2 \in \text{Invariant}(\mathcal{N}) \tag{50}$$

*Proof.* **Inverse:** Let $\tau \in \text{Invariant}(\mathcal{N})$. Then, by definition, $\tau$ is an invertible linear mapping and $\tau \circ \sigma = \sigma \circ \tau$. In particular, $\tau^{-1}$ is also an invertible linear mapping and $\tau^{-1} \circ \sigma = \sigma \circ \tau^{-1}$. Thus, $\tau^{-1} \in \text{Invariant}(\mathcal{N})$.

**Composition:** Let $\tau_1, \tau_2 \in \text{Invariant}(\mathcal{N})$. Then, $\tau_i$ is an invertible linear mapping and $\tau_i \circ \sigma = \sigma \circ \tau_i$ for $i = 1, 2$. In particular, $\tau_1 \circ \tau_2$ is also an invertible linear mapping and $\tau_1 \circ \tau_2 \circ \sigma = \tau_1 \circ \sigma \circ \tau_2 = \sigma \circ \tau_1 \circ \tau_2$. Thus, $\tau_1 \circ \tau_2 \in \text{Invariant}(\mathcal{N})$. □

**Lemma 9.** *Let $\mathcal{N} = \text{SCM}[\sigma]$ with $\sigma$ that is Leaky ReLU with $0 < a \neq 1$. Assume that $p$ obeys identifiability, i.e., that Assumption 1 holds. Then, for any two minimal decompositions $p = F[W_{n+1}, ..., W_1] = F[V_{n+1}, ..., V_1]$, we have:*

$$\forall i \in [n+1] : F[W_i, ..., W_1] \circ F[V_i, ..., V_1]^{-1} \in \text{Invariant}(\mathcal{N})$$
$$\text{and } F[W_{n+1}, ..., W_i] \circ F[V_{n+1}, ..., V_i]^{-1} \in \text{Invariant}(\mathcal{N}) \tag{51}$$

*Proof.* We prove that $F[W_i, ..., W_1] \circ F[V_i, ..., V_1]^{-1} \in \text{Invariant}(\mathcal{N})$. If $i = n + 1$, then, $F[W_i, ..., W_1] \circ F[V_i, ..., V_1]^{-1} = \text{Id} \in \text{Invariant}(\mathcal{N})$. Otherwise, by minimal identifiability,

$$V_1 = \tau_1 \circ W_1, \ \forall i = 2, ..., n : V_i = \tau_i \circ W_i \circ \tau_{i-1}^{-1} \text{ and } V_{n+1} = W_{n+1} \circ \tau_n^{-1} \tag{52}$$

In addition,

$$F[W_i, ..., W_1] = W_i \circ \sigma \circ W_{i-1} \circ ... \circ \sigma \circ W_1$$
$$F[V_i, ..., V_1] = (\tau_i \circ W_i \circ \tau_{i-1}^{-1}) \circ \sigma \circ (\tau_{i-1} \circ W_{i-1} \circ \tau_{i-2}^{-1}) \circ ... \circ \sigma \circ (\tau_1 \circ W_1) \tag{53}$$

Since each for all $k \in [i]$, $\tau_k$ commutes with $\sigma$, we have,

$$F[V_i, ..., V_1] = \tau_i \circ F[W_i, ..., W_1] \tag{54}$$

and

$$F[W_i, ..., W_1] \circ F[V_i, ..., V_1]^{-1} = \tau_i^{-1} \in \text{Invariant}(\mathcal{N}) \tag{55}$$

By similar considerations, $F[W_{n+1}, ..., W_i] \circ F[V_{n+1}, ..., V_i]^{-1} \in \text{Invariant}(\mathcal{N})$. □

**Lemma 10.** *Let $\mathcal{N} = \text{SCM}[\sigma]$ with $\sigma$ that is Leaky ReLU with parameter $0 < a \neq 1$. Then, every invertible linear mapping $W$ is a member of $\mathcal{C}_0$.*

*Proof.* Let $p \in \mathcal{C}^n$. Then, $p = F[W_{n+1}, ..., W_1]$, for invertible linear mappings $W_1, ..., W_{n+1}$. In particular, $W \circ p = F[W \cdot W_{n+1}, ..., W_1] \in \mathcal{C}^n$, $p \circ W = F[W_{n+1}, ..., W_1 \cdot W] \in \mathcal{C}^n$ and similarly, $W^{-1} \circ p, p \circ W^{-1} \in \mathcal{C}^n$. Therefore, $W \in \mathcal{C}_0$. □

**Lemma 11.** $\mathcal{C}_0$ *is closed under inverse and composition, i.e,*

$$u \in \mathcal{C}_0 \iff u^{-1} \in \mathcal{C}_0 \tag{56}$$

*and,*

$$u_1, u_2 \in \mathcal{C}_0 \implies u_1 \circ u_2 \in \mathcal{C}_0 \tag{57}$$

*Proof.* **Inverse:** By definition, $u \in \mathcal{C}_0$ iff for all $n \in \mathbb{N}$ and $q \in \mathcal{C}^n$, we have: $u \circ q, q \circ u, u^{-1} \circ q, q \circ u^{-1} \in \mathcal{C}^n$ iff $u^{-1} \in \mathcal{C}_0$.

**Decomposition:** Let $f \in \mathcal{C}^n$. Then, $g = u_1 \circ f \in \mathcal{C}^n$ and $u_2 \circ u_1 \circ f = u_2 \circ g \in \mathcal{C}^n$. Similarly, $u_1^{-1} \circ u_2^{-1} \circ f, f \circ u_1^{-1} \circ u_2^{-1}, f \circ u_2 \circ u_1 \in \mathcal{C}^n$. □

## C.4 Properties of Inverses

**Lemma 12.** *Let $\mathcal{N} = \mathrm{SCM}[\sigma]$ where $\sigma$ is the Leaky ReLU activation function, with parameter $0 < a \neq 1$. Let $f = F[W_{n+1}, ..., W_1]$ be a minimal decomposition. Then, for all $i \in [n]$, we have:*

$$F[W_{i+1}^{-1}/a, ..., W_n^{-1}/a, -W_{n+1}^{-1}/a] \circ f = -1/a \cdot \sigma \circ F[W_i, ..., W_1] \tag{58}$$

*Proof.* We prove this statement by induction on $i$ from $i = n$ backwards to $i = 1$.

**Case $i = n$:** Then, $F[W_{i+1}^{-1}/a, ..., -W_{n+1}^{-1}/a] = F[-W_{n+1}^{-1}/a] = -W_{n+1}^{-1}/a$. In addition,

$$
\begin{aligned}
F[-W_{n+1}^{-1}/a] \circ f &= -1/a \cdot \sigma \circ W_n \circ \sigma \circ W_{n-1} \circ \sigma \circ ... \circ \sigma \circ W_1 \\
&= -1/a \cdot \sigma \circ F[W_n, ..., W_1] \\
&= -1/a \cdot F[\mathrm{Id}, W_n, ..., W_1]
\end{aligned} \tag{59}
$$

**Induction hypothesis:** We assume that:

$$F[W_{i+1}^{-1}/a, ..., W_n^{-1}/a, -W_{n+1}^{-1}/a] \circ f = -1/a \cdot F[\mathrm{Id}, W_i, ..., W_1] \tag{60}$$

**Case $i - 1$:** We consider that by the induction hypothesis:

$$
\begin{aligned}
F[W_i^{-1}/a, ..., W_n^{-1}/a, -W_{n+1}^{-1}/a] \circ f &= W_i^{-1}/a \circ F[\mathrm{Id}, W_{i+1}^{-1}/a, ..., W_n^{-1}/a, -W_{n+1}^{-1}/a] \circ f \\
&= W_i^{-1}/a \circ \sigma \circ F[W_{i+1}^{-1}/a, ..., W_n^{-1}/a, -W_{n+1}^{-1}/a] \circ f \\
&= W_i^{-1}/a \circ \sigma \circ -1/a \circ \sigma \circ F[W_i, ..., W_1] \\
&= -W_i^{-1}/a \circ F[W_i, ..., W_1] \\
&= -1/a \cdot F[\mathrm{Id}, W_{i-1}, ..., W_1]
\end{aligned} \tag{61}
$$

Finally, we conclude that:

$$F[W_{i+1}^{-1}/a, ..., W_n^{-1}/a, -W_{n+1}^{-1}/a] \circ f = -1/a \cdot \sigma \circ F[W_i, ..., W_1] \tag{62}$$

$\square$

**Lemma 13.** *Let $\mathcal{N} = \mathrm{SCM}[\sigma]$ with $\sigma$ that is Leaky ReLU with parameter $0 < a \neq 1$. Then, for all $\epsilon_0 > 0$, we have $C_{B,A}^{\epsilon_0} = C_{A,B}^{\epsilon_0}$.*

*Proof.* Let $k \geq \left\{ E_{A,B}^{\epsilon_0}, E_{B,A}^{\epsilon_0} \right\}$ and $m \geq k + C_{B,A}^{\epsilon_0}$. We take $y \in H_{\epsilon_0}(A, B; m)$. Then, $C(y) = C_{A,B}^{\epsilon_0}$. In addition, $\mathrm{disc}_m(y \circ D_A, D_B) \leq \epsilon_0$. By the third part of Lem. 5, for $h :\leftarrow y$, we have:

$$\mathrm{disc}_k(y^{-1} \circ D_B, D_A) \leq \mathrm{disc}_m(y \circ D_A, D_B) \leq \epsilon_0 \tag{63}$$

In particular, $C_{B,A}^{\epsilon_0} \leq C(y^{-1})$. In addition, by Lem. 1, $C(y^{-1}) = C(y)$. Therefore, $C_{B,A}^{\epsilon_0} \leq C_{A,B}^{\epsilon_0}$. By symmetric arguments (switching between $A$ and $B$) we also have the opposite side and thus, $C_{B,A}^{\epsilon_0} = C_{A,B}^{\epsilon_0}$. $\square$

# D Proof of Thm. 1 and Its Generalization Thm. 2

## D.1 Covering Numbers

**Definition 10** (Set embedding). *Let $(\mathcal{U}, \sim_{\mathcal{U}})$ and $(\mathcal{V}, \sim_{\mathcal{V}})$ be two tuples of sets and symmetric and reflexive relations on them (resp.). A function $G : \mathcal{U} \to \mathcal{V}$ is an embedding of $(\mathcal{U}, \sim_{\mathcal{U}})$ in $(\mathcal{V}, \sim_{\mathcal{V}})$ and we denote $(\mathcal{U}, \sim_{\mathcal{U}}) \preceq (\mathcal{V}, \sim_{\mathcal{V}})$ if:*

$$\forall u_1, u_2 \in \mathcal{U} : G(u_1) \sim_{\mathcal{V}} G(u_2) \implies u_1 \sim_{\mathcal{U}} u_2 \tag{64}$$

**Lemma 14.** *Let $(\mathcal{U}, \sim_{\mathcal{U}})$ and $(\mathcal{V}, \sim_{\mathcal{V}})$ be two tuples of sets and reflexive and symmetric relations on them (resp.). If $(\mathcal{U}, \sim_{\mathcal{U}}) \preceq (\mathcal{V}, \sim_{\mathcal{V}})$ then $\mathrm{N}(\mathcal{U}, \sim_{\mathcal{U}}) \leq \mathrm{N}(\mathcal{V}, \sim_{\mathcal{V}})$.*

*Proof.* Assume that $(\mathcal{U}, \sim_{\mathcal{U}}) \preceq (\mathcal{V}, \sim_{\mathcal{V}})$. Then, by definition, there is an embedding function $G : \mathcal{U} \rightarrow \mathcal{V}$ such that:

$$\forall u_1, u_2 \in \mathcal{U} : G(u_1) \sim_{\mathcal{V}} G(u_2) \implies u_1 \sim_{\mathcal{U}} u_2 \tag{65}$$

Let $(\mathcal{V}, \equiv_{\mathcal{V}})$ be a covering of $(\mathcal{V}, \sim_{\mathcal{V}})$. We define a covering $(\mathcal{U}, \equiv_{\mathcal{U}})$ of $(\mathcal{U}, \sim_{\mathcal{U}})$ as follows:

$$u_1 \equiv_{\mathcal{U}} u_2 \iff G(u_1) \equiv_{\mathcal{V}} G(u_2) \tag{66}$$

**Part 1:** We would like to prove that $(\mathcal{U}, \equiv_{\mathcal{U}})$ is a covering of $(\mathcal{U}, \sim_{\mathcal{U}})$. It is easy to see that $\equiv_{\mathcal{U}}$ is an equivalence relation since $\equiv_{\mathcal{V}}$ is an equivalence relation. Next, we would like to prove that $u_1 \equiv_{\mathcal{U}} u_2 \implies u_1 \sim_{\mathcal{U}} u_2$. By the definition of $\equiv_{\mathcal{U}}$:

$$u_1 \equiv_{\mathcal{U}} u_2 \implies G(u_1) \equiv_{\mathcal{V}} G(u_2) \tag{67}$$

In addition, since $(\mathcal{V}, \equiv_{\mathcal{V}})$ is a covering of $(\mathcal{V}, \sim_{\mathcal{V}})$:

$$G(u_1) \equiv_{\mathcal{V}} G(u_2) \implies G(u_1) \sim_{\mathcal{V}} G(u_2) \tag{68}$$

Finally, since $G$ is an embedding:

$$G(u_1) \sim_{\mathcal{V}} G(u_2) \implies u_1 \sim_{\mathcal{U}} u_2 \tag{69}$$

We conclude:

$$u_1 \equiv_{\mathcal{U}} u_2 \implies u_1 \sim_{\mathcal{U}} u_2 \tag{70}$$

Therefore, $(\mathcal{U}, \equiv_{\mathcal{U}})$ is indeed a covering of $(\mathcal{U}, \sim_{\mathcal{U}})$.

**Part 2:** We would like to prove that $|\mathcal{U}/\equiv_{\mathcal{U}}| \leq |\mathcal{V}/\equiv_{\mathcal{V}}|$. Let $u_1, u_2 \in \mathcal{U}$ such that $u_1 \not\equiv_{\mathcal{U}} u_2$. Then, by definition of $\equiv_{\mathcal{U}}$ we have: $G(u_1) \not\equiv_{\mathcal{V}} G(u_2)$. Therefore, if we take $u_1, ..., u_n \in \mathcal{U}$ representations of $n$ different equivalence classes in $(\mathcal{U}, \equiv_{\mathcal{U}})$ then, $G(u_1), ..., G(u_n) \in \mathcal{V}$ are $n$ representations of $n$ different equivalence classes in $(\mathcal{V}, \equiv_{\mathcal{V}})$. In particular, $|\mathcal{U}/\equiv_{\mathcal{U}}| \leq |\mathcal{V}/\equiv_{\mathcal{V}}|$. Therefore, the covering number of $(\mathcal{U}, \sim_{\mathcal{U}})$ is at most the covering number of $(\mathcal{V}, \sim_{\mathcal{V}})$. □

**Lemma 15.** *Let* $(\mathcal{U}, \equiv_1)$ *and* $(\mathcal{U}, \equiv_2)$ *be two coverings of* $(\mathcal{U}, \sim_{\mathcal{U}})$. *Then,* $(\mathcal{U}^2, \equiv_1 \times \equiv_2)$ *is a covering of* $(\mathcal{U}^2, \sim_{\mathcal{U}}^2)$. *Where* $\mathcal{U}^2 = \mathcal{U} \times \mathcal{U}$ *and the relation* $\sim_{\mathcal{U}}^2$ *is defined as follows:*

$$(a, b) \sim_{\mathcal{U}}^2 (c, d) \iff a \sim_{\mathcal{U}} c \text{ and } b \sim_{\mathcal{U}} d \tag{71}$$

*and* $\equiv_1 \times \equiv_2$ *is defined as:*

$$(a, b) \equiv_1 \times \equiv_2 (c, d) \iff a \equiv_1 c \text{ and } b \equiv_2 d \tag{72}$$

*Proof.* We have to prove that $\equiv_1 \times \equiv_2$ is an equivalence relation and that $(u_1, u_2) \equiv_1 \times \equiv_2 (v_1, v_2) \implies (u_1, u_2) \sim_{\mathcal{U}}^2 (v_1, v_2)$.

**Reflexivity:**
$$(u_1, u_2) \equiv_1 \times \equiv_2 (u_1, u_2) \iff u_1 \equiv_1 u_1 \text{ and } u_2 \equiv_1 u_2 \tag{73}$$

The RHS is true since $\equiv_1$ and $\equiv_2$ are reflexive relations.

**Symmetry:**
$$(u_1, u_2) \equiv_1 \times \equiv_2 (v_1, v_2) \iff u_1 \equiv_1 v_1 \text{ and } u_2 \equiv_2 v_2 \tag{74}$$

Since $\equiv_1$ and $\equiv_2$ are symmetric, we have:

$$u_1 \equiv_1 v_1 \text{ and } u_2 \equiv_2 v_2 \iff v_1 \equiv_1 u_1 \text{ and } v_2 \equiv_2 u_2 \tag{75}$$

In addition,
$$(v_1, v_2) \equiv_1 \times \equiv_2 (u_1, u_2) \iff v_1 \equiv_1 u_1 \text{ and } v_2 \equiv_2 u_2 \tag{76}$$

Therefore,
$$(u_1, u_2) \equiv_1 \times \equiv_2 (v_1, v_2) \iff (v_1, v_2) \equiv_1 \times \equiv_2 (u_1, u_2) \tag{77}$$

**Transitivity:** follows from similar arguments.

**Covering:**
$$(u_1, u_2) \equiv_1 \times \equiv_2 (v_1, v_2) \iff u_1 \equiv_1 v_1 \text{ and } u_2 \equiv_2 v_2 \tag{78}$$

Since $(\mathcal{U}, \equiv_i)$ is a covering of $(\mathcal{U}, \sim_{\mathcal{U}})$, for $i = 1, 2$, we have:

$$u_1 \equiv_1 v_1 \text{ and } u_2 \equiv_2 v_2 \implies u_1 \sim_{\mathcal{U}} v_1 \text{ and } u_2 \sim_{\mathcal{U}} v_2 \tag{79}$$

By the definition of $\sim_{\mathcal{U}}^2$ we have:

$$u_1 \sim_{\mathcal{U}} v_1 \text{ and } u_2 \sim_{\mathcal{U}} v_2 \iff (u_1, u_2) \sim_{\mathcal{U}} (v_1, v_2) \tag{80}$$

Therefore,

$$(u_1, u_2) \equiv_1 \times \equiv_2 (v_1, v_2) \implies (u_1, u_2) \sim_{\mathcal{U}}^2 (v_1, v_2) \tag{81}$$

$\square$

**Lemma 16.** *Let $(\mathcal{U}, \sim_{\mathcal{U}})$ be a tuple of a set and a reflexive and symmetric relation on it (resp.). Then,*

$$\mathrm{N}(\mathcal{U}^2, \sim_{\mathcal{U}}^2) \leq \mathrm{N}(\mathcal{U}, \sim_{\mathcal{U}})^2 \tag{82}$$

*Proof.* Let $\equiv_{\mathcal{U}}$ be an equivalence relation such that $(\mathcal{U}, \equiv_{\mathcal{U}})$ is a covering of $(\mathcal{U}, \sim_{\mathcal{U}})$. By Lem. 15, $(\mathcal{U}^2, \equiv_{\mathcal{U}}^2)$ is a covering of $(\mathcal{U}^2, \sim_{\mathcal{U}}^2)$. In addition,

$$|\mathcal{U}^2 / \equiv_{\mathcal{U}}^2 | = |\mathcal{U} / \equiv_{\mathcal{U}} |^2 \tag{83}$$

Thus, for every covering $(\mathcal{U}, \equiv_{\mathcal{U}})$ of $(\mathcal{U}, \sim_{\mathcal{U}})$, there is a covering of $(\mathcal{U}^2, \sim_{\mathcal{U}}^2)$ of size $|\mathcal{U} / \equiv_{\mathcal{U}} |^2$. In particular, $\mathrm{N}(\mathcal{U}^2, \sim_{\mathcal{U}}^2) \leq \mathrm{N}(\mathcal{U}, \sim_{\mathcal{U}})^2$. $\square$

**Lemma 17.** *Let $(\mathcal{U}, \sim_{\mathcal{U}})$ be a tuple of a set and a reflexive and symmetric relation on it (resp.). Then,*

$$\mathrm{N}(\mathcal{U}, \sim_{\mathcal{U}}) \leq \mathrm{N}(\mathcal{U}^2, \sim_{\mathcal{U}}^2) \tag{84}$$

*Proof.* We define an embedding from $(\mathcal{U}, \sim_{\mathcal{U}})$ to $(\mathcal{U}^2, \sim_{\mathcal{U}}^2)$ as follows $F(u) = (u, u)$. This is an embedding, because, $F(u) \sim_{\mathcal{U}}^2 F(v) \implies (u, u) \sim_{\mathcal{U}}^2 (v, v) \implies u \sim_{\mathcal{U}} v$. $\square$

**Lemma 18.** *Let $(\mathcal{U}, \sim_{\mathcal{U}})$ and $(\mathcal{V}, \sim_{\mathcal{V}})$ be two tuples of sets and reflexive and symmetric relations on them (resp.). Assume that $\mathcal{U} \subset \mathcal{V}$ and $\sim_{\mathcal{U}} := (\sim_{\mathcal{V}})\big|_{\mathcal{U}}$, i.e,*

$$\forall u, v \in \mathcal{U} : u \sim_{\mathcal{U}} v \iff u \sim_{\mathcal{V}} v \tag{85}$$

*Then,*

$$\mathrm{N}(\mathcal{U}, \sim_{\mathcal{U}}) \leq \mathrm{N}(\mathcal{V}, \sim_{\mathcal{V}}) \tag{86}$$

*Proof.* Let $(\mathcal{V}, \equiv_{\mathcal{V}})$ be a covering of $(\mathcal{V}, \sim_{\mathcal{V}})$. Then, it is easy to see that $(\mathcal{U}, \equiv_{\mathcal{U}})$ is a covering of $(\mathcal{U}, \sim_{\mathcal{U}})$, where $\equiv_{\mathcal{U}} := (\equiv_{\mathcal{V}})\big|_{\mathcal{U}}$. In addition, we have: $|\mathcal{U} / \equiv_{\mathcal{U}} | \leq |\mathcal{V} / \equiv_{\mathcal{V}} |$. Thus, for every covering of $(\mathcal{V}, \sim_{\mathcal{V}})$, we can find a smaller covering for $(\mathcal{U}, \sim_{\mathcal{U}})$. In particular, $\mathrm{N}(\mathcal{U}, \sim_{\mathcal{U}}) \leq \mathrm{N}(\mathcal{V}, \sim_{\mathcal{V}})$. $\square$

## D.2 PERTURBATIONS AND DISCREPANCY

Thm. 1 employs assumption 3. In Lem. 19 we prove that this assumption holds for the case of a continuous risk (assumption 4) if the discriminators have bounded weights.

**Assumption 3.** *Let $\mathcal{N} = \mathrm{SCM}[\sigma]$ with $\sigma$ that is Leaky ReLU with parameter $0 < a \neq 1$. For every $m > 0$ (possibly $\infty$) and $n > 0$, the function $\mathrm{disc}_m(F[W_n, ..., W_1] \circ D_1, D_2)$ is continuous as a function of the weights of $W_1, ..., W_n \in \mathbb{R}^{M \times M}$.*

**Assumption 4.** *Let $\mathcal{N} = \mathrm{SCM}[\sigma]$ with $\sigma$ that is Leaky ReLU with parameter $0 < a \neq 1$. For all $m > 0$, the function $R_D[F[V_m, ..., V_1], F[W_m, ..., W_1]]$ is continuous as a function of $V_m, ..., V_1, W_m, ..., W_1$.*

**Lemma 19.** *Let $\mathcal{N} = \mathrm{SCM}[\sigma]$ with $\sigma$ that is Leaky ReLU with parameter $0 < a \neq 1$ and assume Assumption 4 for $D :\leftarrow D_1$. Let $\mathrm{disc}_{m,E} := \mathrm{disc}_{\mathcal{C}_{m,E}}$ for*

$$\mathcal{C}_{m,E} = \left\{ F[W_m, ..., W_1] \;\middle|\; \forall i \in [m] : ||W_i|| \leq E \right\} \tag{87}$$

*Then, for all $m > 0$, $n > 0$ and $E > 0$, the function $\mathrm{disc}_{m,E}(F[W_n, ..., W_1] \circ D_1, D_2)$ is continuous as a function of $W_n, ..., W_1$.*

*Proof.* Let $W_n, ..., W_1$ and $W_n^k, ..., W_1^k$ be any invertible matrices in $\mathbb{R}^{M \times M}$ such that for all $i \in [n]$, $W_i^k \to W_i$. We denote $G_E = \left\{ W \in \mathbb{R}^{M \times M} \mid ||W|| \leq E \right\}$. By the triangle inequality,

$$\text{disc}_{m,E}(D_1, D_2) \leq \text{disc}_{m,E}(D_1, D_3) + \text{disc}_{m,E}(D_3, D_2)$$
$$\implies \text{disc}_{m,E}(D_1, D_2) - \text{disc}_{m,E}(D_3, D_2) \leq \text{disc}_{m,E}(D_1, D_3) \tag{88}$$

Similarly,

$$\text{disc}_{m,E}(D_3, D_2) \leq \text{disc}_{m,E}(D_1, D_3) + \text{disc}_{m,E}(D_1, D_2)$$
$$\implies \text{disc}_{m,E}(D_3, D_2) - \text{disc}_{m,E}(D_1, D_2) \leq \text{disc}_{m,E}(D_1, D_3) \tag{89}$$

therefore,

$$\left| \text{disc}_{m,E}(D_3, D_2) - \text{disc}_{m,E}(D_1, D_2) \right| \leq \text{disc}_{m,E}(D_1, D_3) \tag{90}$$

In particular,

$$\left| \text{disc}_{m,E}(F[W_n, ..., W_1] \circ D_1, D_2) - \text{disc}_{m,E}(F[W_n^k, ..., W_1^k] \circ D_1, D_2) \right|$$

$$\leq \text{disc}_{m,E}(F[W_n, ..., W_1] \circ D_1, F[W_n^k, ..., W_1^k] \circ D_1)$$

$$\leq \sup_{c_1, c_2 \in \mathcal{C}_{m,E}} \left| R_{D_1}[c_1 \circ F[W_n, ..., W_1], c_2 \circ F[W_n, ..., W_1]] - R_{D_1}[c_1 \circ F[W_n^k, ..., W_1^k], c_2 \circ F[W_n^k, ..., W_1^k]] \right|$$

$$\leq \sup_{V_1, .., V_m, U_1, ..., U_m \in G_E} \left| R_{D_1}[F[V_m, ..., V_1] \circ F[W_n^k, ..., W_1^k], F[U_m, ..., U_1] \circ F[W_n, ..., W_1]] \right.$$

$$\left. - R_{D_1}[F[V_m, ..., V_1] \circ F[W_n^k, ..., W_1^k], F[U_m, ..., U_1] \circ F[W_n^k, ..., W_1^k]] \right|$$

$$\leq \sup_{V_1, .., V_m, U_1, ..., U_m \in G_E} \left| R_{D_1}\left[ F[V_m, ..., V_2, V_1 \cdot W_n, W_{n-1}, ..., W_1], F[U_m, ..., U_2, U_1 \cdot W_n, W_{n-1}, ..., W_1] \right] \right.$$

$$\left. - R_{D_1}\left[ F[V_m, ..., V_2, V_1 \cdot W_n^k, W_{n-1}^k, ..., W_1^k], F[U_m, ..., U_2, U_1 \cdot W_n^k, W_{n-1}^k, ..., W_1^k] \right] \right| \tag{91}$$

Assume by contradiction that the last expression does not converge to $0$. Therefore, there is a sequence $(V_1^k, ..., V_m^k, U_1^k, ..., U_m^k)$ such that $V_1^k, .., V_m^k, U_1^k, ..., U_m^k \in G_E$ and

$$Q_k = \left| R_{D_1}\left[ F[V_m^k, ..., V_2^k, V_1^k \cdot W_n, W_{n-1}, ..., W_1], F[U_m^k, ..., U_2^k, U_1^k \cdot W_n, W_{n-1}, ..., W_1] \right] \right.$$

$$\left. - R_{D_1}\left[ F[V_m^k, ..., V_2^k, V_1^k \cdot W_n^k, W_{n-1}^k, ..., W_1^k], F[U_m^k, ..., U_2^k, U_1^k \cdot W_n^k, W_{n-1}^k, ..., W_1^k] \right] \right| \not\to 0 \tag{92}$$

In particular, there is some $\epsilon > 0$ and an increasing sequence $\{k_j\}_{j=1}^{\infty} \subset \mathbb{N}$ such that $Q_{k_j} > \epsilon$ for all $j \in \mathbb{N}$. With no loss of generality, we can assume that $k_j = j$ (otherwise, we replace the original sequence with the new one). Since $(V_1^{k_j}, ..., V_m^{k_j}, U_1^{k_j}, ..., U_m^{k_j}) \in G_E^{2m}$ and $G_E^{2m}$ is compact in $\mathbb{R}^{M \times M \times 2m}$, by the Bolzano-Weierstrass theorem, it has a converging subsequence. With no loss of generality, we can assume that $(V_1^{k_j}, ..., V_m^{k_j}, U_1^{k_j}, ..., U_m^{k_j})$ converges (otherwise, we replace it with a converging sub-sequence):

$$(V_1^{k_j}, ..., V_m^{k_j}, U_1^{k_j}, ..., U_m^{k_j}) \to (V_1, ..., V_m, U_1, ..., U_m) \in G_E^{2m} \tag{93}$$

In particular,

$$(V_m^{k_j}, ..., V_2^{k_j}, V_1^{k_j} \cdot W_n^{k_j}, W_{n-1}^{k_j}, ..., W_1^{k_j}) \to (V_m, ..., V_2, V_1 \cdot W_n, W_{n-1}, ..., W_1)$$
$$(U_m^{k_j}, ..., U_2^{k_j}, U_1^{k_j} \cdot W_n^{k_j}, W_{n-1}^{k_j}, ..., W_1^{k_j}) \to (U_m, ..., U_2, U_1 \cdot W_n, W_{n-1}, ..., W_1) \tag{94}$$

By Assumption 4, the function $R_{D_1}\left[ F[X_{m+n}, ..., X_1], F[Y_{m+n}, ..., Y_1] \right]$ is continuous. Therefore,

$$\left| R_{D_1}\left[ F[V_m^{k_j}, ..., V_2^{k_j}, V_1^{k_j} \cdot W_n, W_{n-1}, ..., W_1], F[U_m^{k_j}, ..., U_2^{k_j}, U_1^{k_j} \cdot W_n, W_{n-1}, ..., W_1] \right] \right.$$

$$\left. - R_{D_1}\left[ F[V_m, ..., V_2, V_1 \cdot W_n, W_{n-1}, ..., W_1], F[U_m, ..., U_2, U_1 \cdot W_n, W_{n-1}, ..., W_1] \right] \right| \to 0 \tag{95}$$

and,

$$\left| R_{D_1}\left[ F[V_m, ..., V_2, V_1 \cdot W_n, W_{n-1}, ..., W_1], F[U_m, ..., U_2, U_1 \cdot W_n, W_{n-1}, ..., W_1] \right] \right.$$

$$\left. - R_{D_1}\left[ F[V_m^{k_j}, ..., V_2^{k_j}, V_1^{k_j} \cdot W_n^{k_j}, W_{n-1}^{k_j}, ..., W_1^{k_j}], F[U_m^{k_j}, ..., U_2^{k_j}, U_1^{k_j} \cdot W_n^{k_j}, W_{n-1}^{k_j}, ..., W_1^{k_j}] \right] \right| \to 0 \tag{96}$$

Therefore, by the triangle inequality,

$$
\begin{aligned}
Q_{k_j} = \Big| & R_{D_1}\left[F[V_m^{k_j}, ..., V_2^{k_j}, V_1^{k_j} \cdot W_n, W_{n-1}, ..., W_1], F[U_m^{k_j}, ..., U_2^{k_j}, U_1^{k_j} \cdot W_n, W_{n-1}, ..., W_1]\right] \\
& - R_{D_1}\left[F[V_m^{k_j}, ..., V_2^{k_j}, V_1^{k_j} \cdot W_n^{k_j}, W_{n-1}^{k_j}, ..., W_1^{k_j}], F[U_m^{k_j}, ...U_2^{k_j}, U_1^{k_j} \cdot W_n^{k_j}, W_{n-1}^{k_j}, ..., W_1^{k_j}]\right] \Big| \to 0
\end{aligned}
\tag{97}
$$

in contradiction. Thus, we conclude that:

$$
\lim_{k \to \infty} \Big| \mathrm{disc}_{m,E}(F[W_n, ..., W_1] \circ D_1, D_2) - \mathrm{disc}_{m,E}(F[W_n^k, ..., W_1^k] \circ D_1, D_2) \Big| = 0
\tag{98}
$$

$\square$

**Lemma 20.** *Let $\mathcal{N} = \mathrm{SCM}[\sigma]$ with $\sigma$ that is Leaky ReLU with parameter $0 < a \neq 1$. In addition, let $f = F[W_{n+1}, ..., W_1]$ and $g = F[V_{n+1}, ..., V_1]$ be two minimal decompositions. Assume Assumptions 2 and 3. Then, there are functions $\bar{f} = F[\bar{W}_{n+1}, ..., \bar{W}_1]$ and $\bar{g} = F[\bar{V}_{n+1}, ..., \bar{V}_1]$ such that:*

- $C(\bar{f} \circ g) = 2n$.

- $\forall j \in [n+1] : \mathrm{disc}_m(F[\bar{W}_j, ..., \bar{W}_1] \circ D, F[W_j, ..., W_1] \circ D) \leq \epsilon$.

- $\forall j \in [n+1] : \mathrm{disc}_m(F[\bar{V}_j, ..., \bar{V}_1] \circ D, F[V_j, ..., V_1] \circ D) \leq \epsilon$.

*Proof.* We consider that $f \circ g = F[W_{n+1}, ..., W_2, W_1 \cdot V_{n+1}, V_n, ..., V_1]$. By Assumption 2, for each $\delta > 0$, there are $\bar{f} = F[\bar{W}_{n+1}, ..., \bar{W}_1]$ and $\bar{g} = F[\bar{V}_{n+1}, ..., \bar{V}_1]$ such that $C(\bar{f} \circ \bar{g}) = 2n$ and for all $j \in [n+1]$: $||\bar{W}_j - W_j||, ||\bar{V}_j - V_j|| \leq \delta$. By Assumption. 3, for each $\epsilon > 0$, there is a small enough $\delta > 0$ such that: $\bar{f} = F[\bar{W}_{n+1}, ..., \bar{W}_1]$ and $\bar{g} = F[\bar{V}_{n+1}, ..., \bar{V}_1]$ such that for all $j \in [n+1]$: $||\bar{W}_j - W_j||, ||\bar{V}_j - V_j|| \leq \delta$, we have:

- $\forall j \in [n+1] : \mathrm{disc}_m(F[\bar{W}_j, ..., \bar{W}_1] \circ D, F[W_j, ..., W_1] \circ D) \leq \epsilon$.

- $\forall j \in [n+1] : \mathrm{disc}_m(F[\bar{V}_j, ..., \bar{V}_1] \circ D, F[V_j, ..., V_1] \circ D) \leq \epsilon$.

In particular, for any $\epsilon > 0$, there are functions $\bar{f} = F[\bar{W}_{n+1}, ..., \bar{W}_1]$ and $\bar{g} = F[\bar{V}_{n+1}, ..., \bar{V}_1]$ with the desired properties. $\square$

**Lemma 21.** *Let $\mathcal{N} = \mathrm{SCM}[\sigma]$ with $\sigma$ that is Leaky ReLU with parameter $0 < a \neq 1$. Assume Assumption 1. Let $f \underset{m,\epsilon}{\overset{D}{\sim}} g$. Then, for every minimal decomposition $f = F[W'_{n+1}, ..., W'_1]$ there is a minimal decomposition $g = F[V'_{n+1}, ..., V'_1]$ such that:*

$$
\forall i \in [n+1] : F[W'_i, ..., W'_1] \circ D \underset{m,\epsilon}{\sim} F[V'_i, ..., V'_1] \circ D
\tag{99}
$$

*Proof.* Since $f \underset{m,\epsilon}{\overset{D}{\sim}} g$ there are minimal decompositions $f = F[W_{n+1}, ..., W_1]$ and $g = F[V_{n+1}, ..., V_1]$ such that:

$$
\forall i \in [n+1] : F[W_i, ..., W_1] \circ D \underset{m,\epsilon}{\sim} F[V_i, ..., V_1] \circ D
\tag{100}
$$

By Assumption 1, $W'_1 = \tau_1 \circ W_1$, for all $i = 2, ..., n$: $W'_i = \tau_i \circ W_i \circ \tau_{i-1}^{-1}$ and $W'_{n+1} = W_{n+1} \circ \tau_n^{-1}$. Therefore, we define a minimal decomposition for $g$ as follows: $g = F[V'_{n+1}, ..., V'_1]$ such that $V'_1 = \tau_1 \circ V_1$, for all $i = 2, ..., n$: $V'_i = \tau_i \circ V_i \circ \tau_{i-1}^{-1}$ and $V'_{n+1} = V_{n+1} \circ \tau_n^{-1}$. This is a minimal decomposition of $g$, since each invariant function is an invertible linear mapping and commutes with $\sigma$. By Lem. 9 we have:

$$
\forall i \in [n] : F[W'_i, ..., W'_1] = \tau_i \circ F[W_i, ..., W_1] \text{ and } F[V'_i, ..., V'_1] = \tau_i \circ F[V_i, ..., V_1]
\tag{101}
$$

Therefore, by Lem. 4, since $C(\tau_i) = 0$, we have:

$$
\begin{aligned}
\forall i \in [n] : &\mathrm{disc}_m(F[W'_i, ..., W'_1] \circ D, F[V'_i, ..., V'_1] \circ D) \\
&\leq \mathrm{disc}_m(\tau_i \circ F[W_i, ..., W_1] \circ D, \tau_i \circ F[V_i, ..., V_1] \circ D) \leq \epsilon
\end{aligned}
\tag{102}
$$

Alternatively,

$$\forall i \in [n] : F[W'_i, ..., W'_1] \circ D \underset{m,\epsilon}{\sim} F[V'_i, ..., V'_1] \circ D \tag{103}$$

Since $F[W'_{n+1}, ..., W'_1] = f = F[W_{n+1}, ..., W_1]$ and $F[V'_{n+1}, ..., V'_1] = g = F[V_{n+1}, ..., V_1]$ we also have $F[W'_{n+1}, ..., W'_1] \circ D \underset{m,\epsilon}{\sim} F[V'_{n+1}, ..., V'_1] \circ D$. $\qquad\square$

**Lemma 22.** *Let $\mathcal{N} = \text{SCM}[\sigma]$ with $\sigma$ that is Leaky ReLU with parameter $0 < a \neq 1$. We have:*

$$f \underset{k,\epsilon_1}{\overset{D_A}{\not\sim}} f', \bar{f} \underset{k,\epsilon}{\overset{D_A}{\sim}} f \text{ and } \bar{f}' \underset{k,\epsilon}{\overset{D_A}{\sim}} f' \implies \bar{f} \underset{k,\epsilon_1-2\epsilon}{\overset{D_A}{\not\sim}} \bar{f}' \tag{104}$$

*Proof.* Assume by contradiction that $\bar{f} \underset{k,\epsilon_1-2\epsilon}{\overset{D_A}{\sim}} \bar{f}'$. Then, there are decompositions $\bar{f} = F[\bar{W}_{n+1}, ..., \bar{W}_1]$ and $\bar{f}' = F[\bar{W}'_{n+1}, ..., \bar{W}'_1]$ such that:

$$\forall j \in [n+1] : \text{disc}_k(F[\bar{W}_j, ..., \bar{W}_1] \circ D_A, F[\bar{W}'_j, ..., \bar{W}'_1] \circ D_A) \leq \epsilon_1 - 2\epsilon \tag{105}$$

By Lem. 21, since $\bar{f} \underset{k,\epsilon}{\overset{D_A}{\sim}} f$ and $\bar{f}' \underset{k,\epsilon}{\overset{D_A}{\sim}} f'$, there are minimal decompositions $f = F[W_{n+1}, ..., W_1]$ and $f' = F[W'_{n+1}, ..., W'_1]$ such that:

$$\begin{aligned} \forall j \in [n+1] : &\text{disc}_k(F[W_j, ..., W_1] \circ D_A, F[\bar{W}_j, ..., \bar{W}_1] \circ D_A) \leq \epsilon \\ &\text{disc}_k(F[W'_j, ..., W'_1] \circ D_A, F[\bar{W}'_j, ..., \bar{W}'_1] \circ D_A) \leq \epsilon \end{aligned} \tag{106}$$

Since $f \underset{k,\epsilon_1}{\overset{D_A}{\not\sim}} f'$, there is an index $i \in [n+1]$ such that:

$$\text{disc}_k(F[W_i, ..., W_1] \circ D_A, F[W'_i, ..., W'_1] \circ D_A) > \epsilon_1 \tag{107}$$

Therefore, by the triangle inequality, we arrive to a contradiction:

$$\begin{aligned} &\text{disc}_k(F[W_i, ..., W_1] \circ D_A, F[W'_i, ..., W'_1] \circ D_A) \\ \leq &\text{disc}_k(F[\bar{W}_i, ..., \bar{W}_1] \circ D_A, F[W'_i, ..., W'_1] \circ D_A) \\ &+ \text{disc}_k(F[\bar{W}_i, ..., \bar{W}_1] \circ D_A, F[W_i, ..., W_1] \circ D_A) \\ \leq &\text{disc}_k(F[\bar{W}_i, ..., \bar{W}_1] \circ D_A, F[\bar{W}'_i, ..., \bar{W}'_1] \circ D_A) \\ &+ \text{disc}_k(F[\bar{W}_i, ..., \bar{W}_1] \circ D_A, F[W_i, ..., W_1] \circ D_A) \\ &+ \text{disc}_k(F[\bar{W}'_i, ..., \bar{W}'_1] \circ D_A, F[W'_i, ..., W'_1] \circ D_A) \\ \leq &(\epsilon_1 - 2\epsilon) + \epsilon + \epsilon = \epsilon_1 \end{aligned} \tag{108}$$

$\qquad\square$

**Lemma 23.** *Let $\mathcal{N} = \text{SCM}[\sigma]$ with $\sigma$ that is a Leaky ReLU with parameter $0 < a \neq 1$. Let $A = (\mathcal{X}_A, D_A)$ and $B = (\mathcal{X}_B, D_B)$ are two domains. We have:*

$$\text{N}\left(H_{\epsilon_0}(A, B), \underset{\epsilon_1+2\epsilon_0}{\overset{D_A}{\sim}}\right) \leq \text{N}\left(H_{\epsilon_0}(B, A), \underset{\epsilon_1}{\overset{D_B}{\sim}}\right) \tag{109}$$

*Proof.* We would like to show that the function $G(h) = h^{-1}$ is an embedding of $\left(H_{\epsilon_0}(A, B), \underset{\epsilon_1+2\epsilon_0}{\overset{D_A}{\sim}}\right)$ into $\left(H_{\epsilon_0}(B, A), \underset{\epsilon_1}{\overset{D_B}{\sim}}\right)$. First, we consider that if $h \in H_{\epsilon_0}(A, B)$, then,

$$\text{disc}(G(h) \circ D_B, D_A) = \text{disc}(h^{-1} \circ D_A, D_B) \leq \text{disc}(h \circ D_A, D_B) \leq \epsilon_0 \tag{110}$$

and by Lem. 1 and Lem. 13, $C(G(h)) = C(h^{-1}) = C(h) = C^{\epsilon_0}_{A,B} = C^{\epsilon_0}_{B,A}$. Therefore, $G(h) \in H_{\epsilon_0}(B, A)$. Next, we would like to prove that for all $h_1, h_2 \in H_{\epsilon_0}(A, B)$: $G(h_1) \underset{\epsilon_1}{\overset{D_B}{\sim}} G(h_2) \implies h_1 \underset{\epsilon_1+2\epsilon_0}{\overset{D_A}{\sim}} h_2$.

Let $h_1, h_2 \in H_{\epsilon_0}(A, B)$ such that $G(h_1) \overset{D_B}{\underset{\epsilon_1}{\sim}} G(h_2)$. Then, there are minimal decompositions $G(h_1) = F[W_{n+1}, ..., W_1]$ and $G(h_2) = F[V_{n+1}, ..., V_1]$ such that:

$$\forall i \in [n] : \mathrm{disc}(F[W_i, ..., W_1] \circ D_A, F[V_i, ..., V_1] \circ D_A) \leq \epsilon_1$$
$$\text{and: } \mathrm{disc}(G(h_1) \circ D_A, G(h_2) \circ D_A) \leq \epsilon_1 \tag{111}$$

We consider that by Lem. 1, $G(h_1) = F[-W_1^{-1}, W_2^{-1}/a, ..., W_n^{-1}/a, -W_{n+1}^{-1}/a]$ and $G(h_2) = F[-V_1^{-1}, V_2^{-1}/a, ..., V_n^{-1}/a, -V_{n+1}^{-1}/a]$ are minimal decompositions. In addition, by Lem. 12, we have:

$$\forall i \in [n] : F[W_{i+1}^{-1}/a, ...W_n^{-1}/a, -W_{n+1}^{-1}/a] \circ G(h_1) = -1/a \cdot \sigma \circ F[W_i, ..., W_1]$$
$$F[V_{i+1}^{-1}/a, ...V_n^{-1}/a, -V_{n+1}^{-1}/a] \circ G(h_2) = -1/a \cdot \sigma \circ F[V_i, ..., V_1] \tag{112}$$

By the first item of Lem. 5, for $D_1 := D_B$, $D_2 := h_1 \circ D_A$, $D_3 := F[V_{i+1}^{-1}/a, ...V_n^{-1}/a, -V_{n+1}^{-1}/a] \circ D_B$ and $p := F[W_{i+1}^{-1}/a, ...W_n^{-1}/a, -W_{n+1}^{-1}/a]$,

$$\mathrm{disc}(F[W_{i+1}^{-1}/a, ...W_n^{-1}/a, -W_{n+1}^{-1}/a] \circ D_B, F[V_{i+1}^{-1}/a, ...V_n^{-1}/a, -V_{n+1}^{-1}/a] \circ D_B)$$
$$\leq \mathrm{disc}(F[W_{i+1}^{-1}/a, ...W_n^{-1}/a, -W_{n+1}^{-1}/a] \circ h_1 \circ D_A, F[V_{i+1}^{-1}/a, ...V_n^{-1}/a, -V_{n+1}^{-1}/a] \circ D_B)$$
$$+ \mathrm{disc}(h_1 \circ D_A, D_B) \tag{113}$$
$$\leq \mathrm{disc}(F[W_{i+1}^{-1}/a, ...W_n^{-1}/a, -W_{n+1}^{-1}/a] \circ h_1 \circ D_A, F[V_{i+1}^{-1}/a, ...V_n^{-1}/a, -V_{n+1}^{-1}/a] \circ D_B) + \epsilon_0$$

Similarly (by the first item of Lem. 5), we have:

$$\mathrm{disc}(F[W_{i+1}^{-1}/a, ...W_n^{-1}/a, -W_{n+1}^{-1}/a] \circ D_B, F[V_{i+1}^{-1}/a, ...V_n^{-1}/a, -V_{n+1}^{-1}/a] \circ D_B)$$
$$\leq \mathrm{disc}(F[W_{i+1}^{-1}/a, ...W_n^{-1}/a, -W_{n+1}^{-1}/a] \circ h_1 \circ D_A, F[V_{i+1}^{-1}/a, ...V_n^{-1}/a, -V_{n+1}^{-1}/a] \circ h_2 \circ D_A) + 2\epsilon_0$$
$$= \mathrm{disc}(-1/a \cdot \sigma \circ F[W_i, ..., W_1] \circ D_A, -1/a \cdot \sigma \circ F[V_i, ..., V_1] \circ D_A) + 2\epsilon_0$$
$$\leq \mathrm{disc}(F[W_i, ..., W_1] \circ D_A, F[V_i, ..., V_1] \circ D_A) + 2\epsilon_0 \leq \epsilon_1 + 2\epsilon_0 \tag{114}$$

Therefore, we conclude that $h_1 \overset{D_A}{\underset{\epsilon_1 + 2\epsilon_0}{\sim}} h_2$. $\qquad \square$

## D.3 PROOF OF THM. 1

**Theorem 2.** *Let $\mathcal{N} = \mathrm{SCM}[\sigma]$ with $\sigma$ that is a Leaky ReLU with parameter $0 < a \neq 1$. Assume Assumptions 1, 2 and 3. Let $\epsilon_0$, $\epsilon_1$ and $\epsilon_2$ such that $\epsilon_0 < \epsilon_1/2$ and $\epsilon_2 < \epsilon_1 - 2\epsilon_0$ be three positive constants and $A = (\mathcal{X}_A, D_A)$ and $B = (\mathcal{X}_B, D_B)$ are two domains. Assume that $m \geq k + 2C_{A,B}^{\epsilon_0} + 2$. Then,*

$$\mathrm{N}\left(H_{\epsilon_0}(A, B; m), \overset{D_A}{\underset{k,\epsilon_1}{\sim}}\right) \leq \lim_{\epsilon \to 0} \mathrm{N}\left(\mathrm{DPM}_{2\epsilon_0 + \epsilon}\left(B; k, 2C_{A,B}^{\epsilon_0}\right), \overset{D_B}{\underset{m,\epsilon_2}{\sim}}\right) \tag{115}$$

*Proof.* Let $\epsilon$ be any positive constant such that: $\epsilon < \min\{(\epsilon_1 - 2\epsilon_0 - \epsilon_2)/4, \epsilon_2/2\}$. For such $\epsilon$, we have $2\epsilon_0 \leq \epsilon_1 - 4\epsilon$ and $\epsilon_2 \leq \epsilon_1 - 2\epsilon_0 - 4\epsilon$. In addition, let $t := k + C_{A,B}^{\epsilon_0} + 1$. We would like to find an embedding mapping:

$$G : (H_{\epsilon_0}(A, B; m))^2 \to \mathrm{DPM}_{2\epsilon_0 + \epsilon}\left(B; k, 2C_{A,B}^{\epsilon_0} + 2\right) \tag{116}$$

**Part 1:** In this part, we show how to construct $G$. Let $(f, g) \in (H_{\epsilon_0}(A, B; m))^2$. We denote: $f = F[W_{n+1}, ..., W_1]$ and $g = F[V_{n+1}, ..., V_1]$ minimal decompositions of $f$ and $g$ (resp.). By Lem. 20, there are functions $\bar{f} = F[\bar{W}_{n+1}, ..., \bar{W}_1]$ and $\bar{g} = F[\bar{V}_{n+1}, ..., \bar{V}_1]$ such that:

- $C(\bar{f} \circ \bar{g}^{-1}) = 2n$.

- $\forall j \in [n+1] : \mathrm{disc}_m(F[\bar{W}_j, ..., \bar{W}_1] \circ D_A, F[W_j, ..., W_1] \circ D_A) \leq \epsilon$.

- $\forall j \in [n+1] : \mathrm{disc}_m(F[\bar{V}_j, ..., \bar{V}_1] \circ D_A, F[V_j, ..., V_1] \circ D_A) \leq \epsilon$.

We define $G(f, g) = \bar{f} \circ \bar{g}^{-1}$.

**Part 2:** In this part, we show that:

$$(f, g) \in (H_{\epsilon_0}(A, B; m))^2 \implies G(f, g) \in \text{DPM}_{2\epsilon_0 + 2\epsilon}\left(D_B; k, 2C_{A,B}^{\epsilon_0}\right) \tag{117}$$

By Part 1, $C(\bar{f} \circ \bar{g}^{-1}) = 2n = 2C_{A,B}^{\epsilon_0}$. In addition, by the first item of Lem. 5, for $D_1 :\leftarrow \bar{g}^{-1} \circ D_B$, $D_2 :\leftarrow D_A$, $D_3 :\leftarrow D_B$, $p :\leftarrow \bar{f}$, $t \geq k + C_{A,B}^{\epsilon_0}$ we have:

$$\text{disc}_k(\bar{f} \circ \bar{g}^{-1} \circ D_B, D_B) \leq \text{disc}_t(\bar{f} \circ D_A, D_B) + \text{disc}_t(\bar{g}^{-1} \circ D_B, D_A) \tag{118}$$

Since $f \in H_{\epsilon_0}(A, B; m)$:

$$\text{disc}_t(\bar{f} \circ D_A, D_B) \leq \text{disc}_m(f \circ D_A, D_B) + \text{disc}_m(\bar{f} \circ D_A, f \circ D_A) \leq \epsilon_0 + \epsilon \tag{119}$$

In addition, by the third item of Lem. 5, for $h :\leftarrow \bar{g}$ and $m \geq t + C_{A,B}^{\epsilon_0} \geq t + C(\bar{g}^{-1})$, we have:

$$\begin{aligned}
\text{disc}_t(\bar{g}^{-1} \circ D_B, D_A) &\leq \text{disc}_m(\bar{g} \circ D_A, D_B) \\
&\leq \text{disc}_m(g \circ D_A, D_B) + \text{disc}_m(g \circ D_A, \bar{g} \circ D_A) \leq \epsilon_0 + \epsilon
\end{aligned} \tag{120}$$

Finally, $\text{disc}_k(\bar{f} \circ \bar{g}^{-1} \circ D_B, D_B) \leq 2\epsilon_0 + 2\epsilon$ and we conclude that:

$$G(f, g) \in \text{DPM}_{2\epsilon_0 + 2\epsilon}\left(B; k, 2C_{A,B}^{\epsilon_0}\right) \tag{121}$$

**Part 3:** In this part, we show that $G$ is an embedding. It requires showing that

$$G(f, g) \overset{D_B}{\underset{m, \epsilon_2}{\sim}} G(f', g') \implies (f, g) \overset{D_A}{\underset{k, \epsilon_1}{\left(\sim\right)}}^2 (f', g') \tag{122}$$

Assume by contradiction that $G(f, g) \overset{D_B}{\underset{m, \epsilon_2}{\sim}} G(f', g')$ and that $(f, g) \overset{D_A}{\underset{k, \epsilon_1}{\not\sim}} (f', g')$. Then, we have

$$f \overset{D_A}{\underset{k, \epsilon_1}{\not\sim}} f' \text{ or } g \overset{D_A}{\underset{k, \epsilon_1}{\not\sim}} g' \tag{123}$$

We denote $G(f, g) = \bar{f} \circ \bar{g}^{-1}$ and $G(f', g') = \bar{f}' \circ (\bar{g}')^{-1}$ (see Part 1).

**Assume that** $f \overset{D_A}{\underset{k, \epsilon_1}{\not\sim}} f'$**:** By Lem. 22, $\bar{f} \overset{D_A}{\underset{k, \epsilon_1 - 2\epsilon}{\not\sim}} \bar{f}'$. In particular, for every two decompositions:

$$\bar{f} = F[\bar{W}_{n+1}, ..., \bar{W}_1] \text{ and } \bar{f}' = F[\bar{W}'_{n+1}, ..., \bar{W}'_1] \tag{124}$$

there is an index $i \in [n + 1]$ such that:

$$\text{disc}_k(F[\bar{W}_i, ..., \bar{W}_1] \circ D_A, F[\bar{W}'_i, ..., \bar{W}'_1] \circ D_A) > \epsilon_1 - 2\epsilon \tag{125}$$

The option $i = n + 1$ is not a possibility, since:

$$\begin{aligned}
\text{disc}_k(\bar{f} \circ D_A, \bar{f}' \circ D_A) \leq &\text{disc}_k(f \circ D_A, D_B) + \text{disc}_k(\bar{f} \circ D_A, f \circ D_A) \\
&+ \text{disc}_k(D_B, f' \circ D_A) + \text{disc}_k(\bar{f}' \circ D_A, f' \circ D_A) \\
\leq &2\epsilon_0 + 2\epsilon \leq \epsilon_1 - 2\epsilon
\end{aligned} \tag{126}$$

By the first item of Lem. 5, for $D_1 :\leftarrow D_A$, $D_2 :\leftarrow \bar{g}^{-1} \circ D_B$, $D_3 :\leftarrow F[\bar{W}'_i, ..., \bar{W}'_1] \circ D_A$, $p :\leftarrow F[\bar{W}_i, ..., \bar{W}_1]$ and $t \geq k + C_{A,B}^{\epsilon_0} \geq k + C(F[\bar{W}_i, ..., \bar{W}_1])$, we have:

$$\begin{aligned}
&\text{disc}_k(F[\bar{W}_i, ..., \bar{W}_1] \circ D_A, F[\bar{W}'_i, ..., \bar{W}'_1] \circ D_A) \\
&\leq \text{disc}_t(F[\bar{W}_i, ..., \bar{W}_1] \circ \bar{g}^{-1} \circ D_B, F[\bar{W}'_i, ..., \bar{W}'_1] \circ D_A) + \text{disc}_t(\bar{g}^{-1} \circ D_B, D_A) \\
&\leq \text{disc}_t(F[\bar{W}_i, ..., \bar{W}_1] \circ \bar{g}^{-1} \circ D_B, F[\bar{W}'_i, ..., \bar{W}'_1] \circ D_A) + \epsilon_0
\end{aligned} \tag{127}$$

Again, by the first item of Lem. 5, for $D_1 :\leftarrow D_A$, $D_2 :\leftarrow (g')^{-1} \circ D_B$, $D_3 :\leftarrow F[\bar{W}_i, ..., \bar{W}_1] \circ g^{-1} \circ D_B$, $p :\leftarrow F[\bar{W}'_i, ..., \bar{W}'_1]$ and $m \geq t + C_{A,B}^{\epsilon_0} \geq t + C(F[\bar{W}'_i, ..., \bar{W}'_1])$, we have:

$$\begin{aligned}
&\text{disc}_t(F[\bar{W}_i, ..., \bar{W}_1] \circ \bar{g}^{-1} \circ D_B, F[\bar{W}'_i, ..., \bar{W}'_1] \circ D_A) \\
&\leq \text{disc}_m(F[\bar{W}_i, ..., \bar{W}_1] \circ \bar{g}^{-1} \circ D_B, F[\bar{W}'_i, ..., \bar{W}'_1] \circ (\bar{g}')^{-1} \circ D_B) + \text{disc}_m((\bar{g}')^{-1} \circ D_B, D_A) \\
&\leq \text{disc}_m(F[\bar{W}_i, ..., \bar{W}_1] \circ g^{-1} \circ D_B, F[\bar{W}'_i, ..., \bar{W}'_1] \circ (g')^{-1} \circ D_B) + \epsilon_0
\end{aligned} \tag{128}$$

Therefore, we conclude that:

$$\epsilon_1 - 2\epsilon_0 - 2\epsilon < \mathrm{disc}_m(F[\bar{W}_i, ..., \bar{W}_1] \circ \bar{g}^{-1} \circ D_B, F[\bar{W}'_i, ..., \bar{W}'_1] \circ (\bar{g}')^{-1} \circ D_B) \qquad (129)$$

Alternatively, for any minimal decompositions $\bar{f} \circ \bar{g}^{-1} = F[\bar{W}_{n+1}, ..., \bar{W}_1] \circ \bar{g}^{-1}$ and $\bar{f}' \circ (\bar{g}')^{-1} = F[\bar{W}'_{n+1}, ..., \bar{W}'_1] \circ (\bar{g}')^{-1}$ there are right partial functions $F[\bar{W}_i, ..., \bar{W}_1] \circ \bar{g}^{-1}$ and $F[\bar{W}'_i, ..., \bar{W}'_1] \circ (\bar{g}')^{-1}$ such that:

$$\epsilon_1 - 2\epsilon_0 - 2\epsilon < \mathrm{disc}_m(F[\bar{W}_{n+1}, ..., \bar{W}_1] \circ \bar{g}^{-1} \circ D_B, F[\bar{W}'_i, ..., \bar{W}'_1] \circ (\bar{g}')^{-1} \circ D_B) \qquad (130)$$

in contradiction to $F(f, g) \overset{D_B}{\underset{m, \epsilon_2}{\sim}} F(f', g')$.

**Assume that** $g \overset{D_A}{\underset{k, \epsilon_1}{\not\sim}} g'$**:** By Lem. 22, $\bar{g} \overset{D_A}{\underset{k, \epsilon_1 - 2\epsilon}{\not\sim}} \bar{g}'$. Let

$$\bar{g}^{-1} = F[-\bar{V}_1, \bar{V}_2^{-1}/a, ..., \bar{V}_n^{-1}/a, -\bar{V}_{n+1}^{-1}/a]$$
$$\text{and } (\bar{g}')^{-1} = F[-(\bar{V}'_1)^{-1}, (\bar{V}'_2)^{-1}/a, ..., (\bar{V}'_n)^{-1}/a, -(\bar{V}'_{n+1})^{-1}/a] \qquad (131)$$

be any two minimal decompositions of $\bar{g}^{-1}$ and $(\bar{g}')^{-1}$ (resp.). Then, by Lem. 12, there are minimal decompositions $\bar{g} = F[\bar{V}_{n+1}, ..., \bar{V}_1]$ and $\bar{g}' = F[\bar{V}'_{n+1}, ..., \bar{V}'_1]$ such that:

$$\forall j \in [n] : F[\bar{V}_{j+1}^{-1}/a, ..., \bar{V}_n^{-1}/a, -\bar{V}_{n+1}^{-1}/a] \circ \bar{g} \circ D_A = -1/a \cdot \sigma \circ F[\bar{V}_j, ..., \bar{V}_1] \circ D_A$$
$$\text{and: } F[(\bar{V}'_{j+1})^{-1}/a, ..., (\bar{V}'_n)^{-1}/a, -(\bar{V}'_{n+1})^{-1}/a] \circ \bar{g}' \circ D_A = -1/a \cdot \sigma \circ F[\bar{V}'_j, ..., \bar{V}'_1] \circ D_A \qquad (132)$$

Since $\bar{g} \overset{D_A}{\underset{k, \epsilon_1 - 2\epsilon}{\not\sim}} \bar{g}'$, there is an index $i \in [n+1]$ such that:

$$\mathrm{disc}_k(F[\bar{V}_i, ..., \bar{V}_1] \circ D_A, F[\bar{V}'_i, ..., \bar{V}'_1] \circ D_A) > \epsilon_1 - 2\epsilon \qquad (133)$$

The case $i = n + 1$ is not a possibility, similarly to Eq. 126. Therefore, there is $i \in [n]$ such that Eq. 133 holds. In addition,

$$\mathrm{disc}_{k+1}(-1/a \cdot \sigma \circ F[\bar{V}_i, ..., \bar{V}_1] \circ D_A, -1/a \cdot \sigma \circ F[\bar{V}'_i, ..., \bar{V}'_1] \circ D_A)$$
$$= \mathrm{disc}_{k+1}\Big( F[\bar{V}_{i+1}^{-1}/a, ..., \bar{V}_n^{-1}/a, -\bar{V}_{n+1}^{-1}/a] \circ \bar{g} \circ D_A, $$
$$F[(\bar{V}'_{i+1})^{-1}/a, ..., (\bar{V}'_n)^{-1}/a, -(\bar{V}'_{n+1})^{-1}/a] \circ \bar{g}' \circ D_A \Big) \qquad (134)$$

By Lem. 4, for $p :\leftarrow -1/a \cdot \sigma$ of complexity 1 we have:

$$\epsilon_1 - 2\epsilon < \mathrm{disc}_k(F[\bar{V}_i, ..., \bar{V}_1] \circ D_A, F[\bar{V}'_i, ..., \bar{V}'_1] \circ D_A)$$
$$\leq \mathrm{disc}_{k+1}(-1/a \cdot \sigma \circ F[\bar{V}_i, ..., \bar{V}_1] \circ D_A, -1/a \cdot \sigma \circ F[\bar{V}'_i, ..., \bar{V}'_1] \circ D_A) \qquad (135)$$

In addition, by Lem. 5, for $D_1 :\leftarrow \bar{g} \circ D_A$, $D_2 :\leftarrow D_B$, $D_3 :\leftarrow F[(\bar{V}'_{i+1})^{-1}/a, ..., (\bar{V}'_n)^{-1}/a, -(\bar{V}'_{n+1})^{-1}/a] \circ \bar{g}' \circ D_A$, $t \geq (k+1) + C_{A,B}^{\epsilon_0} \geq (k+1) + C(F[\bar{V}_{i+1}^{-1}/a, ..., \bar{V}_n^{-1}/a, -\bar{V}_{n+1}^{-1}/a])$, we have:

$$\mathrm{disc}_{k+1}\Big( F[\bar{V}_{i+1}^{-1}/a, ..., \bar{V}_n^{-1}/a, -\bar{V}_{n+1}^{-1}/a] \circ \bar{g} \circ D_A,$$
$$F[(\bar{V}'_{i+1})^{-1}/a, ..., (\bar{V}'_n)^{-1}/a, -(\bar{V}'_{n+1})^{-1}/a] \circ \bar{g}' \circ D_A \Big)$$
$$\leq \mathrm{disc}_t\Big( F[\bar{V}_{i+1}^{-1}/a, ..., \bar{V}_n^{-1}/a, -\bar{V}_{n+1}^{-1}/a] \circ D_B,$$
$$F[(\bar{V}'_{i+1})^{-1}/a, ..., (\bar{V}'_n)^{-1}/a, -(\bar{V}'_{n+1})^{-1}/a] \circ \bar{g}' \circ D_A \Big) + \mathrm{disc}_t(\bar{g} \circ D_A, D_B) \qquad (136)$$
$$\leq \mathrm{disc}_t\Big( F[\bar{V}_{i+1}^{-1}/a, ..., \bar{V}_n^{-1}/a, -\bar{V}_{n+1}^{-1}/a] \circ D_B,$$
$$F[(V'_{i+1})^{-1}/a, ..., (\bar{V}'_n)^{-1}/a, -(\bar{V}'_{n+1})^{-1}/a] \circ \bar{g}' \circ D_A \Big) + \epsilon_0 + \epsilon$$

Again, by Lem. 5, for $D_1 :\leftarrow \bar{g}' \circ D_A$, $D_2 :\leftarrow D_B$, $D_3 :\leftarrow F[\bar{V}_{i+1}^{-1}/a, ..., \bar{V}_n^{-1}/a, -\bar{V}_{n+1}^{-1}/a] \circ D_B$, $m \geq t + C_{A,B}^{\epsilon_0} \geq t + C(F[(\bar{V}_{i+1}')^{-1}/a, ..., (\bar{V}_n')^{-1}/a, -(\bar{V}_{n+1}')^{-1}/a])$, we have:

$$
\begin{aligned}
\mathrm{disc}_t \bigg( & F[\bar{V}_{i+1}^{-1}/a, ..., \bar{V}_n^{-1}/a, -\bar{V}_{n+1}^{-1}/a] \circ D_B, \\
& F[(\bar{V}_{i+1}')^{-1}/a, ..., (\bar{V}_n')^{-1}/a, -(\bar{V}_{n+1}')^{-1}/a] \circ \bar{g}' \circ D_A \bigg) \\
\leq \mathrm{disc}_m \bigg( & F[\bar{V}_{i+1}^{-1}/a, ..., \bar{V}_n^{-1}/a, -\bar{V}_{n+1}^{-1}/a] \circ D_B, \\
& F[(\bar{V}_{i+1}')^{-1}/a, ..., (\bar{V}_n')^{-1}/a, -(\bar{V}_{n+1}')^{-1}/a] \circ D_B \bigg) + \mathrm{disc}_m(\bar{g}' \circ D_A, D_B) \\
\leq \mathrm{disc}_m \bigg( & F[\bar{V}_{i+1}^{-1}/a, ..., \bar{V}_n^{-1}/a, -\bar{V}_{n+1}^{-1}/a] \circ D_B, \\
& F[(\bar{V}_{i+1}')^{-1}/a, ..., (\bar{V}_n')^{-1}/a, -(\bar{V}_{n+1}')^{-1}/a] \circ D_B \bigg) + \epsilon_0 + \epsilon
\end{aligned}
\tag{137}
$$

Finally,

$$
\begin{aligned}
\epsilon_1 - 2\epsilon < \mathrm{disc}_k(&-1/a \cdot \sigma \circ F[\bar{V}_i, ..., \bar{V}_1] \circ D_A, -1/a \cdot \sigma \circ F[\bar{V}_i', ..., \bar{V}_1'] \circ D_A) \\
\leq \mathrm{disc}_t \bigg( & F[\bar{V}_{i+1}^{-1}/a, ..., \bar{V}_n^{-1}/a, -\bar{V}_{n+1}^{-1}/a] \circ D_B, \\
& F[(\bar{V}_{i+1}')^{-1}/a, ..., (\bar{V}_n')^{-1}/a, -(\bar{V}_{n+1}')^{-1}/a] \circ \bar{g}' \circ D_A \bigg) + \epsilon_0 + \epsilon \\
\leq \mathrm{disc}_m \bigg( & F[\bar{V}_{i+1}^{-1}/a, ..., \bar{V}_n^{-1}/a, -\bar{V}_{n+1}^{-1}/a] \circ D_B, \\
& F[(\bar{V}_{i+1}')^{-1}/a, ..., (\bar{V}_n')^{-1}/a, -(\bar{V}_{n+1}')^{-1}/a] \circ D_B \bigg) + 2\epsilon_0 + 2\epsilon
\end{aligned}
\tag{138}
$$

In particular,

$$
\begin{aligned}
\epsilon_2 \leq \epsilon_1 - 2\epsilon_0 - 4\epsilon < \mathrm{disc}_m \bigg( & F[\bar{V}_{i+1}^{-1}/a, ..., \bar{V}_n^{-1}/a, -\bar{V}_{n+1}^{-1}/a] \circ D_B, \\
& F[(\bar{V}_{i+1}')^{-1}/a, ..., (\bar{V}_n')^{-1}/a, -(\bar{V}_{n+1}')^{-1}/a] \circ D_B \bigg)
\end{aligned}
\tag{139}
$$

Alternatively, for any minimal decompositions

$$
\begin{aligned}
\bar{f} \circ \bar{g}^{-1} &= F[\bar{W}_{n+1}, ..., \bar{W}_2, -\bar{W}_1 \cdot \bar{V}_1, \bar{V}_2^{-1}/a, ..., \bar{V}_n^{-1}/a, -\bar{V}_{n+1}^{-1}/a] \\
\text{and } \bar{f}' \circ (\bar{g}')^{-1} &= F[\bar{W}_{n+1}, ..., \bar{W}_2, -\bar{W}_1 \cdot (\bar{V}_1')^{-1}, (\bar{V}_2')^{-1}/a, ..., (\bar{V}_n')^{-1}/a, -(\bar{V}_{n+1}')^{-1}/a]
\end{aligned}
\tag{140}
$$

there are right partial functions

$$
F[\bar{V}_{i+1}^{-1}/a, ..., \bar{V}_n^{-1}/a, -\bar{V}_{n+1}^{-1}/a] \text{ and } F[(\bar{V}_{i+1}')^{-1}/a, ..., (\bar{V}_n')^{-1}/a, -(\bar{V}_{n+1}')^{-1}/a]
\tag{141}
$$

such that Eq. 139 holds, in contradiction to $F(f, g) \overset{D_B}{\underset{m,\epsilon_2}{\sim}} F(f', g')$.

**Part 3:** Finally, by Lem. 17 and Lem. 14,

$$
\begin{aligned}
\mathrm{N}\left( H_{\epsilon_0}(A, B; m), \overset{D_A}{\underset{k,\epsilon_1}{\sim}} \right) &\leq \mathrm{N}\left( (H_{\epsilon_0}(A, B; m))^2, \left( \overset{D_A}{\underset{k,\epsilon_1}{\sim}} \right)^2 \right) \\
&\leq \mathrm{N}\left( \mathrm{DPM}_{2\epsilon_0 + 2\epsilon}\left( B; k, 2C_{A,B}^{\epsilon_0} \right), \overset{D_B}{\underset{m,\epsilon_2}{\sim}} \right)
\end{aligned}
\tag{142}
$$

Alternatively, for all $\epsilon_0, \epsilon_1, \epsilon_2, \epsilon$ such that $\epsilon < \min\{(\epsilon_1 - 2\epsilon_0 - \epsilon_2)/4, \epsilon_2/2\}$,

$$
\mathrm{N}\left( H_{\epsilon_0}(A, B; m), \overset{D_A}{\underset{k,\epsilon_1}{\sim}} \right) \leq \mathrm{N}\left( \mathrm{DPM}_{2\epsilon_0 + 2\epsilon}\left( B; k, 2C_{A,B}^{\epsilon_0} \right), \overset{D_B}{\underset{m,\epsilon_2}{\sim}} \right)
\tag{143}
$$

In particular, we can replace $\epsilon$ with $\epsilon/2$ in the inequality. By Lem. 18, the function $q_\epsilon = \mathrm{N}\left(\mathrm{DPM}_{2\epsilon_0+\epsilon}\left(B; k, 2C^{\epsilon_0}_{A,B}\right), \underset{m,\epsilon_2}{\overset{D_B}{\sim}}\right)$ is monotonically decreasing as $\epsilon$ tends to $0$ and is lower bounded by $\mathrm{N}\left(\mathrm{DPM}_{2\epsilon_0}\left(B; k, 2C^{\epsilon_0}_{A,B}\right), \underset{m,\epsilon_2}{\overset{D_B}{\sim}}\right)$. Therefore, by the monotone convergence theorem, the limit $\lim_{\epsilon\to 0} q_\epsilon$ exists and upper bounds $\mathrm{N}\left(H_{\epsilon_0}(A, B; m), \underset{k,\epsilon_1}{\overset{D_A}{\sim}}\right)$. $\qquad\square$

**Theorem 1.** *Let $\mathcal{N} = \mathrm{SCM}[\sigma]$ with $\sigma$ that is Leaky ReLU with parameter $0 < a \neq 1$. Assume Assumptions 1, 2 and 3. Let $\epsilon_0$, $\epsilon_1$ and $\epsilon_2$ be three constants such that $\epsilon_0 < \epsilon_1/4$ and $\epsilon_2 < \epsilon_1 - 4\epsilon_0$ be three positive constants and $A = (\mathcal{X}_A, D_A)$ and $B = (\mathcal{X}_B, D_B)$ are two domains. Then,*

$$\mathrm{N}\left(H_{\epsilon_0}(A, B), \underset{\epsilon_1}{\overset{D_A}{\sim}}\right) \leq \lim_{\epsilon\to 0} \min \begin{cases} \mathrm{N}\left(\mathrm{DPM}_{2\epsilon_0+\epsilon}\left(A; 2C^{\epsilon_0}_{A,B}\right), \underset{\epsilon_2}{\overset{D_A}{\sim}}\right) \\ \mathrm{N}\left(\mathrm{DPM}_{2\epsilon_0+\epsilon}\left(B; 2C^{\epsilon_0}_{A,B}\right), \underset{\epsilon_2}{\overset{D_B}{\sim}}\right) \end{cases} \tag{32}$$

*Proof.* By Lem. 2, with $m = k = \infty$, we have:

$$\mathrm{N}\left(H_{\epsilon_0}(A, B), \underset{\epsilon_1}{\overset{D_A}{\sim}}\right) \leq \lim_{\epsilon\to 0} \mathrm{N}\left(\mathrm{DPM}_{2\epsilon_0+\epsilon}\left(B; 2C^{\epsilon_0}_{A,B}\right), \underset{\epsilon_2}{\overset{D_B}{\sim}}\right) \tag{144}$$

Similarly,

$$\mathrm{N}\left(H_{\epsilon_0}(B, A), \underset{\epsilon_1-2\epsilon_0}{\overset{D_B}{\sim}}\right) \leq \lim_{\epsilon\to 0} \mathrm{N}\left(\mathrm{DPM}_{2\epsilon_0+\epsilon}\left(B; 2C^{\epsilon_0}_{A,B}\right), \underset{\epsilon_2}{\overset{D_B}{\sim}}\right) \tag{145}$$

By Lem. 23,

$$\mathrm{N}\left(H_{\epsilon_0}(A, B), \underset{\epsilon_1}{\overset{D_A}{\sim}}\right) \leq \mathrm{N}\left(H_{\epsilon_0}(B, A), \underset{\epsilon_1-2\epsilon_0}{\overset{D_B}{\sim}}\right) \tag{146}$$

Since the limits in the RHS of Eqs. 144 and 145 are limits of positive integers, we have:

$$\mathrm{N}\left(H_{\epsilon_0}(A, B), \underset{\epsilon_1}{\overset{D_A}{\sim}}\right) \leq \min \begin{cases} \lim_{\epsilon\to 0} \mathrm{N}\left(\mathrm{DPM}_{2\epsilon_0+\epsilon}\left(B; 2C^{\epsilon_0}_{A,B}\right), \underset{\epsilon_2}{\overset{D_B}{\sim}}\right) \\ \lim_{\epsilon\to 0} \mathrm{N}\left(\mathrm{DPM}_{2\epsilon_0+\epsilon}\left(A; 2C^{\epsilon_0}_{A,B}\right), \underset{\epsilon_2}{\overset{D_A}{\sim}}\right) \end{cases}$$
$$\leq \lim_{\epsilon\to 0} \min \begin{cases} \mathrm{N}\left(\mathrm{DPM}_{2\epsilon_0+\epsilon}\left(B; 2C^{\epsilon_0}_{A,B}\right), \underset{\epsilon_2}{\overset{D_B}{\sim}}\right) \\ \mathrm{N}\left(\mathrm{DPM}_{2\epsilon_0+\epsilon}\left(A; 2C^{\epsilon_0}_{A,B}\right), \underset{\epsilon_2}{\overset{D_A}{\sim}}\right) \end{cases} \tag{147}$$

$\qquad\square$

# E    WASSERSTEIN GAN RESULTS

It is interesting to check whether the predictions made are valid for other forms of discrepancy such as the one used in the Wasserstein GAN Arjovsky et al. (2017) (WGAN). This is done below for Prediction 2, which predicts that the selection of the right number of layers is crucial in unsupervised learning. In the WGAN experiment, we employ the architecture of (Kim et al., 2017) and vary the number of layers and inspect the influence on the results. For the generator, the architecture is identical while for WGAN's critic, the last sigmoid layer is removed. These experiments were done on the CelebA dataset, obtaining the results in Fig. 30– 35.

Note that since the encoder and the decoder parts of the learned network are symmetrical, the number of layers is always even. As can be seen, changing the number of layers has a dramatic effect on the results. The best overall results are obtained at 6 layers. Using fewer layers, WGAN often fails to produce images of the desired class. Adding layers, the semantic alignment is lost, as expected.

## F   CYCLEGAN RESULTS

While most of our experiments have focused on the DiscoGAN architecture of Kim et al. (2017), an additional experiment was conducted in order to verify that these extend to the CycleGAN architecture of Zhu et al. (2017).

The results are shown in Fig. 36. As can be seen running an experiment on the Aerial images to Maps dataset, we found that 8 layers produces an aligned solution. Using 10 layers produces unaligned map images with low discrepancy. For fewer than 8 layer, the discrepancy is high and the images are not very detailed.

Table 6: Numerical results for the experiment of Cityscapes to Image Segmentation. Standard metrics are used to evaluate the segmentation accuracy for different number of layers.

|  | $k = 2$ | $k = 4$ | $k = 6$ | $k = 8$ | $k = 10$ | $k = 12$ |
|---|---|---|---|---|---|---|
| Mean pixel accuracy | 0.52 | 0.54 | 0.53 | 0.60 | 0.63 | 0.51 |
| Mean class accuracy | 0.16 | 0.16 | 0.19 | 0.15 | 0.18 | 0.11 |
| Mean class IoU | 0.10 | 0.11 | 0.11 | 0.10 | 0.13 | 0.08 |

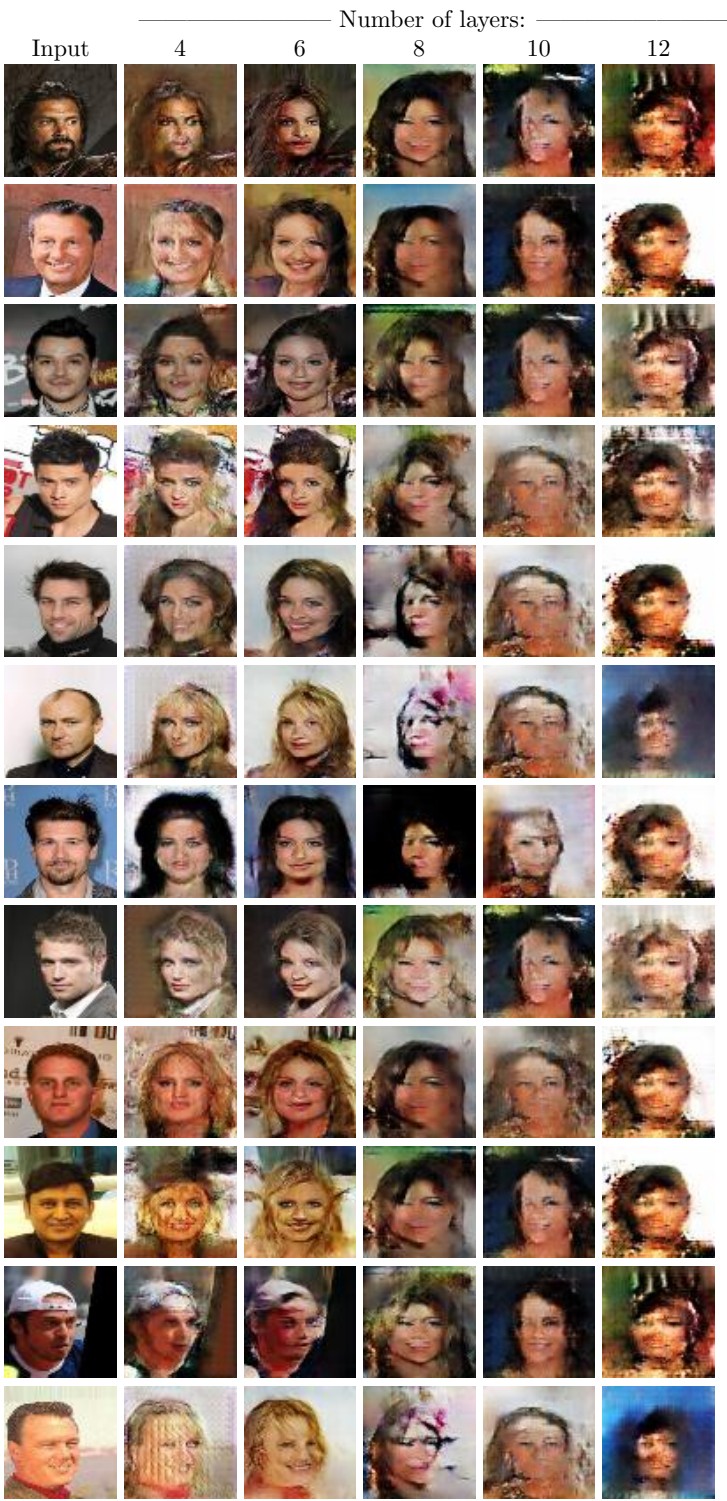

Figure 30: Results for celebA Male to Female transfer for WGAN with different number of layers.

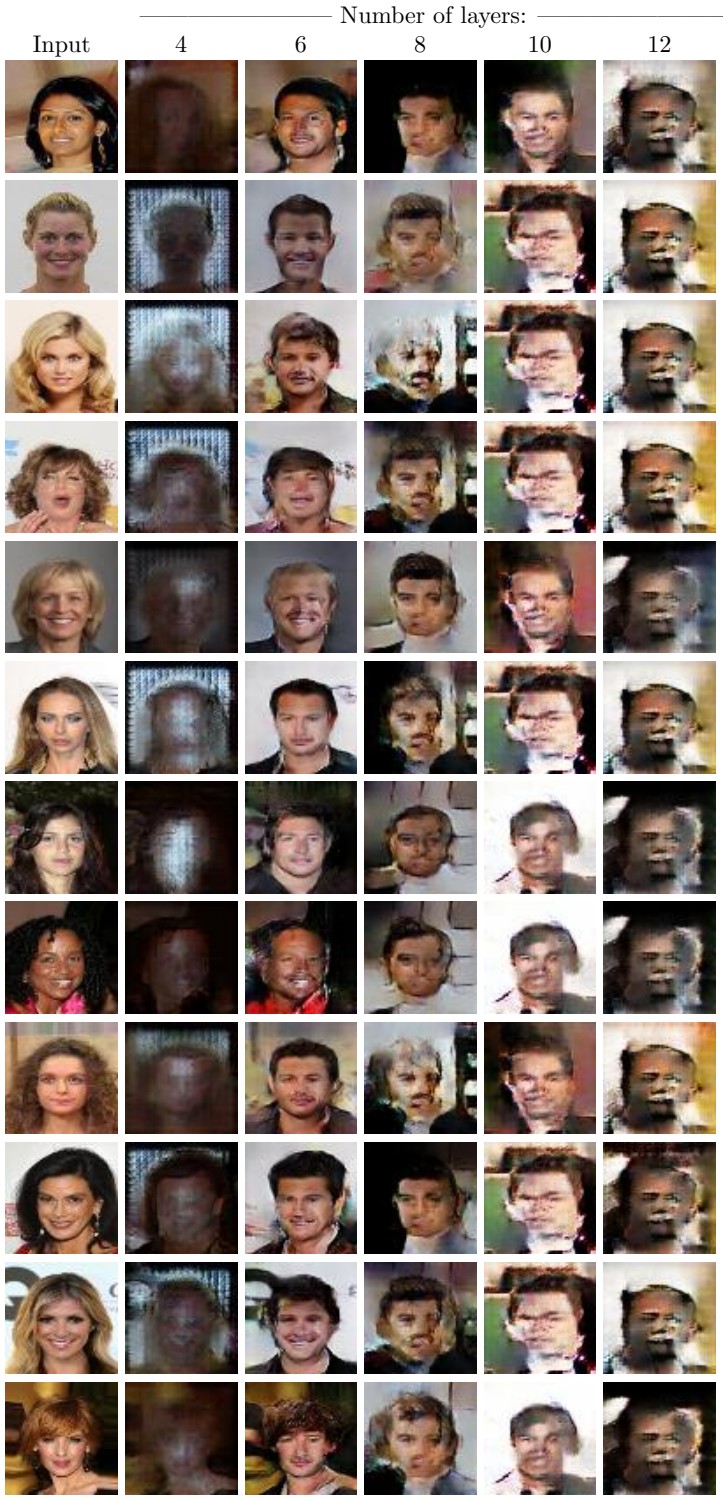

Figure 31: Results for celebA Female to Male transfer for WGAN with different number of layers.

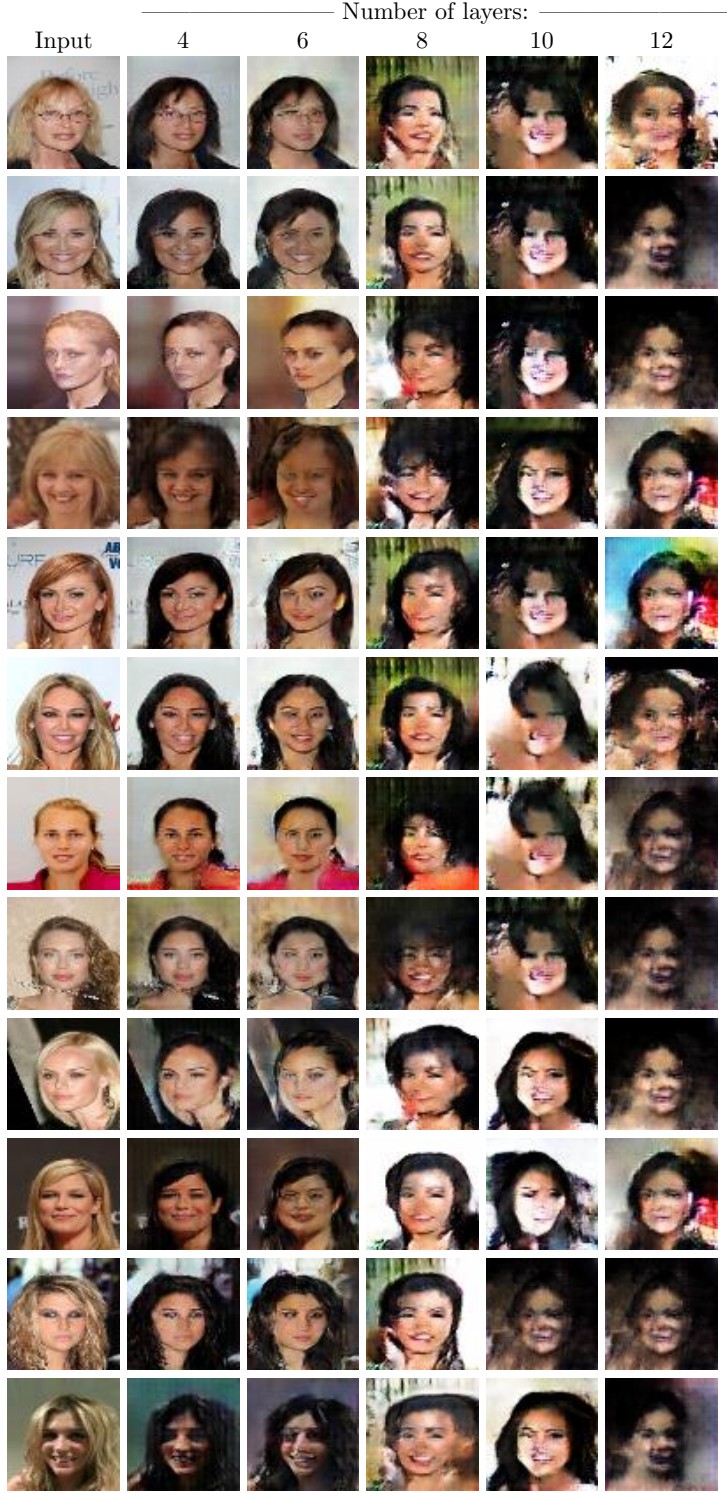

Figure 32: Results for celebA Blond to Black transfer for WGAN with different number of layers.

Number of layers:

Input    4    6    8    10    12

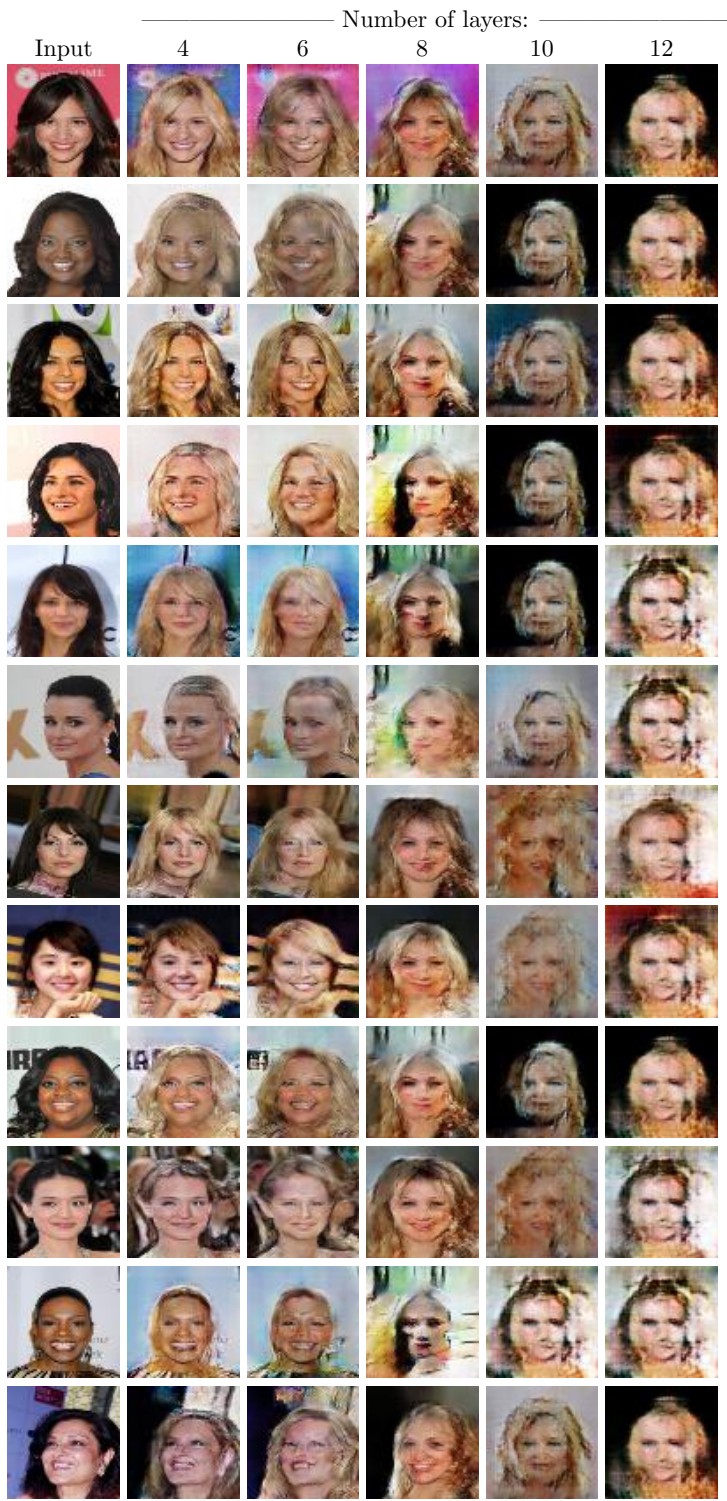

Figure 33: Results for celebA Black to Blond transfer for WGAN networks with different number of layers.

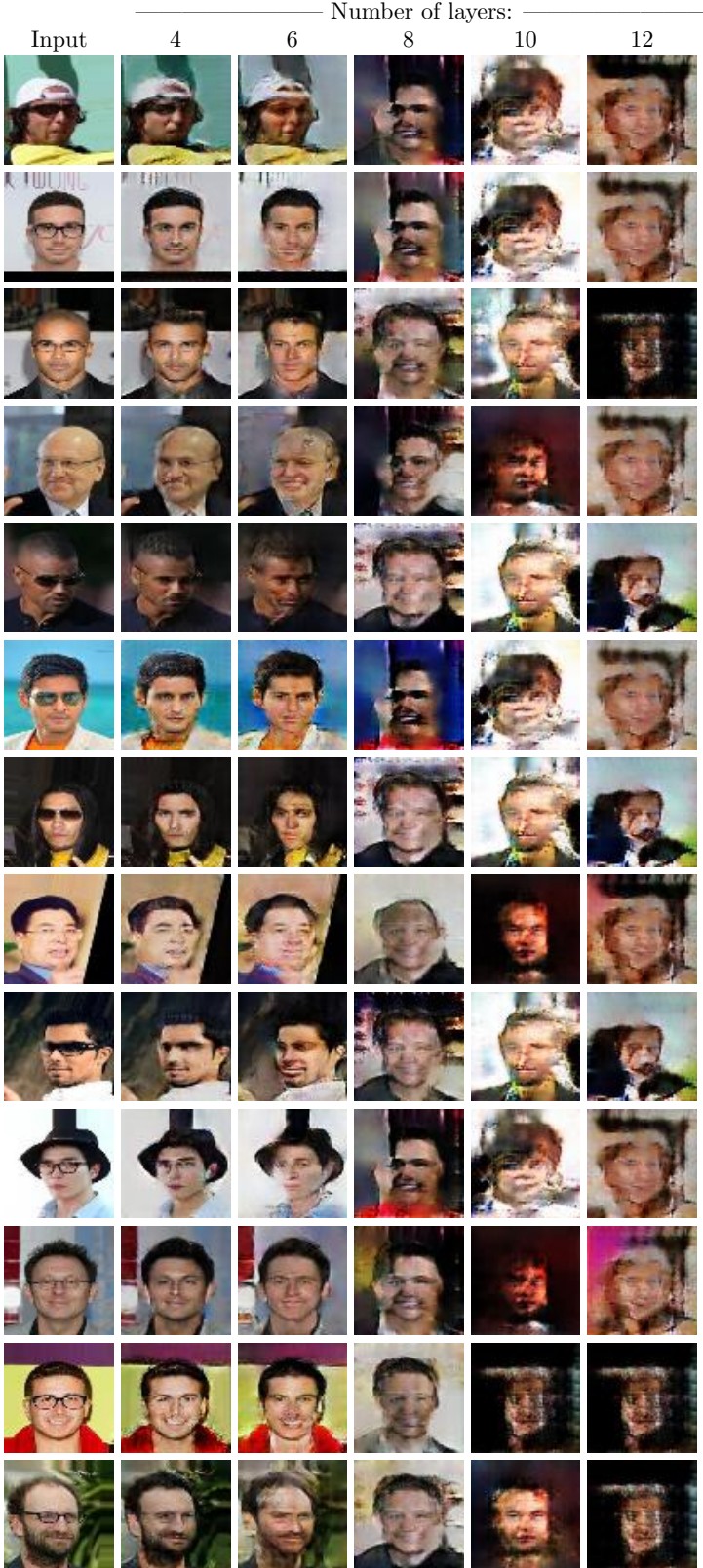

Figure 34: Results for celebA Eyeglasses to Non-Eyeglasses transfer for WGAN with different number of layers.

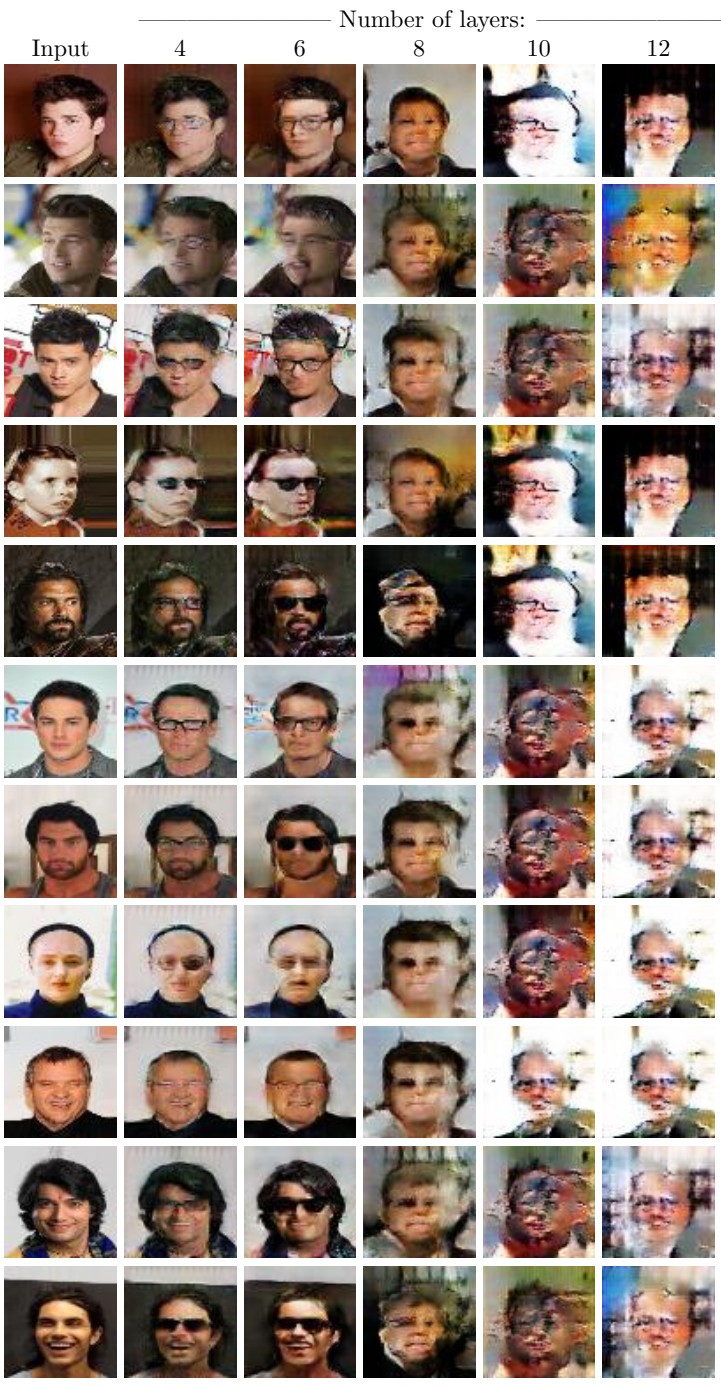

Figure 35: Results for celebA Non-Eyeglasses to Eyeglasses transfer for WGAN with different number of layers.

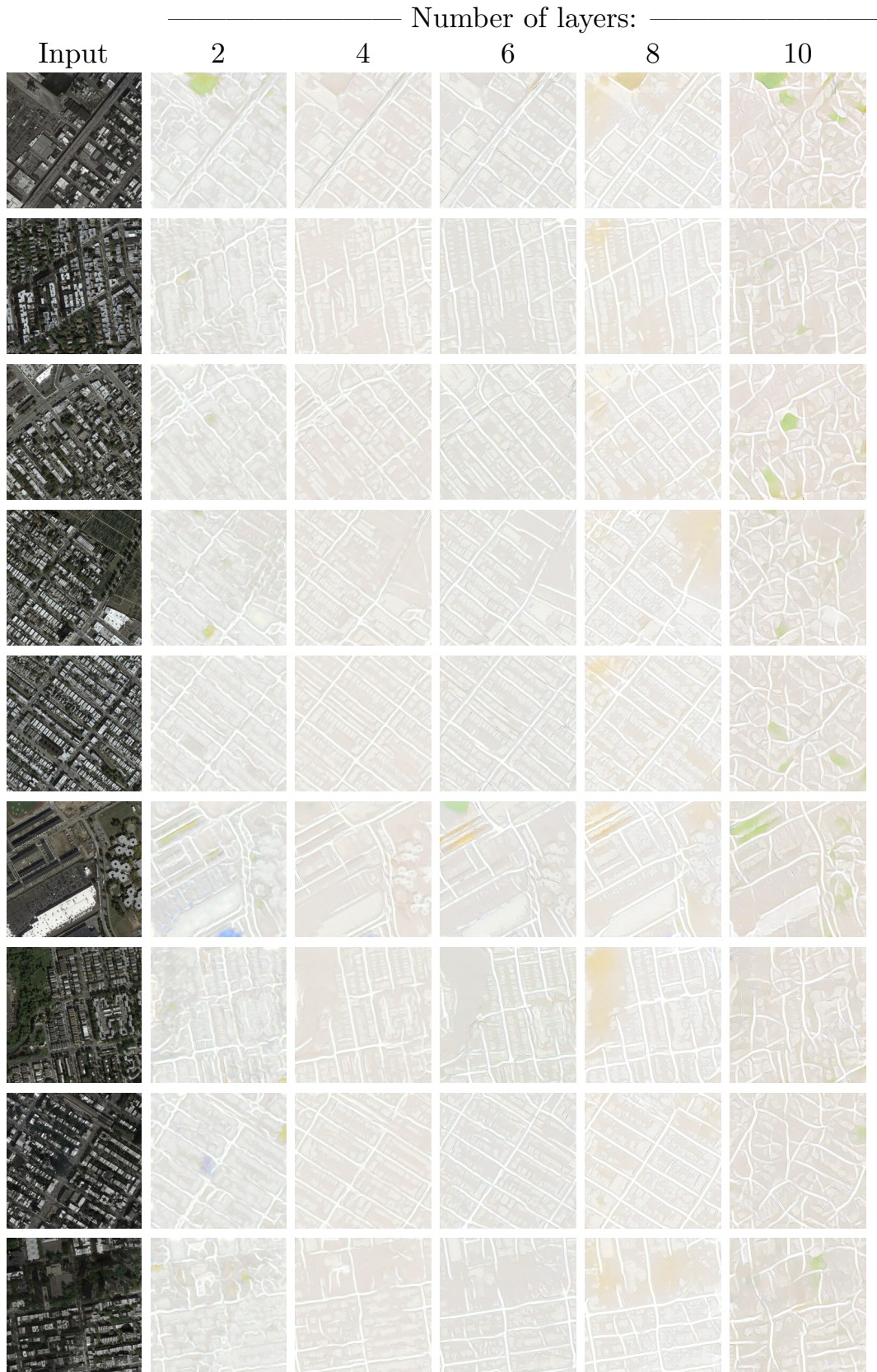

Figure 36: Results for Aerial View Images to Maps transfer for CycleGAN with different number of layers.

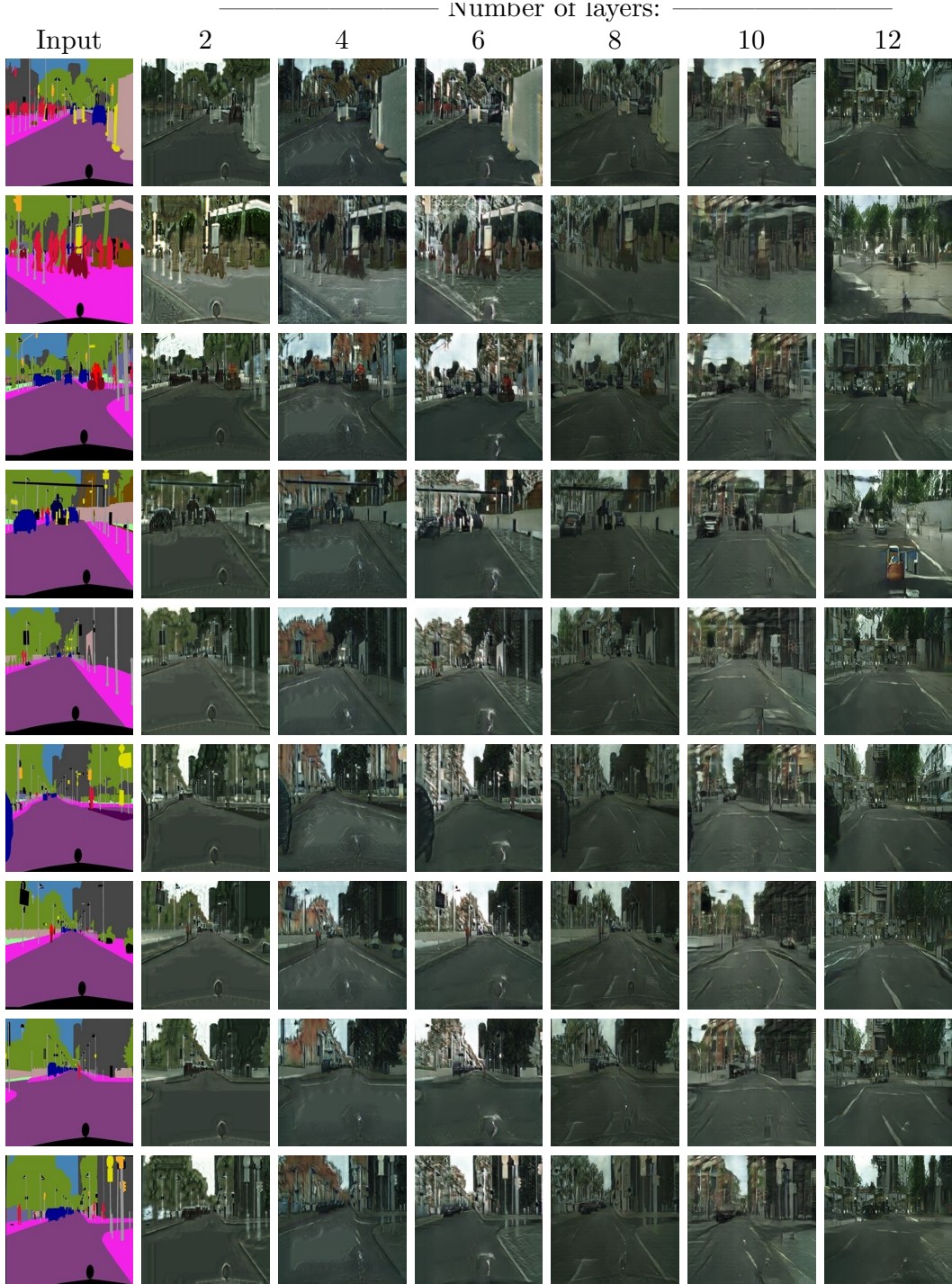

Figure 37: Results for Segmentations to Images transfer for CycleGAN with different number of layers.

