# OpenReview forum: "The Role of Minimal Complexity Functions in Unsupervised Learning of Semantic Mappings"
_ICLR.cc/2018/Conference — Accept (Poster)_

### Official Review · AnonReviewer2 · 2017-11-27
**The paper addresses the problem of learning mappings between different domains without any supervision. It belongs to the recent family of papers based on GANs.**

**Rating:** 7
**Confidence:** 4

**Review:**

The paper addresses the problem of learning mappings between different domains without any supervision. It belongs to the recent family of papers based on GANs.
The paper states three conjectures (predictions in the paper):
1. GAN are sufficient to learn « semantic mappings » in an unsupervised way, if the considered networks are small enough
2. Controlling the complexity of the network, i.e. the number of the layers, is crucial to come up with what is called « semantic » mappings when learning in an unsupervised way.
More precisely there is tradeoff to achieve between the complexity of the model and its simplicity. A rich model is required in order to minimize the discrepancy between the distributions of the domains, while a  not too complex model is necessary to avoid mappings that are not « meaningful ».
 To this aim, the authors  introduce a new notion of function complexity which can be seen as a proxy of Kolmogorov complexity. The introduced notion is very simple and intuitive and is defined as  the depth of a network  which is necessary to  implement the considered function.
Based on this definition, and assuming identifiability (i.e. uniqueness up to invariants), and for networks with Leaky ReLU activations,  the authors prove that if the number of mappings which preserve a degree of discrepancy (density preserving in the text) is small, then the  set of « minimal » mappings  of complexity C   that achieve the same degree of  discrepancy is also small.
This result is related to the third conjecture of the paper that is :
3. the number of the number of mappings which preserve a degree of discrepancy  is small.

The authors also prove a byproduct result stating that identifiability holds for Leaky ReLU networks with one hidden layer.

The paper  comes with a series of experiments to empirically « demonstrate » the conjectures.

The paper is well written. The different ideas are clearly stated and discussed, and hence open interesting questions and debates.

Some of these questions that need to be addressed IMHO:

- A critical general question: if the addressed problem is the alignment between e.g. images and not image generation, why not formalizing the problem as a similarity search one (using e.g. EMD or any other transport metric). The alignment task  hence reduces to computing a ranking from this similarity. I have the impression that we use a jackhammer to break a small brick here (no offence). But maybe that I’m missing something here.
- Several works consider the size and the depth of the network as hyper-parameters to optimize, and this is not new. What is the actual contribution of the paper w.r.t. to this body of work?
- It is considered that the GAN are trained without any problem, and therefore work in an optimal regime. But the training of the GAN is in itself a problem. How does this affect the paper statements and results?
- Are the results still valid for another measure of discrepancy based for instance on another measure, e.g. Wasserstein?


Some minor remarks :
- p3: the following sentence is not clear  «  Our hypothesis is that the lowest complexity small discrepancy mapping approximates the alignment of the target semantic function. »
- p6: $C^{\epsilon_0}_{A,B}$ is used (after Def. 2) before being defined.
- p7: build->built

Section II :
A diagram explaining  the different mappings (h_A, h_B, h_AB, etc.) and their spaces (D_A, D_B, D_Z) would greatly help the understanding.

Papers 's pros :
- clarity
- technical results

cons:
- doubts about the interest and originality


The authors provided detailed and convincing answers to my questions. I thank them for that.  My scores were changed accrodingly.

---

> ### Author Response · Authors · 2017-12-10
> **Thank you for the supportive review and the constructive comments. We have made some modifications and added experiments based on it.**
>
> Thank you for your supportive review, highlighting the technical results, the clarity of the manuscript and the work’s potential in initiating interesting research questions and discussions.
>
> [[if the addressed problem is the alignment between e.g. images and not image generation]]  The alignment problem is between images of A and images of B. GAN is effective in ensuring that the generated images are in the target domain. The alignment problem is how to make sure that the input image in A is mapped (via image generation) to an analog image in B, where this analogy is not defined by training pairs or in any other explicit way (see Sec. 2). Please let us know if this does not answer the question.
>
> [[Several works consider the size and the depth of the network as hyper-parameters to optimize, and this is not new. What is the actual contribution of the paper w.r.t. to this body of work?]]
> The main difference is that when optimizing a supervised loss, as is done in this body of work, the train and validation classification errors, the capacity’s effect on the network’s accuracy, and the network’s generalization capability as a function of the size of the training set are well understood and easy to estimate. In unsupervised learning, changing capacity to reduce the training GAN loss will lead, as we show, to the loss of alignment, and there are no clear guidelines for determining generalization as a function of capacity.
>
> Following the reviewer’s remark, we have added the following text to the paper:
> “Since the method depicted in Alg. 1 optimizes, among other things, the architecture of the network, our method is somewhat related to work that learns the network's structure during training, e.g., (Saxena & Verbeek, 2016; Wen et al., 2016; Liu et al., 2015; Feng & Darrell, 2015; Lebedev & Lempitsky, 2016). This body of work, which deals exclusively with supervised learning, optimizes the networks loss by modifying both the parameters and the hyperparameters. For GAN based loss, this would not work, since with more capacity one can reduce the discrepancy but quickly lose the alignment.”
>
> [[- It is considered that the GAN are trained without any problem, and therefore work in an optimal regime. But the training of the GAN is in itself a problem. How does this affect the paper statements and results?]]  In a subsequent effort to automatically identify a stopping criteria for training cross-domain mappings, we found out that these methods converge and achieve the best results at the last epochs. Therefore, the general issue of GAN instability is not expected to influence our results.
>
> [[- Are the results still valid for another measure of discrepancy based for instance on another measure, e.g. Wasserstein?]] We added additional results in the appendix, where the GAN used is a Wasserstein GAN. As can be seen, our findings seem to hold for WGAN as well.
>
> Thank you for noting a few minor issues with the text. These are corrected in the new version, which also includes the diagram you requested (new Fig. 1).

---

### Official Review · AnonReviewer1 · 2017-11-27
**The paper presents an interesting new analysis about unsupervised learning of (semantic) mappings with new assumptions and theoretical results that could lead to new theoretical developments in representation learning. On the other hand, the paper is dense and some discussions lack of theoretical justification.**

**Rating:** 7
**Confidence:** 2

**Review:**

Quality:
The paper appears to be correct

Clarity:
the paper is clear, although more formalization would help sometimes

Originality
The paper presents an analysis for unsupervised learning of mapping between 2 domains that is totally new as far as I know.

Significance
The points of view defended in this paper can be a basis for founding a general theory for unsupervised learning of mappings between domains.

Pros/cons
Pros
-Adresses an important problem in representation learning
-The paper proposes interesting assumptions and results for measuring the complexity of semantic mappings
-A new cross domain mapping is proposed
-Large set of experiments
Cons
-Some parts deserve more formalization/justification
-Too many materials for a conference paper
-The cost of the algorithm seems high

Summary:
This paper studies the problem of unsupervised learning of semantic mappings. It proposes a notion of low complexity networks in this context used for identifying  minimal complexity mappings which is assumed to be central for recovering the best cross domain mapping. A theoretical result shows that the number of low-discrepancy (between cross-domains) mappings of low complexity is rather small.
A large set of experiments are provided to support the claims of the paper.


Comments:

-The work is interesting, for an important problemin representation learning, while in machine learning in general with the unsupervised aspect.

-In a sense, I find that the approach suggested by algorithm 1 has some connections with structural risk minimization: by increasing k1 and k2 - when looking for the mapping - you increase the complexity of the model searched while trying to optimize the risk which is measured by the discrepancies and loss.
The approach seems costly anyway and I wonder if the authors could think of a smoother version of the algorithm to make it more efficient.

-For counting the minimal complexity mappings, I wonder if one can make a connection with Algorithm robustness of Xu&Mannor(COLT,2012) where instead of comparing losses, you work with discrepancies.

Typo:
Section 5.1 is build of -> is built of

---

> ### Author Response · Authors · 2017-12-10
> **Thank you for the supportive review and the constructive comments. We have made some modifications based on the review.**
>
> Thank you for your supportive review and for the constructive comments, highlighting the significance of the paper. As limitations, the length of the paper and the occasional lack of formalism are mentioned. These issues are interleaved and we have made an effort to balance them by moving the most accurate formal statements to the supplementary appendices. Another aspect of the length is the extensive set of experiments done (as noted in the review) in order to demonstrate the validity of our hypothesis and the consequences it leads to.
>
> Regarding the computational cost of the method, we do not necessarily agree that it is costly.  The training of the networks G1 and G2 is done in a sequential manner where the first step of the method identifies the complexity of G1 that provides alignment. This is done automatically in our method. We believe that a similar effort is being conducted by others when applying their methods, only there, the selection is being done manually. Therefore, the cost of this step is similar to other methods.
>
> The second step of training G2 has a similar complexity. Therefore, our method’s computational cost is just twice the computational cost of what is already being practiced.
>
> Even if the assumption behind our analysis is debatable, the computational cost is a small constant times training one network. In addition, multiple architectures of G1 or of G2 can be trained in parallel.
>
> AnonReviewer1 wonders if a smoother method is conceivable. A smoother method can be based, for example, on skip connections, in which depth varies dynamically, depending on whether a skip connection is used. Then, one can use two networks G1 and G2; G1 is restricted to employ all skips, while G2 is optimized to have low discrepancy, small risk w.r.t G1, and not to use skip connections. This is worth pursuing as a future work.
>
> A connection to Structural Risk Minimization is mentioned in Sec. 6, first paragraph of the original submission. Following the review, a stronger linking to Alg. 1 is  added to the discussion as follows:
> “A major emphasis in SRM is the dependence on the number of samples: the algorithm selects the hypothesis from one of the nested hypothesis classes depending on the amount of training data. In our case, one can expect higher values of k_2 to be beneficial as the number of training samples grows. However, the exact characterization of this relation is left for future work.“
>
> The work of Xu and Mannor proposes a measure of complexity that is related to, but different from, algorithmic stability. We cannot find direct links to our method, which is based on a straightforward notion of complexity. One can combine the two methods together and test, for example, the robustness of the discrepancies. However, we are not yet sure what would be the advantage of doing so.
>
> Thank you for noting the typo in Sec. 5.1. It is now fixed.

---

### Official Review · AnonReviewer3 · 2017-12-12
**interesting topic, the measure of complexity for CNNs does not seem appropriate**

**Rating:** 6
**Confidence:** 4

**Review:**

This  paper is  on ab important topic : unsupervised learning on unaligned data.

The paper shows that is possible to learn the between domains mapping using GAN only without a reconstruction (cyclic) loss. The paper postulates that learning should happen on shallower networks first, then on a deeper network that uses the GAN cost function and regularizing discrepancy between the deeper and the small network.  I did not get the time to go through the proofs, but they handle the fully connected case as far as I understand. Please find my comments are below.

Overall it is an interesting  but long paper, the claims are a bit strong for CNN and need further theoretical and experimental verification. The number of layer as a complexity is not appropriate , as we need to take in account many parameters:  the pooling or the striding for the resolution, the presence or the absence of residual connections (for content preservation), the number of feature maps. More experimentation is needed.



Pros:

Important and challenging topic to analyze and any progress on unsupervised learning is interesting.

Cons:

I have some questions on the shallow/deep in the context of CNN, and to what extent the cyclic cost is not needed, or it is just distilled from the shallow training:

- Arguably the shallow to deep distillation can be understood as a reconstruction cost , since the shallow network will keep a lot of the spatial information. If the deep network match the shallow one , this can be understood as a form of “distilled content “ loss? and the disc of the deep one will take care of the texture , style content? is this intuition correct?

- original cyclic reconstruction constraint is in the pixel space using l1 norm usually, the regularizer introduced matches in a feature space , which is known to produce better results as a “perceptual loss”, can the author comment on this? is this what is really happening here, moving from cyclic constraint on pixels to a  cyclic constraint in a feature space  (shallow network)?

-  *Spatial resolution*: 1) The analysis seems to be done with respect to DNN not to a  CNN. did you study the effect of the architectures in terms of striding and pooling how it affects the results?  I think just counting number of layers as a complexity is not reasonable when we deal with images, with respect to  what preserves contents and what matches texture or style.

2) - Have you tried resnets generators and discriminators  at various depths , with padding so that the spatial resolution is preserved?

- Depth versus width: Another measure that is missing is also the number of feature maps how wide is the network , how does this interplays with the depth?

3) Regularizing deeper networks: in the experiments of varying the length did you see if the results can be stabilized using dropout with deep networks and small feature maps?

4) between training g and h ? how do you initialize h? fully at random ?

5) seems the paper is following implementation by Kim et al. what happens if the discriminator is like in cycle GAN acting on pixels. Pixel GAN rather then only giving a global score for the whole image?

---

> ### Author Response · Authors · 2017-12-22
> **Reply (part 2)**
>
> [[1) did you study the effect of the architectures in terms of striding and pooling how it affects the results?  I think just counting number of layers as a complexity is not reasonable when we deal with images, with respect to  what preserves contents and what matches texture or style. ]] Our experiments show that the number of layers is directly linked to the success obtaining the correct alignment. There is no pooling in the DiscoGAN architecture, the stride is fixed to 2 and kernel size is 4. We therefore did not manipulate these factors. Following the review, we tried a stride of 3. Prediction 2 still holds, but the image quality is slightly worse.
>
> [[2) - Have you tried resnets generators and discriminators  at various depths , with padding so that the spatial resolution is preserved?]] Following the review, we conducted experiments using CycleGan's architecture, which uses a Resnet Generator. We varied the number of layers used in the encoder/decoder part of the architecture, as for DiscoGAN's experiments. Running an experiment on the Aerial images to Maps dataset, we found that 8 layers produces an aligned solution. Using 10 layers produces an unaligned map image with low discrepancy. For fewer than 8 layer, the discrepancy is high and the images are not very detailed. This is exactly in line with our hypothesis.
>
> Adding residual connections to the discriminator of the DiscoGAN experiments, seems to leave the results reported for the original DiscoGAN network mostly unchanged.
>
> [[- Depth versus width: Another measure that is missing is also the number of feature maps how wide is the network , how does this interplays with the depth?]] In this paper we focus on networks that have approximately similar number of neurons in each layer. Therefore, in this case, it is more reasonable to treat the depth as a form of complexity and not the width of each layer.  In a way, depth multiplies the complexity, while width adds to it [1,2]. Therefore, it is a much better determinant of complexity.
>
> Consider this experiment that we conducted following the review:  taking a network that is too shallow by only one layer than what is needed in order to achieve low discrepancy, we double the number of channels (and therefore neurons) in each layer of the architecture and train. The modified network, despite the added complexity, does not achieve a low discrepancy.
>
> [1] Hrushikesh N. Mhaskar and Tomaso Poggio. Deep vs. Shallow Networks: an Approximation Theory Perspective. Analysis and Applications 2016.
> [2] Zhou Lu, Hongming Pu, Feicheng Wang, Zhiqiang Hu, Liwei Wang. The Expressive Power of Neural Networks: A View from the Width. NIPS 2017.
>
> [[3) Regularizing deeper networks: in the experiments of varying the length did you see if the results can be stabilized using dropout with deep networks and small feature maps?]] Following the review, we did the following experiments: we took a network architecture that is too deep by one layer and does not deliver the correct alignment (i.e., it returns another low discrepancy solution) and added to the architecture, at each layer, dropout varying from 10% to 95%. In none of these rates the correct alignment was recovered.
>
> [[4) between training g and h ? how do you initialize h? fully at random ?]]
> h is initialized fully at random.
>
> [[5) seems the paper is following implementation by Kim et al. what happens if the discriminator is like in cycle GAN acting on pixels. Pixel GAN rather than only giving a global score for the whole image?]] Following the review, we run CycleGAN with a varying number of layers (see point 2 above). The results are in full agreement with our hypothesis. In another experiment, one on celebA male to female, we changed  the discriminator of Kim et al. to a Pixel GAN. While the results are of better quality (but also less geometric and more texture like), the alignment is achieved and lost at exactly at the same complexities as with a regular discriminator.

---

> > ### Comment · AnonReviewer3 · 2018-01-11
> > **prediction 1 would benefit from some clarification**
> >
> > Thank you for your reply and clarifications on the role of the 1)architecture (striding etc) 2) fully connected versus CNN 3) depth versus width.
> >
> > I think the ideas in the paper are nice and would benefit from some clarifications putting in it more in context of style and content losses:
> >
> > 1- The claim stated in prediction 1 that :"1. GAN are sufficient to learn « semantic mappings » in an unsupervised way " as understood from the paper, is misleading. Since we still have a reconstruction cost, but where the matching is in the feature space that distills the content. Better guarantees for matching in a deep feature space for reconstruction where analyzed in this paper https://arxiv.org/pdf/1705.07576.pdf
> >
> > I encourage the authors to rephrase those claims and to mention that we still have a style cost function (a la GAN), and content cost function (matching in a feature space rather then in pixels space )
> >
> > 2- on perceptual loss: The matching in the feature space of the low complexity network can be seen as a perceptual loss.
> > - Another option for this matching can be to use the feature map of the discriminator to do this content matching at various depths (this was done in some recent papers)
> > - If one uses VGG to do the content matching this would be a "pretrained perceptual loss" for matching content.
> > Comparing this to the approach of the paper would be interesting.

---

> > > ### Author Response · Authors · 2018-01-11
> > > **reply**
> > >
> > > Thank you for your kind reply.
> > >
> > > Regarding Prediction 1: Predictions 1-3 are made independently of Alg. 1. Therefore, no reconstruction-type loss term is employed in the experiments presented in Sec. 5.1, which validate these predictions. The experiments testing the performance of Alg. 1, which do employ such a term, are the focus of Sec. 5.2. We will make sure this is clearer in the next version.
> > >
> > > The suggested reference will be added. While our work supports the utility of the optimization function of Alg. 1, this reference is a first step in understanding the convergence of this and many other methods that employ distillation.
> > >
> > > Following the reviewer’s comment, we conducted an experiment testing whether employing the perceptual loss [1], instead of the L1 loss of CycleGAN, would lead to maintaining the alignment in deeper networks. This was tested for the task of mapping handbags to shoes, using the imagenet trained VGG-16 network.
> > >
> > > When training a network of depth 10 with this loss, we observe that the solutions are not aligned. We verified that with depth 8 (remember that depths increase by jumps of 2 due to the encoder/decoder structure) an aligned solution is found, same as with the L1 loss. This means that the perceptual loss with the pretrained network cannot eliminate, in the experiment conducted, the inherent ambiguity of deeper networks. With domains that are more closely related and with a more relevant pretrained network, training deeper network this way would probably succeed, as is done with the DTN method of [2].
> > >
> > > Edit (12 Jan): Sample results obtained with the perceptual loss can be seen at  https://imgur.com/hDSusn4 for the task of mapping handbags to shoes. We also ran the perceptual loss experiment for the mapping of celebA males to females and share the results anonymously at https://imgur.com/a/pGF2V
> > > In both cases, the perceptual loss, which employs a pretrained network, results in low discrepancy, i.e., the generated images are in the target class. However, it does not solve the alignment problem for non minimal architectures.
> > >
> > > [1] Johnson, Justin, Alexandre Alahi, and Li Fei-Fei. "Perceptual losses for real-time style transfer and super-resolution." European Conference on Computer Vision, 2016.
> > >
> > > [2] Y. Taigman, A. Polyak, and L. Wolf. “Unsupervised cross-domain image generation.” ICLR 2017.

---

> ### Author Response · Authors · 2017-12-22
> **Reply (part 1)**
>
> Thank you for your supportive review and for the constructive comments, highlighting the significance of the treatment of unsupervised learning.
>
> [[Overall it is an interesting but long paper, the claims are a bit strong for CNN and need further theoretical and experimental verification.]] The paper is indeed quite long, much of the length can be attributed to the need to hold a clear discussion and to the extensive set of experiments done in order to demonstrate the validity of our hypothesis and the consequences it leads to.  The experiments suggested in the review seem to be driven mostly by curiosity regarding alternatives and interest in the boundaries of the claims, and do not seem to point to a major gap in the original set of experiments. We ran most if not all of the requested experiments following the review, see below. In all cases, the results support our findings.
>
> Our original experiments employ the DiscoGAN CNN architecture as is. As noted, the theoretical analysis deals with the case of fully connected networks. Fully connected networks are used as an accessible model on which we prove our theorems. This is similar to other contributions with a theoretical component, in which the analysis is done on simplified models.
>
> For example, in [1], the authors write: “Since a convergence analysis for deep learning is beyond our reach even in the noise free setting, we focus on analyzing properties of our algorithm for linearly separable data, which is corrupted by random label noise, and while using the perceptron as a base algorithm”.
>
> In [2], the authors prove several theorems regarding the expressivity of fully connected neural networks. The experiments validate their theory on convolutional neural networks.
>
> [1] Eran Malach, Shai Shalev-Shwartz. Decoupling "when to update" from "how to update". NIPS, 2017.
> [2] Maithra Raghu, Ben Poole, Jon Kleinberg, Surya Ganguli, Jascha Sohl-Dickstein. On the Expressive Power of Deep Neural Networks. ICML 2017.
>
> Note that in our case (and in [2]), the theoretical model resembles the employed CNNs, since convolutional layers can be written as fully connected layers with restrictions on the weights that arise from locality and weight sharing. Therefore, we can put convolutions in the linear mappings in Eq. 5. In a network with k different convolution types (each specified by stride, kernel size, number of channels), the linear matrices W would be of one of k patterns. The theory would then hold without modifications, except that strictly speaking, the structure of encoder-decoder does not guarantee invertibility. However, as discussed in Sec. 2, invertibility occurs in practice, e.g., autoencoders succeed in replicating the input.
>
> [[- Arguably the shallow to deep distillation can be understood as a reconstruction cost , since the shallow network will keep a lot of the spatial information. If the deep network match the shallow one , this can be understood as a form of “distilled content “ loss? and the disc of the deep one will take care of the texture , style content? is this intuition correct? ]] This intuition is correct, except that the distillation loss is much more restrictive than the cycle (reconstruction) loss. The latter, as we show, is not enough to specify the correct alignment. Indeed, the discrepancy loss of the deeper network makes sure the fine details are correct.
>
> [[- original cyclic reconstruction constraint is in the pixel space using l1 norm usually, the regularizer introduced matches in a feature space , which is known to produce better results as a “perceptual loss”, can the author comment on this?]] The loss between the shallow and the deep network (R_DA[h,g] in Alg. 1) is the L1 loss in our experiments. We do not use the perceptual loss. Since the networks h and g have different architectures it is not immediately clear how to use this loss.

---

### Decision · Program_Chairs · 2018-01-29
**ICLR 2018 Conference Acceptance Decision**

**Decision:**

Accept (Poster)

**Comment:**

The reviewers were generally positive about this paper with a few caveats:

PROS:
1. Important and challenging topic to analyze and any progress on unsupervised learning is interesting.
2. the paper is clear, although more formalization would help sometimes
3. The paper presents an analysis for unsupervised learning of mapping between 2 domains that is totally new as far as I know.
4. A large set of experiments

CONS:
1. Some concerns about whether the claims are sufficiently justified in the experiments
2. The paper is very long and quite dense